



# The role of termite $CH_4$ emissions on ecosystem scale: a case study in the Amazon rain forest

Hella van Asperen[1], João Rafael Alves-Oliveira[2], Thorsten Warneke[1], Bruce Forsberg[3], Alessandro Carioca de Araújo[4,5], and Justus Notholt[1]

[1]Institute of Environmental Physics (IUP), University of Bremen, Otto-Hahn-Allee 1, Bremen, 28359, Germany
[2]Coordenação de Pesquisas em Entomologia (CPEN), Instituto Nacional de Pesquisas da Amazônia (INPA), Av. André Araújo, 2936, Aleixo, AM 69060-001, Manaus, Brazil
[3]Coordenação de Dinâmica Ambiental (CODAM), Instituto Nacional de Pesquisas da Amazônia (INPA), Av. André Araújo, 2936, Petrópolis, AM 69067-375, Manaus, Brazil, (currently at Vermont Agency of Natural Resources, Department of Environmental Conservation, Vermont-USA)
[4]Programa de Grande Escala da Biosfera-Atmosfera na Amazônia (LBA), Instituto Nacional de Pesquisas da Amazônia (INPA), Av. André Araújo, 2936, Aleixo, AM 69060-001, Manaus, Brazil
[5]Brazilian Agricultural Research Corporation (EMBRAPA), Embrapa Amazônia Oriental, Tv. Dr. Enéas Piheiro, s/n, Marco, PA 66095-903, Caixa postal 48, Belém, Brazil

**Correspondence:** Hella van Asperen (v_asperen@iup.physik.uni-bremen.de)

**Abstract.** The magnitude of termite methane ($CH_4$) emissions is still an uncertain part of the global $CH_4$ budget and current emission estimates are based on limited field studies. We present in-situ $CH_4$ emission measurements of termite mounds and termite mound sub samples, performed in the Amazon rain forest. Emissions of five termite mounds of the species *Neocapritermes brasiliensis* were measured by use of a large flux chamber connected to a portable gas analyser, measuring $CH_4$ and $CO_2$.

In addition, the emission of mound sub samples was measured, after which termites were counted, so that a termite $CH_4$ and $CO_2$ emission factor could be determined.

Mound emissions were found to range between 17.0-34.8 nmol mound$^{-1}$ s$^{-1}$ for $CH_4$ and between 1.6-13.5 $\mu$mol mound$^{-1}$ s$^{-1}$ for $CO_2$. A termite emission factor of 0.32 $\mu$mol $CH_4$ g$_{termite}^{-1}$ h$^{-1}$ was found, which is twice as high as the only other reported average value for the Amazon. By combining mound emission measurements with the termite emission factor, colony

sizes could be estimated, which were found to range between 50-120 thousand individuals. Estimates were similar to literature values, and we therefore propose that this method can be used as a quick non-intrusive method to estimate termite colony size in the field.

The role of termites in the ecosystems $CH_4$ budget was evaluated by use of two approaches. Termite mound emission values were combined with local termite mound density numbers, leading to an estimate of 0.15-0.71 nmol $CH_4$ m$^{-2}$ s$^{-1}$ on average

emitted by termite mounds. In addition, the termite $CH_4$ emission factor from this study was combined with termite density numbers, resulting in an estimate of termite emitted $CH_4$ of $\sim$1.0 nmol m$^{-2}$ s$^{-1}$. Considering the relatively low net $CH_4$ emissions previously measured at this ecosystem, we expect that termites play an important role in the $CH_4$ budget of this Terra Firme ecosystem.





## 1  Introduction

Methane (CH$_4$) is the second most important long-lived anthropogenic greenhouse gas, but its natural sources are still not well understood. Anaerobic decomposition processes in wetlands are expected to represent the largest natural CH$_4$ source, but estimates remain a large source of uncertainty (Kirschke et al., 2013; Saunois et al., 2020). Recently, alternative CH$_4$ production mechanisms and their possible important role on ecosystem scale have been proposed, such as the CH$_4$ production by living vegetation (Bruhn et al., 2012; Wang et al., 2014), the CH$_4$ emission due to photo and thermal degradation (Lee et al., 2012), or the transport of anaerobic soil-produced CH$_4$ through wetland trees (Pangala et al., 2015; Rice et al., 2010). An additional known CH$_4$ source in tropical ecosystems is the emission by termites.

Termites (isoptera) can mostly be found between 45°N and 45°S, and are especially abundant in warm ecosystems (Bignell, 2006; Brian and Brian, 1978; Gomati et al., 2011; Wood, 1988). They are highly socialised insects, living in large communities of up to several million individuals (Wood, 1988). Termites are considered 'ecosystem engineers': they are known for decomposing organic substances, and moving and mixing organic and mineral materials, thereby enhancing humus formation, modifying soil structure, and improving soil fertility (Bignell, 2006; Brian and Brian, 1978; Bignell and Eggleton, 2000; Mishra et al., 1980; De Bruyn and Conacher, 1990; Wood, 1988). In addition, they are able to modify their environment to their needs: most termite species live in complex above or (partly) below-ground nests where temperature and moisture remain stable (Bignell, 2019; Noirot and Darlington, 2000; Wood, 1988). Recently, it was shown that termites have a mitigating effect during droughts in tropical rain forests (Ashton et al., 2019). Three main groups of termites can be distinguished, based on their main feeding habits: soil-feeding (humiverous) termites, who can mainly be found in and on the soil, decomposing decayed organic soil material, xylophagous termites, feeding on (decomposed) wood, which can also be found in living trees, and fungus-eating termites, which live in a symbiotic relationship with fungus (Eggleton, 2000; Sanderson, 1996).



CH$_4$ production by termites was first described and measured by Cook (1932). Follow up studies found that methane is produced by almost all termite species, and that its production takes place in the termite gut: in higher termites (dominant in tropical forests, more evolved species with respect to diet and community complexity) CH$_4$ production is caused by symbiotic bacteria, and in lower termites the production is caused by flagellate protozoa (Bignell et al., 1997; Brune, 2018; Lee et al., 1971). In a laboratory experiment Zimmerman et al. (1982) measured the emission strength of individual termites and, by use of termite biomass estimates, presented a global termite emission estimate of 150 Tg CH$_4$ yr$^{-1}$, which was estimated to be 40% of the global natural CH$_4$ emissions. Different estimates followed, resulting in lower estimates, such as by Seiler et al. (1984) of 2-5 Tg yr$^{-1}$, by Fraser et al. (1986) of < 15 Tg yr$^{-1}$, by Khalil et al. (1990) of 12 Tg yr$^{-1}$, and by Martius et al. (1993) of 26 Tg yr$^{-1}$. More recent literature uses estimates in the range of 2-15 Tg CH$_4$ per year (Ciais et al., 2014; Kirschke et al., 2013; Sanderson, 1996; Saunois et al., 2020), which is around 2.5% of the total natural source CH$_4$ emission (Saunois et al., 2020). While on global scale termite emissions can be considered small in comparison to natural sources like wetland emis-







sions ($\sim$147 Tg yr$^{-1}$) or fresh water emissions ($\sim$159 Tg yr$^{-1}$) (Saunois et al., 2020), the question remains what their role can be in the CH$_4$ budget of a local tropical ecosystem.

Estimates of global termite CH$_4$ emissions are based on field and laboratory measurements. To estimate global CH$_4$ termite emissions, most commonly the CH$_4$ emission per termite (mg CH$_4$ termite$^{-1}$ h$^{-1}$) or termite mass (mg CH$_4$ g$^{-1}_{termite}$ h$^{-1}$) is measured, whereby termite mass can either be measured directly or be taken from literature (Sanderson, 1996). The disadvantage of this approach is that termites are removed from their natural environment, thereby possibly changing their emission and behaviour. Another approach is to measure termite nest CH$_4$ emissions in-situ in the field. In this case, emissions are expressed
per mound or nest (mg CH$_4$ mound$^{-1}$ h$^{-1}$). While this method does not disturb the natural environment, correct estimation of termite nest colony size is challenging, wherefore values are hard to convert to emission-per-termite values (Jones et al., 2005). Besides CH$_4$, termite emissions of other gases have also been investigated, such as for CO$_2$, O$_2$, CO, H$_2$, CHCl$_3$, N$_2$O and different hydrocarbons (Cook, 1932; Khalil et al., 1990; Zimmerman et al., 1982). In previous studies, termite CO$_2$ measurements were often performed alongside CH$_4$ emission measurements, and often a clear relationship between CH$_4$ and CO$_2$
emissions was found, of which the ratio is expected to be species dependent (Seiler et al., 1984; Jamali et al., 2013). For termite emitted CO$_2$, reported global estimates are 50 Gt yr$^{-1}$ (Zimmerman et al., 1982), 4 Gt yr$^{-1}$ (Khalil et al., 1990), and 3.5 Gt yr$^{-1}$ (Sanderson, 1996) (1 Gt= 1000 Tg). In addition, Khalil et al. (1990) observed mound CO uptake and emissions, but reported them to be irregular and small. Strong termite mound N$_2$O emissions have also been detected (Brümmer et al., 2009b; Brauman et al., 2015), although they were also found to be very irregular or undetectable (Khalil et al., 1990; Zimmerman et al.,
1982). Brauman et al. (2015) suggested that termite mound N$_2$O emissions occur if N-rich organic matter is available.

Current global CH$_4$ termite emission estimates are based on relatively few studies, and there is still a lack of data on termite CH$_4$ emission rates (Brune, 2018). In addition, existing studies have mostly focused on Australian or Asian species (Eggleton et al., 1999; Fraser et al., 1986; Jamali et al., 2011a, b, 2013; Khalil et al., 1990; Macdonald et al., 1998; Sugimoto et al.,
1998b, a) or African species (Brauman et al., 1992; Brümmer et al., 2009a; Macdonald et al., 1998; Rouland et al., 1993; Sawadogo et al., 2012, 2011; Seiler et al., 1984). To our knowledge, only two studies focused on CH$_4$ emissions of termites in the Amazon (Martius et al., 1993; Queiroz, 2004), and only one study reported CH$_4$ emission values for Amazonian termites (Martius et al., 1993). Martius et al. (1993) performed field measurements on wood-feeding termites by semi-field and laboratory measurements, and suggested that Amazonian termites release more methane than species in other regions. In addition,
for the Amazon, it is expected that most termites are soil-feeding, a group which are expected to be the strongest emitters of CH$_4$ (Bignell and Eggleton, 2000; Brauman et al., 1992).

In this paper, we are presenting a case study performed in a tropical rain forest in the Amazon, where we measured the emission of CH$_4$ and other gases of epigeal (above-ground) termite nests of the species *Neocapritermes brasiliensis*, a soil-feeding





species[1] abundant in the Amazon (Constantino, 1992; Pequeno et al., 2013). In addition we measured the $CH_4$ emission of
countable groups of termites. The goal of our research was twofold. Firstly, we are providing the first $CH_4$ and other gas emis-
sion measurements of the species *N. brasiliensis*, thereby expanding the limited literature on $CH_4$ emissions from Amazonian
termites. Secondly, we are aiming to quantify the role of termite emissions in the $CH_4$ budget of this specific ecosystem, as part
of a larger ecosystem $CH_4$ budget study (van Asperen et al., in preparation). In addition, we are presenting a possible quick
non-intrusive field method to estimate termite colony size in-situ.

## 2 Material and methods

### 2.1 Study site

The study was conducted at the experimental field site Reserva Biológica do Cuieiras – ZF2 (2 °36" 32.67 S, 60 °12"33.48
W, 40-110 m above sea level (a.s.l.), managed by the *Instituto Nacional de Pesquisas da Amazônia* (INPA), located ∼50 km
northwest of Manaus (Brazil). Field site ZF2 consists of plateaus and valleys with typical terra firme forest with tree heights
of 35-40 m on the plateaus and 20-35 m in the valleys. Soils on the plateau are clayey and can be classified as Oxisols and
Ultisols. Soils in the valleys contain more sand and can be classified as Spodosols (Luizão et al., 2004; Zanchi et al., 2014).
The field site has a strong seasonality, with a wet season from December to April, and a dry season from June to September.
Annual average temperatures range between 26-28 °C, and annual average precipitation is around 2400 mm. More information
about the field site can be found in Araújo et al. (2002); Chambers et al. (2004); Luizão et al. (2004); Quesada et al. (2010);
Zanchi et al. (2014). Measurements took place at the end of the wet season (March 2020).

### 2.2 Selection of termite mounds

In the study area, two main trails exist, following the topography from valley to plateau, and termite nests in vicinity of these
trails were inventoried. For practical reasons, only free-standing epigeal (above-ground) nests were considered, from here
on called mounds. Twenty termite mounds were selected for further research, and of each mound the termite species was
determined. For flux chamber measurements, five mounds with the same termite species were selected. For practical reasons,
chosen mounds were in close proximity of each other, and all located in the valley. For comparison, an additional mound was

---

[1]The species *Neocapritermes brasiliensis* is a wood/soil interface feeding species. Species feeding on extremely decomposed wood are in the centre of
the 'wood-soil decomposition gradient' termite classification (Bourguignon et al., 2011), but are classified as soil-feeders according to Eggleton and Tayasu
(2001).





selected of a different species on the plateau. Of each mound, height and perimeter were measured. Termite mound volumes
were estimated by use of the following formula, as also used in Ribeiro (1997) and in Pequeno et al. (2013):

$$V = \frac{\pi HWT}{6} \tag{1}$$

wherein V is the mound volume ($cm^3$), H is the height (cm), W is the width (cm), and T is the thickness (cm) of the mound.
Termite mound surface was estimated by mathematically considering the lower part of the mound as a column, and the upper
part as half a sphere. Details of each mound (dimensions, species, location) are given in Table 1.

### 2.3 Mound flux chamber set up

Collars (stainless steel, 15 cm height, 56.5 cm diameter) were placed around the five selected termite mounds a week before
the start of the measurements. Collars were inserted for approximately 5 cm into the soil/litter layer. In addition, one collar was
placed at some distance from mound 15, containing only soil and litter, representing a blank (non-termite) measurement. From
here on, this collar will be referred to as 'blank measurement'.
A flux chamber was created by use of a 220 L slightly cone-shaped bucket, with a diameter of 57.5 cm. A strip of closed-pore
foam (1 cm x 1 cm x 57.5 cm) was attached over the whole inner perimeter, so that if the bucket was placed on the collar,
the foam strip would seal the part between the bucket and the collar. Two one-touch fittings (1/4 inch, SMC Pneumatics) were
installed on each side of the bucket. The set up (chamber and tubing) were tested for internal emissions of all measured gases.
For CO (see Appendix), an internal emission of <0.014 nmol $s^{-1}$ was found: presented CO flux values are not corrected for
this possible internal emission.

$CH_4$ and $CO_2$ concentrations were measured with a Los Gatos Ultraportable Greenhouse Gas Analyser. The instrument was
connected in a closed loop with the flux chamber (2 x 2 meter PTFE tubing, 1/4 inch). For air circulation, the internal pump of
the Los Gatos was used, with a flow of ~0.3 L $min^{-1}$. The instrument measures concentrations every second; 10-sec averaged
concentrations were saved and used for flux calculations. For each measurement, the flux chamber was closed for 25 minutes,
during which time concentrations were measured continuously. All five mounds were always measured on the same day and in
the same order. Over one week, each mound was measured three times, each time at approximately the same hour of the day.


### 2.4 Flux calculations

Fluxes were calculated as follows. By use of the ideal gas law, mole fractions (ppb/ppm) were converted to molar densities
(nmol/$\mu$mol $m^{-3}$). For chamber temperature, a standard temperature of 25 °C was assumed. For chamber volume (CV), the
termite mound volume (Table 1) was deducted from the bucket volume (220 L).






Fluxes could be calculated as follows:

$$F = \frac{dC}{dt} * \frac{CV}{A} \tag{2}$$

wherein $\frac{dC}{dt}$ is the concentration increase (nmol or $\mu$mol m$^{-3}$ s$^{-1}$), CV the corrected chamber volume (m$^3$), and A the collar area (0.25 m$^2$). Linear regression was used to derive the concentration increase. Given error bars are the propagated standard error of the linear regression slope. All reported fluxes showed an overall $\frac{dC}{dt}$ increase with R$^2$ >0.95. In addition, all fluxes were corrected for dilution effects caused by the filling of the sampling bags (see Appendix). Fluxes are expressed in nmol/$\mu$mol collar$^{-1}$ s$^{-1}$ or nmol/$\mu$mol mound$^{-1}$ s$^{-1}$, depending on whether a termite mound is present in the collar.

**2.5  Soil flux measurements around termite mound**

To quantify the CH$_4$ and CO$_2$ emissions of the soils surrounding the termite mounds, four soil collars were installed around each mound: two soil collars were placed at 20 and 45 cm distance from the mound (distance between mound collar and middle of soil collar), and two additional soil collars were placed on the opposite side of the mound at the same distances. The soil collars were of 20 cm diameter, with a height of 10 cm, and were inserted for 5 cm into the soil. The flux chamber height was

15 cm, so that the soil chamber volume was 4.7 L. The soil chamber had two one-touch fittings on top, to be able to connect the Los Gatos instrument in the same way as to the 220 L-flux chamber. Every soil flux measurement was 4 minutes, and was performed once per mound.

**2.6  Termite mound sub sample emission measurements**

After each last mound flux measurement, a mound sample was taken of approximately 1 L volume. From this, three small

sub samples were taken (volume not determined). When selecting a piece, we tried to look for solid not crumbling pieces, so that the inside of the sub sample was undisturbed. From the sample from mound 19, only one suitable sub sample was found. Each sub sample was placed in a small closed box (12.6 cm x 19.2 cm x 6.8 cm), with two one-touch fittings, functioning as a small closed flux chamber. A blank measurement was made with the small box, and no internal emissions were found. Each mound sub sample was measured with the Los Gatos instrument for 5 minutes, to determine the CH$_4$ and CO$_2$ production in

the chamber over time. After each measurement, the mound sample was carefully broken open and termites were counted, so that the CH$_4$ and CO$_2$ emission per termite could be calculated. The measurements took place next to the mound, and time between sampling and measuring was always less than 15 min.



## 2.7 Termite mass measurement

Termite mass was measured in the Laboratory of Systematics and Ecology of Soil Invertebrates at INPA. 80 living workers
of the species *N. brasiliensis* were weighted by use of a precision scale (FA2104N). Reported individual termite mass is fresh
weight per termite (mg termite$^{-1}$).

## 3 Results

### 3.1 Mound $CH_4$ and $CO_2$ emissions

Mound $CH_4$ emissions ranged between 17.0-34.8 nmol mound$^{-1}$ s$^{-1}$ (Fig. 1), with an average emission of 25.2 nmol mound$^{-1}$
s$^{-1}$. The blank measurements (collar with only soil and litter) showed an average $CH_4$ emission of 1.15 nmol collar$^{-1}$ s$^{-1}$.
Mound $CO_2$ emissions were between 1.6-13.5 $\mu$mol mound$^{-1}$ s$^{-1}$, with an average emission of 8.7 $\mu$mol mound$^{-1}$ s$^{-1}$. The
blank measurements showed smaller $CO_2$ fluxes with an average emission of 0.47 $\mu$mol collar$^{-1}$ s$^{-1}$ (Fig. 1).
The $CH_4$ and $CO_2$ concentration increases inside the closed flux chamber were strongly correlated ($R^2$ >0.95 for each cham-
ber closure). The mound emission $CH_4$/$CO_2$ ratios, shown in Fig. 2, varied between 2.0 and 11.6 * $10^{-3}$ (average ratio: 3.9 *
$10^{-3}$), but showed little variation when data from the blank measurements and data from mound 19 (furthest away from other
mounds) and mound 6 (different species and location) were excluded (average ratio: 2.6 * $10^{-3}$). The smallest mound (mound
19) clearly showed smaller emissions than the other four mounds of the same species, but in general no strong correlation was
found between measured mound $CH_4$ emissions, and mound height ($R^2$=0.08) or volume ($R^2$=0.08), and a small correlation
was found between mound $CO_2$ emissions and mound volume ($R^2$=0.44) and mound height ($R^2$=0.54) (Fig. 3).

*Mound adjacent soil $CH_4$ and $CO_2$ emissions*
Mound adjacent $CH_4$ and $CO_2$ soil emissions were measured around each mound once. For mound 13 and 14, this was done on
the $2^{nd}$ measurement day, for mound 15 and 16, this was done on the $3^{rd}$ measurement day. Due to some practical issues, the
measurements performed around mound 19 could not be used. Figure 4 shows the soil $CH_4$ and $CO_2$ emissions around each
mound, expressed in emission per 0.25 m$^2$: this unit was chosen to be able to compare soil flux measurements to mound (and
blank) flux measurements, measured by the larger collar of 0.25 m$^2$. The small set-in figure in the figures left corner shows the
soil emissions in comparison to the day-specific mound emission. Average soil $CH_4$ and $CO_2$ emissions were respectively 0.5
nmol $CH_4$ 'collar'$^{-1}$ s$^{-1}$ and 1.3 $\mu$mol $CO_2$ 'collar'$^{-1}$ s$^{-1}$ (wherein collar stands for 0.25 m$^2$). The measurements show that
there is no clear emission pattern with increasing distance from the mound, and that mound-adjacent soil fluxes are not strongly
enhanced in comparison to the blank measurements (average blank flux measurements: 1.15 nmol and 0.47 $\mu$mol collar$^{-1}$ s$^{-1}$
for resp. $CH_4$ and $CO_2$).





## 3.2 Termite weight, individual termite emission, and colony size estimation

The living weight of 80 workers was measured to be 0.264 g, which is 3.3 mg per worker. This value is similar to what was

found by Pequeno et al. (2017), who measured 3.0 ($\pm$ 0.4) mg for workers and 6.6 ($\pm$ 0.3) mg for soldiers. The species *N. brasiliensis* has a relatively low soldiers:workers ratio of 1:100 (Krishna and Araujo, 1968). For our calculations we will use an average fresh weight of 3.33 mg termite$^{-1}$ for the species *N. brasiliensis*.

$CH_4$ and $CO_2$ emissions of 13 mound sub samples were measured. For each sub sample, the measured gas production was plotted over the counted termites (Fig. 5). The fitted line has a forced intercept at y=0. For $CH_4$, an emission of 0.0002985

nmol termite$^{-1}$ s$^{-1}$ was found (se=1.77*10$^{-5}$), fitted with an $R^2$ of 0.95 (n=13). For $CO_2$, an emission of 0.0001316 $\mu$mol termite$^{-1}$ s$^{-1}$ was found (se=2.59*10$^{-5}$), with an $R^2$ of 0.68 (n=13). Excluding the out lier (313, 0.81 $\mu$mol s$^{-1}$) gives an $R^2$ of 0.80 (n=12), with a $CO_2$ emission of 0.000076 $\mu$mol termite$^{-1}$ s$^{-1}$ (se=1.14*10$^{-5}$). Converting the emission rates from termite to termite-mass (fresh weight), and from seconds to hourly rates gives a termite emission factor of 0.32 $\mu$mol g$_{termite}^{-1}$ h$^{-1}$ (se=0.02) for $CH_4$ and of 82.2 $\mu$mol g$_{termite}^{-1}$ h$^{-1}$ (se=0.01) for $CO_2$.

By combining the termite emission factors with the termite mound $CH_4$ emissions, colony sizes were estimated. Colony size estimates were based on highest measured emissions and were found to range between 50-120 thousand individuals (Table 4). Population size can also be estimated by use of mound volume or mound external surface. Table 4 shows the population estimates, based on values as given by Lepage and Darlington (2000) for termites in general, and also reports the population estimate based on the work of Pequeno et al. (2013) specifically for the species *N. brasiliensis*.


## 4 Discussion

### 4.1 $CH_4$ and $CO_2$ emissions

Termite $CH_4$ emissions of the soil-feeding species *N. brasiliensis* were found to be 0.32 $\mu$mol g$_{termite}^{-1}$ h$^{-1}$, which is similar to most values found in literature (Table 2, upper part), but two times higher than the average value reported by Martius et al.

(1993) for a wood-feeding species in the Amazon (2.5 $\mu$g $CH_4$ g$_{termite}^{-1}$ h$^{-1}$ = 0.16 $\mu$mol $CH_4$ g$_{termite}^{-1}$ h$^{-1}$). Our emission rate is within the reported range of 0.1-0.4 $\mu$mol g$_{termite}^{-1}$ h$^{-1}$ for soil feeders (Sugimoto et al., 2000). Measured $CH_4$ mound emissions (61-125 $\mu$mol mound$^{-1}$ h$^{-1}$) are in the same range as mound emissions found by previous studies (Table 2).

There is a large variety in type of termite mounds (shape and size are dependent on o.a. species, ecosystem, climate

(Noirot and Darlington, 2000)), explaining the wide range of reported termite mound $CH_4$ emissions (Table 2, middle and lower part). In-situ measurement of termite mound emissions gives information about termite $CH_4$ production under natural conditions, but are unable to distinguish sources and sinks inside the mound. Methanotrophic bacteria are responsible for the $CH_4$ uptake in aerobic soils, and their possible presence in termite mounds was already suggested by Seiler et al. (1984). Other studies have confirmed their presence (Chiri et al., 2019; Ho et al., 2013), and recent studies have been focusing on whether





methanotrophic bacteria are also present in the termite guts, a topic still under discussion (Ho et al., 2013; Pester et al., 2007; Reuß et al., 2015). Different estimates exist on the effect of these bacteria on the net mound flux. Sugimoto et al. (1998a) compared the $\delta^{13}$C of $CH_4$ emitted by mounds to the $\delta^{13}$C of $CH_4$ emitted by termites, and found a fractionation of 0.987 ($CH_4$ emitted by mound/$CH_4$ produced by termites). Other estimates range widely between no observable uptake to very strong uptake rates (up to 80%) (Khalil et al., 1990; Macdonald et al., 1998; Nauer et al., 2018; Sugimoto et al., 1998a). A

more elaborate overview of recent findings on termite mound $CH_4$ uptake processes can be found in Nauer et al. (2018) and Chiri et al. (2019). The role of possible mound $CH_4$ uptake should also be acknowledged for the measurement of individual termite emissions (Table 2, upper part): most literature values, including values from this study, are based on termite incubation in presence of mound material, with ongoing $CH_4$ uptake, wherefore actual termite $CH_4$ emission values might be higher.

Mound $CO_2$ emissions ranged between 6-49 mmol mound$^{-1}$ h$^{-1}$, which fits in the wide range of reported values (Table 3). The relation between the amount of termites and emitted $CO_2$ was found to be 82.2 $\mu$mol g$^{-1}_{termite}$ h$^{-1}$, which is higher than most reported values before. Also here it should be considered that mound material and termites were measured together. Considering the presumably ongoing soil and mound material decomposition processes, the termite-produced $CO_2$ emission rates are likely lower.


The measured $CH_4$ and $CO_2$ emissions of individual mounds showed small variation, such as an $CH_4$ emission increase of 25.3 to 29.5 nmol mound$^{-1}$ s$^{-1}$ at mound 15. One explanation is a variation in colony size (due to foraging activities) or termite activity, driven by temperature or radiation fluctuations (Jamali et al., 2011a; Ohiagu and Wood, 1976; Sands, 1965; Seiler et al., 1984). However, as our measured termite mounds are on the forest floor of a tropical rain forest with relatively

constant temperatures and with only indirect daylight, strong diurnal temperature and radiation patterns are not expected. In addition, since each mound measurement was performed at the same time of the day ($\pm$1 hour), it is unlikely that this variation is caused by a diurnal cycle. Another possibility is that the variation can be explained by the degree of air flow below the soil collar. Preliminary test measurements *without* a collar revealed that the lightest forest breeze already caused strong chamber concentration drops. It is likely that even with a collar not all below-collar air flow was prevented, especially considering the

depth and the porosity of the valley litter layer. This theory is supported by the overall coherent $CH_4$ and $CO_2$ concentrations during chamber closure, which followed the same pattern at all times ($R^2 > 0.99$). In case our set up was subject to minor air transport around the collar, the given mound estimates will be an underestimation of the actual mound fluxes.
An additional possible underestimation is caused by the estimated corrected chamber volume, as used in Eq. (2). In this study, we considered the mound volume as a solid body. A previous study considered the solid nest volume as 10% of the actual

mound volume (Martius et al., 1993), leading to a larger corrected chamber volume, and therefore to larger calculated mound emissions. By use of this approach, average measured emissions would be 32.7 nmol $CH_4$ mound$^{-1}$ s$^{-1}$ instead of 25.2 nmol $CH_4$ mound$^{-1}$ s$^{-1}$.





The mound emission $CH_4/CO_2$ ratio was found to be relatively constant over 4 of the 5 mounds, with an average ratio
of 2.6 $*10^{-3}$. Mound 19, the furthest located from the other mounds, showed relatively low $CO_2$ emissions in comparison
to its $CH_4$ emissions, and showed an average $CH_4/CO_2$ ratio of 8.4 $*10^{-3}$. Values in literature indicate a wide range of
reported $CH_4/CO_2$ ratios (Table 3). However, both Seiler et al. (1984) and Jamali et al. (2013) found little variation between
mounds of the same species, and concluded that the $CH_4/CO_2$ emission ratio is species-specific. Our variation of a factor of
$\sim$3 for the $CH_4/CO_2$ ratio of mound emissions of the same species is of the same magnitude as what was observed in earlier
studies (Seiler et al., 1984; Jamali et al., 2013). Since mound 19 was located in a different part of the valley, it is likely that
the characteristics of the surrounding organic matter were slightly different, affecting the $CH_4/CO_2$ ratio, as also suggested by
Seiler et al. (1984).

## 4.2 Colony size estimate

To estimate colony sizes of (epigeal) nest building termites, different methods exists. Excavation of a termite nest causes a
strong disturbance, initiating an evacuation of the nest. To prevent this, fumigation with methyl bromide is usually applied,
after which termites can be removed from the nest debris by flotation in water, and can be counted. This process is labour
intensive, and can take five persons up to three weeks to finish one nest (Darlington, 1984; Jones et al., 2005). A faster method
is by sub sampling known volumes of the mound, counting the termites in the sub sample, and extrapolating this to the total
mound volume. Termite mounds can have irregular shapes, wherefore volume estimates strongly depend on which volume
estimation approach (hemisphere, cone, column) is used (Jones et al., 2005). So while this method is faster and less intrusive,
it depends strongly on correct volume estimation and it still takes several hours per mound to estimate a colony size.

The population estimation method we tested combined $CH_4$ mound emissions with an in-situ measured termite emission
factor. We estimated colony sizes ranging between 54.6-116.6 $*10^3$ termites per mound. For all mounds, our population es-
timate was in the estimated range based on mound volume or external surface area, as taken from literature equations (Table
4). Comparison to estimates based on a species-specific equation showed differences of maximum 33% (Pequeno et al., 2013):
it should be noted that the relation found between mound volume and termite population by Pequeno et al. (2013) was quite
weak ($R^2$=0.41), and our estimates would fit in the general spread they observed in their data (Pequeno et al., 2013). Interest-
ingly, Pequeno et al. (2013) concluded that mound volume is a weak indicator for population size for nests of the species *N.*
*brasiliensis*, as also indicated by the weak correlation we found between mound volume and mound $CH_4$ emissions (Fig. 3).
The influence of mound $CH_4$ uptake on our population estimate method should be contemplated: mound methanotrophic $CH_4$
uptake likely decreases the net mound $CH_4$ emission, resulting in an underestimation of the colony size when linking it to
termite emission factors, as also suggested by Nauer et al. (2018). However, our termite emission factor was determined inside
small pieces of undisturbed mound material, wherefore the materials $CH_4$ uptake rate was likely only little affected. We hy-
pothesise therefore that our termite emission factor is underestimated to the same degree as our mound emissions, wherefore
both values can still be combined.



Overall, our colony size estimation approach can be considered as a test case for a quick population estimation method. The combination of one mound flux measurement (15 minutes) in combination with 5 sub sample measurements (5x5 minutes) can be performed within 1 hour, including the counting of the termites, being thereby faster than the original methods. Also, the method is applicable to epigeal mounds of all species, independent of internal mound structure (Josens and Soki, 2010) or species characteristics (Pequeno et al., 2013). In addition, the population estimation method we present is not strongly dependent on a correct mound volume estimate, which remains a source of uncertainty (Jones et al., 2005), and which has been shown to be a weak indicator of population size for some species (Pequeno et al., 2013; Josens and Soki, 2010). Furthermore, mounds can also be measured several times in a row before sub sample measurement, so that colony size dynamics over time can be studied noninvasively. A drawback of this method is that it is only applicable for freestanding epigeal mounds, at least with the current type of chamber set up. For a possible follow up study, a direct comparison of population estimation methods is proposed.

### 4.3 Role of termites on ecosystem scale

Mound adjacent soil flux measurements showed no enhanced $CH_4$ and $CO_2$ fluxes in comparison to soils in the blank collar. Additional measurements in the valley showed lower soil $CH_4$ and $CO_2$ fluxes than our blank collar soil fluxes, as also shown by Moura (2012), possibly indicating that our blank collar location might show unrepresentatively high $CH_4$ and $CO_2$ fluxes. However, to avoid overestimation, it was decided to treat termite mounds as very local hot spots, with measured fluxes only representative for the collar area of 0.25 m$^2$.

To estimate the role of termites on ecosystem scale, one approach is to combine mound emission values with termite mound density numbers. A local study reported a density value of 21.6 mounds ha$^{-1}$ for the species *N. brasiliensis* specifically, which deducts to an average $CH_4$ emission of 0.05 nmol m$^{-2}$ s$^{-1}$ caused by mounds of this species alone. Non-species specific mound densities are known to vary strongly between and within ecosystems (Ackerman (2006), Appendix B8). We found five local studies reporting mound (epigeal nest) density values, which were ~100 mounds ha$^{-1}$ (Queiroz, 2004), 193 mounds ha$^{-1}$ (Oliveira et al., 2016), 250 mounds ha$^{-1}$ (Dambros et al., 2016), 60 and 280 mounds ha$^{-1}$ (de Souza and Brown, 1994), and even 760 mounds ha$^{-1}$ (Ackerman et al., 2007). When excluding the strong out lier of 760 mound ha$^{-1}$, the emission of termite mounds on ecosystem scale was estimated to range between 0.15-0.71 nmol m$^{-2}$ s$^{-1}$ for $CH_4$ and between 0.05-0.24 $\mu$mol m$^{-2}$ s$^{-1}$ for $CO_2$.

Since (epigeal) mounds only represent a part of the total termite community, and not the termites located in the subsoil, in dead wood or on trees (arboreal nests), this emission value likely underestimates the actual role of termites on ecosystem scale. Different studies reported ratios of epigeal nest-building colonies in relation to total amount of colonies, such as Constantino (1992) (0.05-0.13), de Souza and Brown (1994) (0.02-0.09), and Martius et al. (1996) (~0.1). However, since colony size can differ strongly between species, these ratios cannot be used to correctly upscale mound $CH_4$ emissions to ecosystem scale. To our knowledge, only Bandeira and Torres (1985) (as given in Martius et al. (1996)) assessed the ratio between nest-building





termite biomass vs total termite biomass, and estimated it to be ∼0.16. Considering the limited literature on this subject, we prefer not too further extrapolate our mound $CH_4$ emission measurements.

A different approach is to use termite biomass estimates and combine them with termite emission factors, a method which is commonly used for global $CH_4$ budget studies (Kirschke et al., 2013; Saunois et al., 2020). For tropical ecosystems, generally a termite biomass of ∼11 g termite $m^{-2}$ is assumed (Bignell and Eggleton, 2000; Kirschke et al., 2013; Sanderson, 1996; Saunois et al., 2020). Considering the previously found value of 0.175 $\mu$mol $g_{termite}^{-1}$ $h^{-1}$ for wood-feeding termites in the Amazon (Martius et al., 1993), and our newly found termite emission factor of 0.32 $\mu$mol $CH_4$ $g_{termite}^{-1}$ $h^{-1}$ for a soil-feeding

termite, a termite-derived ecosystem $CH_4$ emission range of 0.5-1.0 nmol $m^{-2}$ $s^{-1}$ can be calculated. For $CO_2$, our termite emission factor of 82.2 $\mu$mol $CO_2$ $g_{termite}^{-1}$ $h^{-1}$ leads to a termite-derived ecosystem $CO_2$ emission of 0.25 $\mu$mol $CO_2$ $m^{-2}$ $s^{-1}$.

        An overview of the different estimates is given in Table 5. For each of these estimates, it should be considered that our

values are based on measurements from mounds and termites which were all found in the valley, and which were only measured during the wet season. Nevertheless, an exploratory measurement of a small mound of a different species on the plateau (mound 6) indicated $CH_4$ fluxes of a similar magnitude in comparison to a similar-sized mound in the valley (mound 19). Furthermore, additional measurements of the same mounds (and of mound sub samples) during the dry season (September 2020) revealed emission values of the same magnitude (not shown). We therefore do not expect that mound $CH_4$ emissions are

only of importance in the valleys, or only present in the wet season.

        The emission estimate based on mound density, accounting only for epigeal nest building species, is likely underestimating the actual role of termites on ecosystem scale. It therefore makes sense that the other emission estimate (based on termite density) is higher for $CH_4$ as well as for $CO_2$ (Table 5). To put both estimates in perspective, not-termite specific ecosystem

$CH_4$ and $CO_2$ fluxes, measured at this field site during earlier studies, are given. Ecosystem termite $CO_2$ emissions are estimated to range between 0.05-0.25 $\mu$mol $m^{-2}$ $s^{-1}$, which is around ∼1-3 % of the estimated total ecosystem respiration (7.8 $\mu$mol $m^{-2}$ $s^{-1}$, (Chambers et al., 2004)). However, as discussed before, for $CO_2$ as well the 'emission per mound' as the 'termite emission factor' are likely overestimated, wherefore the actual role of termite-emitted $CO_2$ on ecosystem scale is probably smaller. For $CH_4$, termite-derived fluxes are estimated to be between 0.15-1.0 nmol $m^{-2}$ $s^{-1}$. For $CH_4$, as earlier

discussed, we rather expect an underestimation than an overestimation of our termite and mound emission values, wherefore we expect that these ecosystem estimates are conservative. For $CH_4$, it is difficult to judge the role on ecosystem scale, since the earlier measured $CH_4$ flux (above canopy EC measurements, ∼2.0 nmol $m^{-2}$ $s^{-1}$ (Querino et al., 2011)), is a net flux of uptake and emission processes with relatively unknown individual magnitudes. Nevertheless, considering the magnitude of our estimated termite-derived $CH_4$ emissions (0.15-1.0 nmol $m^{-2}$ $s^{-1}$), it is expected that termites play a significant role in this

Terra Firme ecosystem.



### 4.4 Implications for global $CH_4$ termite emission estimate

As described before, $CH_4$ budget studies combine termite density values with termite emission factors to estimate global termite $CH_4$ emissions. In current budget studies, an emission factor of 0.175 $\mu$mol $g_{termite}^{-1}$ $h^{-1}$ (2.8 $\mu$g $g_{termite}^{-1}$ $h^{-1}$)[2] is used for

*'Tropical ecosystems and Mediterranean shrub lands'* (Kirschke et al., 2013; Saunois et al., 2020), which is mainly based on field studies in Africa and Australia (Brümmer et al., 2009a; Jamali et al., 2011a, b; Macdonald et al., 1998; MacDonald et al., 1999). The only termite emission factor measured in the Amazon rain forest is by Martius et al. (1993) (2.5 $\mu$g $g_{termite}^{-1}$ $h^{-1}$) for a wood-feeding termite species, which are expected to emit less $CH_4$ than soil-feeding termites (Bignell and Eggleton, 2000; Brauman et al., 1992). Based on our measurements, we report an emission factor of 0.32 $\mu$mol $CH_4$ $g_{termite}^{-1}$ $h^{-1}$ ($\sim$5.1 $\mu$g $CH_4$ $g_{termite}^{-1}$ $h^{-1}$), which is $\sim$2 times higher than the ecosystem emission factor which is currently used in $CH_4$ budget

studies. Our study points out that termite emissions are still an uncertain source in the $CH_4$ budget, and are especially poorly quantified for the Amazon rain forest. Measurement of $CH_4$ emissions from different termite species, preferably covering species of different feeding or nesting habits, such as wood-feeders or arboreal nest builders, allied with more precise termite distribution and abundance data, would allow more precise estimates and a better understanding of the role of each micro habitat on termite $CH_4$ emission.


### 5 Conclusions

In-situ measurement of termite mound $CH_4$ and $CO_2$ emissions confirmed that mounds can be considered as important local hot spots, playing a considerable role on ecosystem scale. Measured termite mound emissions of the species *N. brasiliensis* were of similar magnitude of what has been observed before for different soil-feeding species, and emissions showed a rela-

tively constant $CH_4/CO_2$ ratio. By performing emission measurements on small groups of termites, we derived a termite $CH_4$ emission factor, so far only the second value reported for the Amazon rain forest. The newly found termite emission factor, measured for a soil-feeding species, is twice as high as the previously reported average value for the Amazon, which was determined for a wood-feeding species. By combining mound and termite emission values, mound colony sizes were estimated, and values were similar to estimates based on literature review. Considering the quick, wide applicable and non-intrusive nature of

this method, we hypothesise that it can be used as a better population estimate approach than the traditional methods, that are either destructive or too specific.

Assessment of the magnitude of termite emitted $CH_4$ on ecosystem scale was attempted by two approaches. Mound emission values were combined with mound density numbers, leading to an estimate of 0.15-0.71 nmol $CH_4$ $m^{-2}$ $s^{-1}$ emitted by mounds on average; since this estimate neglects emission from termite activity outside mounds, the number is likely an under-

estimation. Termite $CH_4$ emission values from this study, and from the only other Amazon field study, were combined with termite density numbers, resulting in an estimate of termite emitted $CH_4$ of 0.5-1.0 nmol $m^{-2}$ $s^{-1}$. Considering the relatively

---

[2]Kirschke et al. (2013) and Saunois et al. (2020) stated a termite emission factor 2.8 (1.0) mg $CH_4$ ($g_{termite}^{-1}$). Correspondence with the authors clarified that a termite emission factor of 2.8 (1.0) $\mu$g $CH_4$ ($g_{termite}^{-1}$ $h^{-1}$) was meant.





low $CH_4$ emissions previously measured at this ecosystem, we expect that termites play an important role in the $CH_4$ budget of this Terra Firme ecosystem.

## Appendix A: Termite mounds: $N_2O$, CO, and $\delta^{13}C$ of $CO_2$

### A1  Methodology

In addition to the direct mound $CH_4$ and $CO_2$ emission measurements (performed with the Los Gatos instrument), mound $N_2O$ and CO fluxes and the $\delta^{13}C$ of the mound $CO_2$ flux were determined by the following method. Three bags (5L inert foil, Sigma-Aldrich) were sampled consecutively from the closed mound flux chamber (see section 2.4). The bags were measured on the same or the consecutive day with a Spectronus FTIR analyser, which can quantify concentrations of $CO_2$, $CH_4$, $N_2O$

and CO, and can determine the $\delta^{13}C$ of $CO_2$. The $\delta^{13}C$ of $CO_2$ measurements of the FTIR analyser have a cross sensitivity for $CO_2$ concentrations, which is well quantified for the $CO_2$ range 380-800 ppm (Hammer et al., 2013). In order to sample air with $CO_2$ concentrations <800 ppm, air samples were taken in the first minutes after chamber closure (2 min, 5 min, 8 min). Out of the 45 taken bag samples, 2 bag samples could not be used.

Before measurement of the bag sample, sample lines were flushed with bag sample air. Air samples were dried by a Nafion dryer and by a column of magnesium perchlorate. Measurements were corrected for pressure and temperature variations as well as for cross-sensitivities (Hammer et al., 2013). For more information on this instrument, please refer to Griffith et al. (2012). For calibration of the instrument, 2 calibration gases were used with values 381.8 $\mu$mol mol$^{-1}$, 2494.9 nmol mol$^{-1}$, 336.6 nmol mol$^{-1}$, 431.0 nmol mol$^{-1}$, -7.95 ‰ for gas 1, and 501.6 $\mu$mol mol$^{-1}$, 2127.0 nmol mol$^{-1}$, 327.8 nmol mol$^{-1}$,

256.7 nmol mol$^{-1}$, -14.41‰ for gas 2, for respectively $CO_2$, $CH_4$, $N_2O$, CO, and $\delta^{13}C$ of $CO_2$.

To calculate the fluxes of $N_2O$ and CO, FTIR-measured bag concentrations of $N_2O$, CO and $CO_2$ were used. For each chamber closure, the $\frac{dN_2O}{dt}$, $\frac{dCO}{dt}$ and $\frac{dCO_2}{dt}$ were calculated, so that ratios the $\frac{dN_2O}{dCO_2}$ and $\frac{dCO}{dCO_2}$ could be derived. To calculate the fluxes of $N_2O$ and CO, the ratios were combined with the in-situ measured mound $CO_2$ flux, as measured by the Los Gatos

instrument. To determine the $\delta^{13}C$ of the $CO_2$ emitted by the termite mounds, Keeling plots were used (Pataki et al., 2003).

### A2  Mound $N_2O$ and CO fluxes

Gas samples taken from the closed flux chamber revealed stable $N_2O$ concentrations between 333.7 and 342.4 ppb. No consistent concentration changes (increase or decrease) during chamber closure were observed, indicating no or very low $N_2O$ emissions. Since the ecosystem, and especially the valleys, are known to be low on nitrogen (Quesada et al., 2010), this is in

agreement with conclusions from a previous study (Brauman et al., 2015).

Chamber CO concentrations ranged between 120 and 220 ppb, and showed a clear uptake on all days and for all mounds,





ranging between -0.04 to -0.78 nmol mound$^{-1}$ s$^{-1}$ (Fig. A1). The 'blank' soil location showed CO emissions between 0.31 and 0.52 nmol collar$^{-1}$ s$^{-1}$. Termite mound uptake has been observed before by Khalil et al. (1990). We expect that

the observed uptake is caused by aerobic CO-oxidising bacteria in the mound, which are also responsible for the CO uptake in (tropical) soils (Conrad, 1996; Kisselle et al., 2002; Liu et al., 2018; Potter et al., 1996; Whalen and Reeburgh, 2001; Yonemura et al., 2000a). Soil CO uptake is dependent on atmospheric CO and therefore often limited by low soil diffusivity (Sun et al., 2018; Yonemura et al., 2000b). The dry porous mound material (Martius et al., 1993) is therefore a suitable place for CO uptake. The observed CO emissions of the blank (soil) collar (0.31-0.52 nmol collar$^{-1}$ s$^{-1}$) are likely caused by the coun-

teracting abiotic CO production, driven by temperature and radiation (King et al., 2012; Lee et al., 2012; Pihlatie et al., 2016; Van Asperen et al., 2015), or by a lesser studied anaerobic biological process (Moxley and Smith, 1998). While we expect that both soil uptake and emission are taking place in the blank soil collar (Kisselle et al., 2002; Liu et al., 2018; Potter et al., 1996; Van Asperen et al., 2015), it is likely that soil uptake is limited due to the low diffusivity of the wet valley soil, wherefore CO production becomes the dominant process.


## A3 $\delta^{13}$C of the mound emitted CO$_2$

Despite our effort to sample air with low CO$_2$ concentrations (cross sensitivity corrections are well determined for CO$_2$ <800 ppm), only 19 out 43 samples showed CO$_2$ concentrations lower than 800 ppm. Nevertheless, for each chamber measurement, a mound-specific $\delta^{13}$C value of the CO$_2$ flux was determined. Figure A2 shows the Keeling plot intercepts, wherein error bars

represent the standard errors of the intercept. Per mound, an average was calculated, which were -38.1‰ (mound 13, se=0.9), -36.2 ‰ (mound 14, se=1.0), -35.7‰ (mound 15, se=0.1), -34.7‰ (mound 16, se=1.4), and -34.7‰ (mound 19, se=1.3). For calculation of these averages, values with a linear regression of R$^2$ <0.99, or values based on a linear regression of only two measurements, were excluded (indicated as dark red squares in Fig. A2). The $\delta^{13}$C of the blank collar (soil) CO$_2$ flux was -33.7 ‰ (se=2.5).


Previous studies have found that mound material can be enriched or depleted in $^{13}$C in comparison to surrounding soils, although differences are usually small (∼1‰) (Siebers et al., 2015; Spain and Reddell, 1996). Studies reporting values on mound emitted $\delta^{13}$C of CO$_2$ have not been found. Based on our measurements, no *significant* difference in the $\delta^{13}$C between mound and soil emitted CO$_2$ was found (-33.7 ‰ (se=2.5) for soil CO$_2$, in comparison to -38.1‰ to -34.7‰ for termite mound emitted CO$_2$). In general, the values were more depleted than values found by De Araujo et al. (2008), who found a $\delta^{13}$C of

-30.1 ‰ for valley litter during the dry season (August 2004). To investigate whether our values are representative for other mounds or soils in the valley, and to investigate whether an isotopic difference exists between mound and soil emitted CO$_2$, more measurements would be needed.



*Author contributions.* HA designed and performed the field experiment, and wrote the paper, JA was responsible for the determination of the termite species, and gave input on the entomology part of the research, BF and AA provided access to the logistics and infrastructure of the field site, JA, TW, BF, AA and JN reviewed and commented on the paper.

*Competing interests.* The authors declare that they have no conflict of interest.

*Acknowledgements.* The study was funded by the DFG-project 'Methane fluxes from seasonally flooded forests in the Amazon basin' (project nr. 352322796). We are thankful for the support of the crew of the experimental field site ZF2, the research station managed by INPA-LBA (National Institute for Amazonian Research (INPA)- The Large Scale Biosphere-Atmosphere Research Program in the Amazon (LBA)). We would also like to express our gratitude to the staff of LBA, for providing logistics, advice, and support during different phases of this research. In addition, we would like to thank Thiago de Lima Xavier and Leonardo Ramos de Oliveira for their advice in planning the technical parts of the experiment. Furthermore, we would like to acknowledge the group 'Department of Aquatic Biology and Limnology' (working group MAUA, INPA) for lending us an additional Los Gatos analyser. Last but not least, we would like to thank Sipko Bulthuis for his assistance and ongoing support during the challenging field measurements days.





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

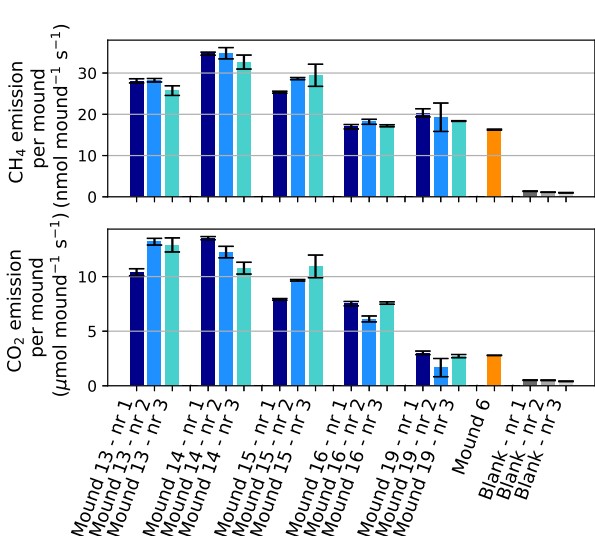

**Figure 1.** $CH_4$ and $CO_2$ emissions of mounds 13 -19 (in valley), of mound 6 (on plateau), and of a blank collar (in valley), expressed in nmol and $\mu$mol mound$^{-1}$ s$^{-1}$, which represents a collar-area of 0.25 m$^2$. All mounds (except mound 6) were measured 3 times during one week, and each series-nr was measured on the same day and in the same order. Error bars are propagated standard errors of the linear regression slope, as described in section 2.4.

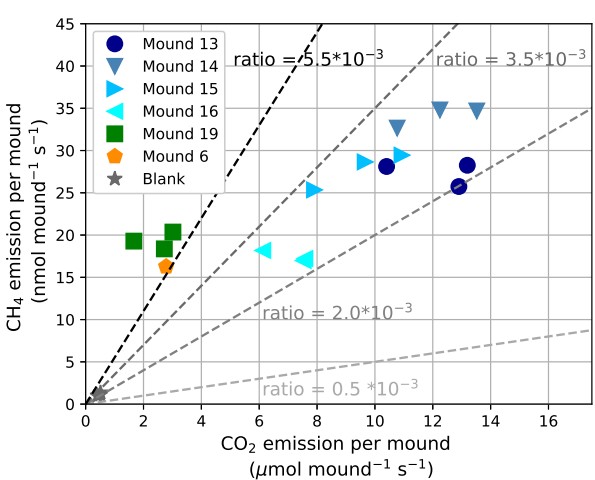

**Figure 2.** Mound $CO_2$ emissions ($\mu$mol mound$^{-1}$ s$^1$) versus mound $CH_4$ emissions (nmol mound$^{-1}$ s$^1$). Dotted lines indicate the different dCH$_4$/dCO$_2$ emission ratios.





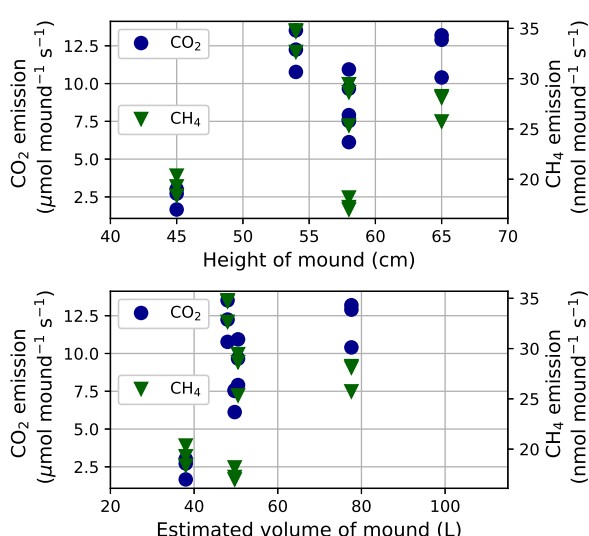

**Figure 3.** Measured mound $CO_2$ emissions (left axis) and mound $CH_4$ emissions (right axis) versus mound height (cm) (upper figure) and estimated mound volume (L) (lower figure). Blue circles indicate $CO_2$ emissions, green triangles indicate $CH_4$ emissions.





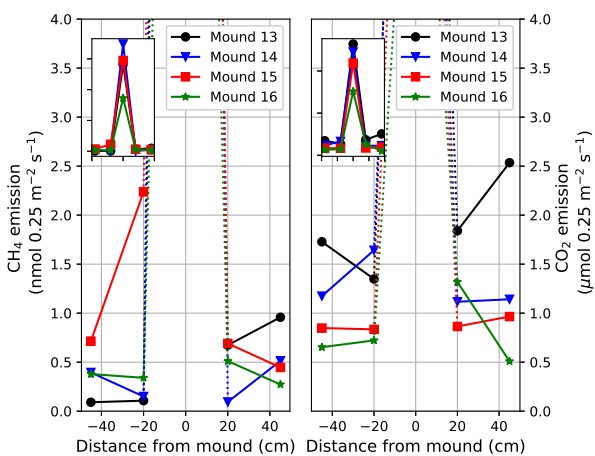

**Figure 4.** Mound-adjacent soil $CH_4$ fluxes (left) and soil $CO_2$ fluxes (right) for mound 13, 14, 15 and 16 expressed in nmol collar$^{-1}$ s$^{-1}$ for $CH_4$ and $\mu$mol collar$^{-1}$ s$^{-1}$ for $CO_2$ (collar is 0.25 m$^2$). Small inserted figures show mound emission of respective mound on same measurement day.

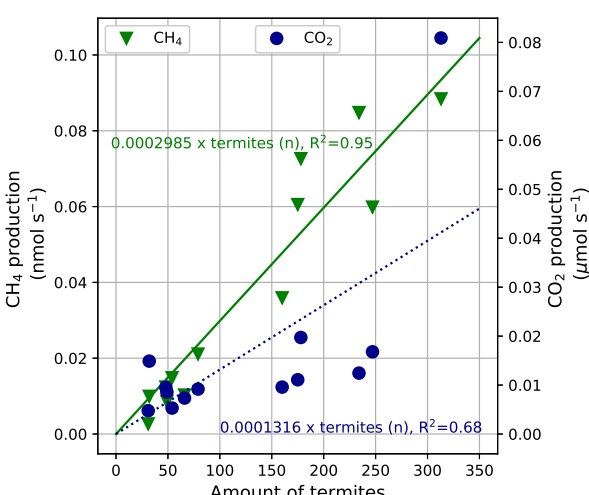

**Figure 5.** $CH_4$ production (left axis, green triangles) and $CO_2$ production (right axis, blue circles), measured in the closed small flux chamber, over counted termites. The lines (green solid for $CH_4$, blue dashed for $CO_2$) represent a linear regression fit with forced intercept at y=0. For $CH_4$, a production of 0.0002985 nmol termite$^{-1}$ s$^{-1}$ (se=1.77*10$^{-5}$) was found, and for $CO_2$, a production of 0.1316 $\mu$mol termite$^{-1}$ s$^{-1}$ (se=2.59*10$^{-5}$) was found.





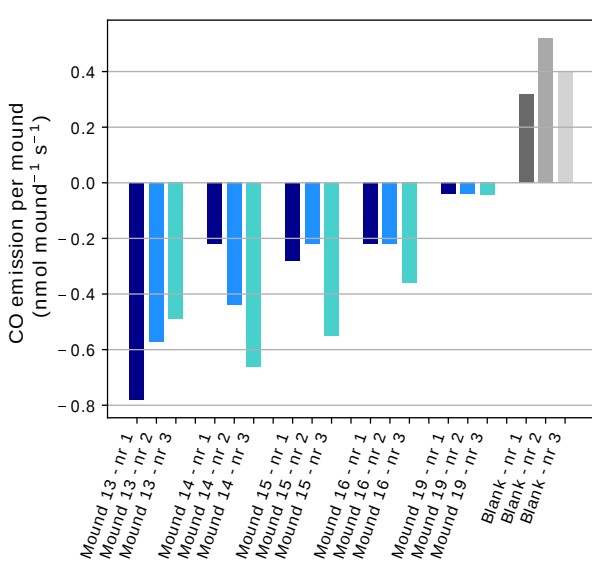

**Figure A1.** CO emissions of mounds 13 -19 (in valley), mound 6 (on plateau) and of a blank collar (in valley), expressed in nmol mound$^{-1}$ s$^{-1}$, which represents a collar-area of 0.25 m$^2$. All mounds (except mound 6) were measured 3 times during one week, and each series-nr was measured on the same day and in the same order.



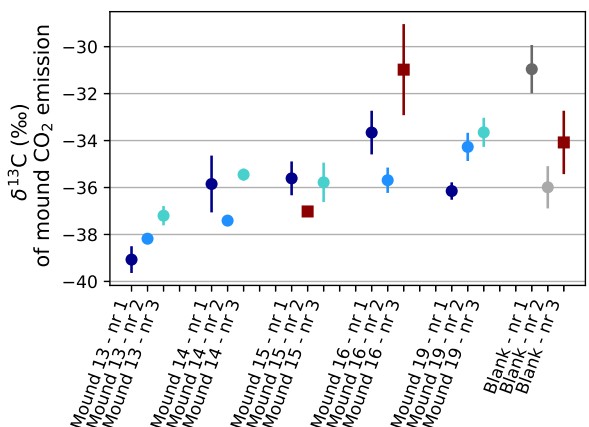

**Figure A2.** $\delta^{13}$C of $CO_2$ emitted by mounds 13 -19 and by soil in a blank collar, derived by use of Keeling plots. Error bars represent standard error of the linear regression intercept. Red squares indicate intercepts based on linear regression fits with $R^2$ <0.99, or based on linear regression with only 2 instead of 3 sample points. All mounds were measured 3 times during one week, and each series-nr was measured on the same day and in the same order.



**Table 1.** Termite mounds: location, dimensions, and observed species. Termite mound volumes were estimated by Eq. (1), and mound surfaces were estimated by mathematically considering the lower part of the mound as a column, and the upper part as half a sphere. In mound 1, two different termite species were found. Mounds indicated in bold were the mounds selected for flux measurements.

| Nr | Location | Height | Perimeter | Estimated mound volume | Estimated mound surface | Species |
|---|---|---|---|---|---|---|
| 1 | valley | 50 cm | 128 cm | | | *Neocapritermes brasiliensis, Heterotermes tenuis* |
| 2 | slope | 45 cm | 145 cm | | | *Neocapritermes brasiliensis* |
| 3 | plateau | 35 cm | 128 cm | | | *Neocapritermes brasiliensis* |
| 4 | plateau | 55 cm | 138 cm | | | *Neocapritermes brasiliensis* |
| 5 | plateau | 45 cm | 148 cm | | | *Rotunditermes bracantinus* |
| **6** | **plateau** | **47 cm** | **99 cm** | **33.8 L** | **4653 cm$^2$** | ***Enbiratermes neotenicus*** |
| 7 | plateau | 50 cm | 160 cm | | | *Enbiratermes neotenicus* |
| 8 | slope | 35 cm | 160 cm | | | *Enbiratermes neotenicus* |
| 9 | valley | 37 cm | 105 cm | | | *Neocapritermes brasiliensis* |
| 10 | valley | 50 cm | 94 cm | | | *Neocapritermes brasiliensis* |
| 11 | valley | 45 cm | 111 cm | | | *Neocapritermes brasiliensis* |
| 12 | valley | 65 cm | 125 cm | | | *Neocapritermes brasiliensis* |
| **13** | **valley** | **65 cm** | **150 cm** | **77.6 L** | **9750 cm$^2$** | ***Neocapritermes brasiliensis*** |
| **14** | **valley** | **54 cm** | **118 cm** | **48.0 L** | **6372 cm$^2$** | ***Neocapritermes brasiliensis*** |
| **15** | **valley** | **58 cm** | **121 cm** | **50.5 L** | **7018 cm$^2$** | ***Neocapritermes brasiliensis*** |
| **16** | **valley** | **58 cm** | **120 cm** | **49.7 L** | **6960 cm$^2$** | ***Neocapritermes brasiliensis*** |
| 17 | valley | 55 cm | 157 cm | | | *Neocapritermes brasiliensis* |
| 18 | valley | 75 cm | 130 cm | | | *Neocapritermes brasiliensis* |
| **19** | **valley** | **45 cm** | **105 cm** | **38.0 L** | **4725 cm$^2$** | ***Neocapritermes brasiliensis*** |
| 20 | slope | 30 cm | 92 cm | | | *Neocapritermes brasiliensis* |



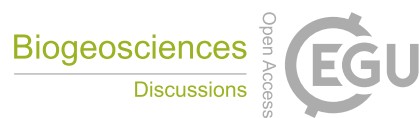

**Table 2.** Overview of literature values for $CH_4$ emission of termites per weight (upper part table), emission per termite mound (middle part table), and emission per area (lower part table). Values from this study for the soil-feeding species *N. brasiliensis* are indicated in bold. If multiple values were found in literature, measurements from higher soil-feeding termite species were chosen to report. For each study, the graph or table where the data was found, is indicated in the first column. a) Sawadogo et al. (2011) reported emissions per dry weight mass. To convert to fresh weight, a formula as reported by Pequeno et al. (2017) was used: log10(fresh weight) =0.51+1.04 log10(dry weight). Assuming a dry weight of ~0.5 mg, gives a fresh weight of 1.57 mg, and a conversion factor of 3.14; b) Mound emissions are divided by collar area of 0.25 m²; c) *Neocapritermes brasiliensis*; d) *Crenetermes albotarsalis, Cubitermes fungifaber, Cubitermes speciosus, Noditermes sp., Procubitermes sp., Thoracotermes macrothorax*; e) *Dicuspiditermes santschii, Dicuspiditermes nemorosus, Pericapritermes semarangi, Procapritermes nr. Sandakanensis, Homallotermes eleanorae, Proaciculitermes sp. A, Pericapritermes nitobei*; f) *Coptotermes lacteus*; g) *Nasutitermes macrocephalus, Nasutitermes corniger, Nasutitermes surinamensis, Nasutitermes sp., Nasutitermes ephratae, Nasutitermes araujoi*; h) *Noditermes sp., Crenetermes albotarsalis, Cubitermes speciosus, Thoracotermes macrothorax, Astratotermes sp.*; i) *Macrotermes bellicosus*; j) *Microcerotermes sp., Globitermes sulphureus, Termes sp., Dicuspiditermes sp.*; k) Sugimoto et al. (1998b), see Appendix 1; l) *Drepanotermes perniger, Nasutitermes magnus, Nasutitermes triodiae, Tumulitermes pastinator, Amitermes laurensis, Coptotermes lacteus*; m) *Bulbitermes sp. C, Dicuspiditermes nemorosus, Dicuspiditermes santschii*; n) *Macrotermes and Odontotermes (Macrotermitinae), Trinervitermes (Nasutitermitinae), Amitermes and Cubitermes (Termitinae), Hodotermes (lower termite)*; o) *Cubitermes fungifaber*; p) *Microcerotermes nervosus, Turnulitermes pastinator, Turnulitermes hastilis, Amitermes meridionalis*.

| Study | Study area | Reported values | Reported unit | Converted values | Converted unit | Species |
|---|---|---|---|---|---|---|
| **Studies reporting $CH_4$ emission per gram termite** | | | | | | |
| **This study, Fig. 5** | **Amazon** | **0.0002985** | **nmol $CH_4$ termite$^{-1}$ s$^{-1}$** | **0.32** | **$\mu$mol $CH_4$ g$_{termite}^{-1}$ h$^{-1}$** | **Soil feeders [c]** |
| Brauman et al. (1992), Table 1 | Congo | 0.39-1.09 | $\mu$mol $CH_4$ g$_{termite}^{-1}$ h$^{-1}$ | 0.39-1.09 | $\mu$mol $CH_4$ g$_{termite}^{-1}$ h$^{-1}$ | Soil feeders [d] |
| Eggleton et al. (1999), Table 4 | Australia | 0.17-0.27 | $\mu$mol $CH_4$ g$_{termite}^{-1}$ h$^{-1}$ | 0.17-0.27 | $\mu$mol $CH_4$ g$_{termite}^{-1}$ h$^{-1}$ | Soil feeders [e] |
| Fraser et al. (1986), Table 2 | Australia | 0.67 | mg $CH_4$ kg$_{termite}^{-1}$ h$^{-1}$ | 0.04 | $\mu$mol $CH_4$ g$_{termite}^{-1}$ h$^{-1}$ | Wood feeders [f] |
| Martius et al. (1993), Table 1 | Amazon | 0.4-4.9 | $\mu$g $CH_4$ g$_{termite}^{-1}$ h$^{-1}$ | 0.03-0.31 | $\mu$mol $CH_4$ g$_{termite}^{-1}$ h$^{-1}$ | Wood feeders [g] |
| Rouland et al. (1993), Table 1 | Congo | 0.53-1.09 | $\mu$mol $CH_4$ g$_{termite}^{-1}$ h$^{-1}$ | 0.53-1.09 | $\mu$mol $CH_4$ g$_{termite}^{-1}$ h$^{-1}$ | Wood feeders [h] |
| Sawadogo et al. (2011), Table 1 | Burkina Faso | 0.30-0.39 | $\mu$mol $CH_4$ g$_{termite}^{-1}$ h$^{-1}$ [a] | 0.10-0.12 | $\mu$mol $CH_4$ g$_{termite}^{-1}$ h$^{-1}$ | Wood feeders [i] |
| Sugimoto et al. (1998a), Table 3 | Thailand | $3.4\text{-}20.3*10^{-8}$ | mol $CH_4$ g$_{termite}^{-1}$ h$^{-1}$ | 0.03-0.20 | $\mu$mol $CH_4$ g$_{termite}^{-1}$ h$^{-1}$ | Soil feeders [j] |
| **Studies reporting $CH_4$ emission per nest or mound** | | | | | | |
| **This study, Fig. 1** | **Amazon** | **17.0-34.8** | **nmol $CH_4$ mound$^{-1}$ s$^{-1}$** | **61-125** | **$\mu$mol $CH_4$ mound$^{-1}$ h$^{-1}$** | **Soil feeders [c]** |
| Khalil et al. (1990), Fig. 4 | Australia | 0.04-0.6 | $\mu$g $CH_4$ mound$^{-1}$ s$^{-1}$ | 9-135 | $\mu$mol $CH_4$ mound$^{-1}$ h$^{-1}$ | Wood feeders [l] |
| MacDonald et al. (1999), Table 4 | Cameroon | 4.5-49 | ng $CH_4$ mound$^{-1}$ s$^{-1}$ | 1-11 | $\mu$mol $CH_4$ mound$^{-1}$ h$^{-1}$ | Soil & wood feeders [m] |
| Martius et al. (1993), Table 1 | Amazon | 0.01-9.4 | mg $CH_4$ nest$^{-1}$ h$^{-1}$ | 0.6-588 | $\mu$mol $CH_4$ nest$^{-1}$ h$^{-1}$ | Wood feeders [g] |
| Seiler et al. (1984), Table 1 | South Africa | 0.07-10.3 | mg $CH_4$ nest$^{-1}$ h$^{-1}$ | 4-644 | $\mu$mol $CH_4$ nest$^{-1}$ h$^{-1}$ | Soil & wood feeders [n] |
| Sugimoto et al. (1998a), Table 3 | Thailand | $4.2\text{-}18.7*10^{-7}$ | mol $CH_4$ nest$^{-1}$ h$^{-1}$ | 0.4-1.9 | $\mu$mol $CH_4$ mound$^{-1}$ h$^{-1}$ | Soil feeders [j] |
| **Studies reporting $CH_4$ emission per area** | | | | | | |
| **This study, Fig. 1** | **Amazon** | **17.0-34.8** | **nmol $CH_4$ mound$^{-1}$ s$^{-1}$** | **245-501** | **$\mu$mol $CH_4$ m$^{-2}$ h$^{-1}$ [b]** | **Soil feeders [c]** |
| Brümmer et al. (2009a), Fig. 5 | Burkina Faso | 2000-5000 | $\mu$g $CH_4$-C m$^{-2}$ h$^{-1}$ | 167-417 | $\mu$mol $CH_4$ m$^{-2}$ h$^{-1}$ | Soil feeders [o] |
| Jamali et al. (2013), Fig. 1 | Australia | 379-~6000 | $\mu$g $CH_4$-C m$^{-2}$ h$^{-1}$ | 32-500 | $\mu$mol $CH_4$ m$^{-2}$ h$^{-1}$ | Wood feeders [p] |
| Queiroz (2004), Table 4 | Amazon | 0.16-0.38 | mg $CH_4$ m$^{-2}$ h$^{-1}$ | 10-24 | $\mu$mol $CH_4$ m$^{-2}$ h$^{-1}$ | unknown |





**Table 3.** Elaboration of Table 2 on studies which reported as well $CH_4$ as $CO_2$ termite emission values. Upper part: $CO_2$ emission of termites per weight; middle part: $CO_2$ emission per termite mound; lowest part: termite $CO_2$ emission per area. Values from this study for the soil-feeding species *N. brasiliensis* are indicated in bold. If multiple values were found in literature, measurements from higher soil-feeding termite species were chosen to report. For each study, the graph or table where the data was found, is indicated in the first column. a) Calculated based on values in study; b) Converted from $CO_2/CH_4$ to $CH_4/CO_2$ and given values are for higher soil-feeding termites; c) Sawadogo et al. (2011) reported emissions per dry weight mass. To convert to fresh weight, a formula as reported by Pequeno et al. (2017) was used: log10(fresh weight) =0.51+1.04 log10(dry weight). Assuming a dry weight of ∼0.5 mg, gives a fresh weight of 1.57 mg, and a conversion factor of 3.14.; d) Mound emissions are divided by collar area of 0.25 m².

| Study | Reported values | Reported unit | Converted values | Converted unit | Ratio $CH_4/CO_2$ (*$10^{-3}$) |
|---|---|---|---|---|---|
| **Studies reporting $CO_2$ emission per gram termite** | | | | | |
| **This study, Fig. 5** | **0.000076** | $\mu$mol $CO_2$ termite$^{-1}$ s$^{-1}$ | **82.2** | $\mu$mol $CO_2$ g$_{termite}^{-1}$ h$^{-1}$ | **3.9** |
| Fraser et al. (1986), Fig. 2 | 4.7 | g $CO_2$ kg$_{termite}^{-1}$ h$^{-1}$ | 294 | $\mu$mol $CO_2$ g$_{termite}^{-1}$ h$^{-1}$ | 0.1 [a] |
| Sugimoto et al. (1998b), Appendix 1 | - | - | - | - | 15 - 91 [b] |
| Eggleton et al. (1999), Table 4 | 1.4-36.4 | $\mu$mol $CO_2$ g$_{termite}^{-1}$ h$^{-1}$ | 1.4-36.4 | $\mu$mol $CO_2$ g$_{termite}^{-1}$ h$^{-1}$ | 10-154 |
| Sawadogo et al. (2011), Table 1 | 59.4-78.4 | $\mu$mol $CO_2$ g$_{termite}^{-1}$ h$^{-1}$ [c] | 19-25 | $\mu$mol $CO_2$ g$_{termite}^{-1}$ h$^{-1}$ | 5.0-5.3 [a] |
| **Studies reporting $CO_2$ emission per nest or mound** | | | | | |
| **This study, Fig. 1** | **1.6 - 13.5** | $\mu$mol $CO_2$ mound$^{-1}$ s$^{-1}$ | **6 - 49** | **mmol $CO_2$ mound$^{-1}$ h$^{-1}$** | **2.4-5.9** |
| Khalil et al. (1990), Fig. 4, Table 3 | 0.05-1 | mg $CO_2$ mound$^{-1}$ s$^{-1}$ | 11 - 225 | mmol $CO_2$ mound$^{-1}$ h$^{-1}$ | 0.12-11 |
| Seiler et al. (1984), Table 1 | 0.03-10.6 | g $CO_2$ nest$^{-1}$ h$^{-1}$ | 2 - 663 | mmol $CO_2$ nest$^{-1}$ h$^{-1}$ | 0.1 - 8.7 |
| **Studies reporting $CO_2$ emission per area** | | | | | |
| **This study, Fig. 1** | **1.6 - 13.5** | $\mu$mol mound$^{-1}$ s$^{-1}$ | **23 -194** | **mmol $CO_2$ m$^{-2}$ h$^{-1}$ [d]** | **2.4-5.9** |
| Jamali et al. (2013), Fig. 3 | 0-1550 | mg $CO_2$-C m$^{-2}$ h$^{-1}$ | 0-129 | mmol $CO_2$ m$^{-2}$ h$^{-1}$ | 2.7-11.0 |
| Brümmer et al. (2009a), Fig. 5 | 100-700 | mg $CO_2$-C m$^{-2}$ h$^{-1}$ | 8-58 | mmol $CO_2$ m$^{-2}$ h$^{-1}$ | - |




**Table 4.** Colony size estimates based on different methods: a) population estimate based on highest measured mound $CH_4$ emission, and combined with the observed emission factor of 0.0002985 nmol $CH_4$ termite$^{-1}$ s$^{-1}$; b) population estimate based on mound volume (given in Table 1), by use of mound termite density values (0.2-5.6 termite cm$^{-3}$ (Lepage and Darlington, 2000)); c) population estimate based on mound surface area (given in Table 1), by use of mound termite surface values (5.6-16.7 termite cm$^{-2}$ (Lepage and Darlington, 2000)); d) Population estimate based on mound volume (given in Table 1), by species-specific volume-population equation of y=47.94*x$^{0.47}$ (x is mound volume (L), y is colony biomass (g)), as given by Pequeno et al. (2013). To convert from population mass to population numbers, a termite mass of 3.33 mg termite$^{-1}$ was taken. Mound 6 contained a different species, wherefore this formula was not applied.

| Nr | Estimated volume | Highest measured emission | Estimated colony size by emission (*$10^3$)$^a$ | Estimated colony size by mound volume (*$10^3$)$^b$ | Estimated colony size by surface area (*$10^3$)$^c$ | Estimated colony size by volume Pequeno et al. (2013) (*$10^3$)$^d$ |
|---|---|---|---|---|---|---|
| 6 | 33.8 L | 16.3 nmol mound$^{-1}$ s$^{-1}$ | 54.6 | 6.5 - 182.3 | 26.1-77.7 | |
| 13 | 77.6 L | 28.3 nmol mound$^{-1}$ s$^{-1}$ | 94.8 | 15.5 - 434.6 | 54.6-162.8 | 111.3 |
| 14 | 48.0 L | 34.8 nmol mound$^{-1}$ s$^{-1}$ | 116.6 | 9.6 - 268.8 | 35.7-106.4 | 88.8 |
| 15 | 50.5 L | 29.5 nmol mound$^{-1}$ s$^{-1}$ | 98.8 | 10.1 - 282.8 | 39.3-117.2 | 90.9 |
| 16 | 49.7 L | 18.2 nmol mound$^{-1}$ s$^{-1}$ | 61.0 | 9.9 - 278.3 | 39.0-116.2 | 90.3 |
| 19 | 38.0 L | 20.4 nmol mound$^{-1}$ s$^{-1}$ | 68.3 | 7.6 - 212.8 | 26.5-78.9 | 79.6 |





**Table 5.** Overview of termite-derived $CH_4$ and $CO_2$ emissions, based on two different approaches. For comparison, the lowest row shows total (not termite-specific) ecosystem $CH_4$ and $CO_2$ flux values, measured at the same field site by previous studies. a) Querino et al. (2011) performed Eddy Covariance (EC) above-canopy $CH_4$ flux measurements, and reported an averaged EC $CH_4$ flux of $\sim$2 nmol m$^{-2}$ s$^{-1}$; b) Chambers et al. (2004) quantified different respiratory $CO_2$ sources in this ecosystem, and estimated the total ecosystem respiration to be 7.8 $\mu$mol $CO_2$ m$^{-2}$ s$^{-1}$.

| | Approach | $CH_4$ (nmol m$^{-2}$ s$^{-1}$) | $CO_2$ ($\mu$mol m$^{-2}$ s$^{-1}$) |
|---|---|---|---|
| Method 1 | Mound per hectare (nr) * emission per mound (mol mound$^{-1}$ s$^{-1}$) | 0.15-0.71 | 0.05-0.24 |
| Method 2 | Termite density (g m$^{-2}$) * termite emission factor (mol g$_{termite}^{-1}$ s$^{-1}$) | 0.5-1.0 | 0.25 |
| Literature | Ecosystem flux values from local studies | $\sim$2 [a] | 7.8 [b] |