# Peer review of "The role of termite CH4 emissions on ecosystem scale: a case study in the Amazon rain forest"

_Biogeosciences, 2020_

## Referee Comment (RC1) · Lukas Kohl (Referee) · 5 Dec 2020

Van Asperen and co-authors studied methane and CO2 emissions by a termite species at an upland site in the amazon basin. They report individual and mound-bsed emission factors compareable to previous studies, and suggest that methane emissions can be employed as as rapid and non-invasive method to estimate mount populations.

Strength: The manuscript addresses a timely and important research question (methane emissions by termites) and provides a much needed data point in a previously understudied areas (termites in the neotropics). The authors followed state of the art measurements at a surely logistically challanging field location. As a bonus, the authors present both a comprehensive literature review and some very rare data on

emissions of other trace gases (N2O, CO) in the appendix. The manuscript is generally well written and surely of great interest for the Biogeosciences readership.

Limitations: Some of the measurements were poorly replicated: Only one control collar was placed at distance from termite mounts, and for termite weight estimates, only one measurement is presented.

Possible improvement: - While the manuscript is generally very well written, I would encourage the authors to focus on editing the discussion section, which reads less easily than the rest of the manuscript. Some of this could be done by shortening and streamlining this section, which is rather long and at times meandering. - The authors could also improve the quality figures and tables (see below), most importantly remove the grid lines from the figures for easier readability.

Overall, this is a very nice contribution and it was a pleasure to review it!

L48: 'which is around' - aprroximately instead of around, also better state the range in % as well given that 2-15 Tg is quite a wide range. L60: 'termite CO2 measurements' - measurements of termite CO2 emissions L64: to avoid mixing weight units (gramm and tons), Pg instead of Gt L119: what material was your chamber built out of? L130: when were your measurements conducted (date, in what season?) L134: 'molar density': concentration L142: 'increase': concentration change, as you could see uptake L176-181: The section could be improved. L195: can you state an uncertainty of this weight per individual? L217: no need to state the original unit here, just state the values converted to the unit used in your study. L223-235: I recommend streamlining/shortening this segment. Acknowledging mount uptake is important, but it's not the focus of your study and comes out of left field here. Focus on why this is important to understanding your results. L237-241: I would move this comparison with literature data up to L215-219. L243 ff: To be honest, these variations among individual measurements look pretty trivial to me and may not need such extensive discussion (which ends up questioning your measurements.). L249-254: This can be tested by looking at

the concentration curves wihtin individual closures. If a relevant air exchange between chamber and ambient air occurred, concentrations should be non-linear (d[CH4]/dt decreasing over time, following a y=a + b*eˆ(-c*t) function). If this is the case, fluxes should be calculated by fitting such an exponential function and calculating the slope as d[CH4/dt] at t=0. L280ff: it would be good to add an uncertainty range to your population estimates L288: 'contemplated': considered? L291-292: 'hypothesize': don't use hypothesize that for claims you do not test. 'It is therefore likely that ..' L303: 'drawback': disadvantage L304-305: 'is proposed': by whom? The authors? If that's that's the case, say so (ok, sorry for the snarky tone. Use active voice here - 'We propose a follow-up study to directly compare') L311: 'it was decided': same here, use active voice: 'we decided .. to avoid overestimating ..'

L418-419: 'indicating no or very low N2O emissions': Can you provide an uncertainty range for that estimate (e.g., limit of detection for fluxes?)

Figures: remove grid lines (counterintuitively, this makes figures easier to read), place ticks inwards Fig 1: remove 'per mount' on the y axis, it's redundant with the unit on that axis. Fig 4: A broken axis might work better than the inserts here (if you keep the inserts, state the y axis scale). The figure could also be simplified by showing the means + SD of the four mounts instead of values for individual mounts. Also, I think the direction in which you placed the soil collars from the mount wasn't chosen deliberately, so your x axis could be just 'distance from the center of the mount', combining your flux measurements at the same distance at either side of the mount. Fig 5: number instead of amount

Tables 2-3: I recommend combining Table 2 and 3 after removing reported value and reported unit (these can be placed in a supplement) to keep the table easier to read. State the unit of the converted values in the table header. This leaves the following columns: [Study] [Study area] [CH4 emission (state units)] [CO2 emission (state units)] [CH4:CO2 ratio (state units)] [Species]. Such a table would give a much better overview.

Place footnotes as footnotes and not in the table caption.

---

## Referee Comment (RC2) · Anonymous Referee #2 · 22 Dec 2020

This manuscript presents a well thought out study to quantify methane emissions by termite in the Amazon rain forest. The authors reviewed the literature extensively and compared/discussed with their findings. I have some comments that I think will make the study more valuable. 1. Please provide the estimate of CH4 emissions by termite and put in context with the overall CH4 budget globally or in the Amazon. This manuscript presents CH4 emission factors only. Without knowing how many termite mounds in Amazon, it's difficult to imagine the scale of the global CH4 budget. I think this is one of a key messages for readers. 2. The first sentence in the Introduction section, it says "Methane (CH4) is the second most important long-lived anthropogenic greenhouse gas." I think CH4 has been recognized to be "short-lived" climate pollutant. 3. In the Introduction section, Line 35, it says "Recently, it was shown that termites

have a mitigating effect during droughts in tropical rain forests". Please elaborate what mitigating effect. 4. In the Introduction and in Appendix, the authors touched on N2O emissions from termite but didn't give conclusive results. 5. Section 2.3, Line 129, LGR GHG analyzer was mentioned to be the instrument deployed to quantify CH4 emissions in flux chambers. I think authors should add brief instrument performance specifications and details of what calibration and drift evaluation have been done in Amazon. While the absolute CH4 concentrations in flux chamber measurements are not very critical, since it's to measure the CH4 concentration increase, but the manuscript does not provide the measured concentrations and jumped directly to the emission factor estimates. For example, LGR UGGA precision is about 2 ppb. Does it perform the same in Amazon? Also, what CH4 concentration increments measured in the flux chambers? If it was only 2 ppb, then that data would not be useful. I think it should be many times more than the instrument precision and drift. 6. Well-designed flux chambers should have a small mixing fan or internal distribution tubing to quantify fluxes. Section 2.5, describes how LGR sampling tubes were connected on top of a 220 L chamber, if the air inside is not well mixed, the two fittings on top of the chamber may not detect CH4 at the bottom of the chamber. 7. Appendix A1 and A2 talk about N2O calibrations and measured concentrations. The measured N2O concentrations are outside of the calibration range. While the lower range (333.7 ppb) is similar to NOAA's measurements in Brazil, the manuscript does not provide the FTIR instrument precision and therefore, it's difficult to determine whether the detected range (333.7-342.4 ppb) is within instrument drift or it's actually an increment of N2O. I don't think the authors can conclude there isn't N2O emissions.

---

## Referee Comment (RC3) · Anonymous Referee #3 · 7 Jan 2021

This study presented a global interesting issue of termite CH4/CO2 emission in an Amazonian tropical rainforest. As a case study, this in-situ measurement of termite mound emissions provided information about termite CH4/CO2 production under natural conditions, it will contribution some knowledge to Biogeosciences. However, the field experiment was not well designed, and the limited data was not well analyzed. I would like to encourage the authors to revise the manuscript following my comments.

General comments: 1. "The blank measurements (collar with only soil and litter) showed an average CH4 emission of 1.15 nmol collar-1 s-1" (L175) means the forest soil was a VERY LARGE CH4 SOURCE (4.6 nmol m-2 s-1 or 23.2 kg CH4 ha-1 y-1). It was a FUNDAMENTAL PROBLEM! Actually, the "blank" soil should be CH4 sink. Even "1.15 nmol collar-1 s-1" was "-1.15 nmol collar-1 s-1", the soil CH4 sink of

"-23.2 kg CH4 ha-1 y-1" was an unbelievable large value. 2. An early study in a Southeast Asian tropical forest showed that the populations of termites was 3,000 – 4,000 m-2, 60% of which being wood-feeding termites and 30% being either litter-feeding or humus-feeding species (Chiba, 1978). This population density was supported by many recent studies showed in this manuscript (L356-358). Why this study did not include the major termite species (wood-feeding)? 3. Large variations in both CH4 and CO2 emissions (figure 1; L221-222, L240) among the mounds suggest that the five applicates (mounds) was not enough to represent the ecosystem level CH4 and CO2 emissions. From your statement (2.6: sub sample), I would guess that your CH4/CO2 flux measurements were conducted for all the 19 mounds but not only 5 mounds (figure 1). If my guess is correct, the authors should explain (in the Method) the reasons for not including the data from other mounds, for example, the other mounds were not active mounds. Moreover, the authors should explain why the sub sample experiment was only conducted for one mound (L161: "only one sub sample was found suitable from the all 19 mounds"). 4. In tropical forest, the termite mounds have different size and different shapes, and many are already not active mounds. This study only selected the relatively small size of termite mound (Table 1), thus it is not surprised that the authors gave the conclusion of weak correlation between CH4 emission and mound size (3.1; Fig. 3). 5. This in-situ measurement could not be able to partition the contribution of mound soil (CO2 source but CH4 sink) from termite, thus the termite CH4 emission could be underestimated but termite CO2 emission could be overestimated. The results should be calibrated, because the structure and nutrients of the mound-soil are different from the normal soil (blank soil in this study). 6. Chamber volume (CV in L145; L159-163, L258-262) is a major parameter for calculation of flux rate (Equation 2). If the exact volume of the sample mound was not known, means CV was not known, based on the calculation using equation 2, the estimated both CH4 and CO2 fluxes (Table 2, 3; L218-222, L241-243) would be absolutely under- or over-estimated. 7. In my experience, this R2 > 0.95 (L178 and other places) was non-believable. The chamber was relatively (or very) large (220 L), UGGA internal (pump) flow was only

about 350 mL min-1, the chamber air could not be mixed without installing one or two micro fans inside the chamber, because it takes about 630 min to replace the chamber air if only depending on the UGGA internal pump. Particularly, the chamber was about 1 m high, the emitted CH4 and CO2 was not be able to be mixed inside the chamber if only depending on both diffusion and UGGA internal pump. Moreover, based on the bag sampling (A1), CH4 flux could be estimated. The authors are suggested to compare the result with that of mound chamber and sub sample. 8. Data was too limited; I strongly encourage the authors to show the data measured in the dry season (L348-350) and compare it with that of wet season showed in this manuscript. 9. Overall, using the limited data to scale up it to ecosystem (4.3) and global (4.4) levels would no doubt create large uncertainty. The authors are suggested to cancel or at least shorten these two issues.

Minor Comments: L10 (L211, L284): Reads are easily be confused by the colony size and population, also the colony size of 50-120 thousands individuals and 54.6-116.6*103 termites per mound should be unified. L120: Change "mount 15" to "mount #15". L120: Only one control (blank) made this result (also see above) weaker. L130: The distance between the UGGA and chamber was 2 m. L131: It is about 350 mL/min (from LGR). L150-157 (2.5): Soil flux chamber had no mixing fan would have the same problem with the mound chamber (see above) L177: Soil CO2 emission of 0.47 $\mu$mol collar-1 s-1 (1.87 $\mu$mol m-2 s-1) was too small. The authors are suggested to compare it with other studies in tropical forests. L187-189: Move to the Method, and L189-192 move to the caption of Figure 4. L252-257: The statement of "air flow below the soil collar" does not make sense. Equation 2: not completed; missed chamber pressure and chamber temperature. L311: The statement of "Mound adjacent soil flux measurements showed no enhanced CH4 and CO2 fluxes in comparison to soils in the blank collar" does not consist with the results. For example, adjacent CO2 flux (1.3) was almost three times of blank soil (0.47). L337: 11 g is the maximum value; the variation range should be listed. Consequentially, the following value of 0.5-1.0 nmol m-2 s-1 was overestimated. L415: Check the grammar. A3: Shorten or discuss the

scientific meaning of ïĄď13CO2 in this study. Unify the concentration unit of ppm and $\mu$mol mol-1.

Please also note the supplement to this comment:
https://bg.copernicus.org/preprints/bg-2020-384/bg-2020-384-RC3-supplement.pdf

[Figure]

**Supplement:**

This study presented a global interesting issue of termite $CH_4/CO_2$ emission in an Amazonian tropical rainforest. As a case study, this *in-situ* measurement of termite mound emissions provided information about termite $CH_4/CO_2$ production under natural conditions, it will contribution some knowledge to Biogeosciences. However, the field experiment was not well designed, and the limited data was not well analyzed. I would like to encourage the authors to revise the manuscript following my comments.

General comments:
1.  "The blank measurements (collar with only soil and litter) showed an average $CH_4$ emission of 1.15 nmol collar$^{-1}$ s$^{-1}$" (L175) means the forest soil was a VERY LARGE $CH_4$ SOURCE (4.6 nmol m$^{-2}$ s$^{-1}$ or 23.2 kg $CH_4$ ha$^{-1}$ y$^{-1}$). It was a FUNDAMENTAL PROBLEM! Actually, the "blank" soil should be $CH_4$ sink. Even "1.15 nmol collar$^{-1}$ s$^{-1}$" was "-1.15 nmol collar$^{-1}$ s$^{-1}$", the soil $CH_4$ sink of "-23.2 kg $CH_4$ ha$^{-1}$ y$^{-1}$" was an unbelievable large value.
2.  An early study in a Southeast Asian tropical forest showed that the populations of termites was 3,000 – 4,000 m$^{-2}$, 60% of which being wood-feeding termites and 30% being either litter-feeding or humus-feeding species (Chiba, 1978). This population density was supported by many recent studies showed in this manuscript (L356-358). Why this study did not include the major termite species (wood-feeding)?
3.  Large variations in both $CH_4$ and $CO_2$ emissions (figure 1; L221-222, L240) among the mounds suggest that the five applicates (mounds) was not enough to represent the ecosystem level $CH_4$ and $CO_2$ emissions. From your statement (2.6: sub sample), I would guess that your $CH_4/CO_2$ flux measurements were conducted for all the 19 mounds but not only 5 mounds (figure 1). If my guess is correct, the authors should explain (in the Method) the reasons for not including the data from other mounds, for example, the other mounds were not active mounds. Moreover, the authors should explain why the sub sample experiment was only conducted for one mound (L161: "only one sub sample was found suitable from the all 19 mounds").
4.  In tropical forest, the termite mounds have different size and different shapes, and many are already not active mounds. This study only selected the relatively small size of termite mound (Table 1), thus it is not surprised that the authors gave the conclusion of weak correlation between $CH_4$ emission and mound size (3.1; Fig. 3).

5. This *in-situ* measurement could not be able to partition the contribution of mound soil ($CO_2$ source but $CH_4$ sink) from termite, thus the termite $CH_4$ emission could be underestimated but termite $CO_2$ emission could be overestimated. The results should be calibrated, because the structure and nutrients of the mound-soil are different from the normal soil (blank soil in this study).

6. Chamber volume (CV in L145; L159-163, L258-262) is a major parameter for calculation of flux rate (Equation 2). If the exact volume of the sample mound was not known, means CV was not known, based on the calculation using equation 2, the estimated both $CH_4$ and $CO_2$ fluxes (Table 2, 3; L218-222, L241-243) would be absolutely under- or over-estimated.

7. In my experience, this $R^2 > 0.95$ (L178 and other places) was non-believable. The chamber was relatively (or very) large (220 L), UGGA internal (pump) flow was only about 350 mL $min^{-1}$, the chamber air could not be mixed without installing one or two micro fans inside the chamber, because it takes about 630 min to replace the chamber air if only depending on the UGGA internal pump. Particularly, the chamber was about 1 m high, the emitted $CH_4$ and $CO_2$ was not be able to be mixed inside the chamber if only depending on both diffusion and UGGA internal pump. Moreover, based on the bag sampling (A1), $CH_4$ flux could be estimated. The authors are suggested to compare the result with that of mound chamber and sub sample.

8. Data was too limited; I strongly encourage the authors to show the data measured in the dry season (L348-350) and compare it with that of wet season showed in this manuscript.

9. Overall, using the limited data to scale up it to ecosystem (4.3) and global (4.4) levels would no doubt create large uncertainty. The authors are suggested to cancel or at least shorten these two issues.

Minor Comments:

L10 (L211, L284): Reads are easily be confused by the colony size and population, also the colony size of 50-120 thousands individuals and 54.6-116.6*$10^3$ termites per mound should be unified.

L120: Change "mount 15" to "mount #15".

L120: Only one control (blank) made this result (also see above) weaker.

L130: The distance between the UGGA and chamber was 2 m.

L131: It is about 350 mL/min (from LGR).

L150-157 (2.5): Soil flux chamber had no mixing fan would have the same problem with the mound chamber (see above)

L177: Soil $CO_2$ emission of 0.47 µmol collar$^{-1}$ s$^{-1}$ (1.87 µmol m$^{-2}$ s$^{-1}$) was too small. The authors are suggested to compare it with other studies in tropical forests.

L187-189: Move to the Method, and L189-192 move to the caption of Figure 4.

L252-257: The statement of "air flow below the soil collar" does not make sense.

Equation 2: not completed; missed chamber pressure and chamber temperature.

L311: The statement of "Mound adjacent soil flux measurements showed no enhanced CH4 and CO2 fluxes in comparison to soils in the blank collar" does not consist with the results. For example, adjacent $CO_2$ flux (1.3) was almost three times of blank soil (0.47).

L337: 11 g is the maximum value; the variation range should be listed. Consequentially, the following value of 0.5-1.0 nmol m$^{-2}$ s$^{-1}$ was overestimated.

L415: Check the grammar.

A3: Shorten or discuss the scientific meaning of $\delta^{13}CO_2$ in this study.

Unify the concentration unit of ppm and µmol mol$^{-1}$.

---

## Author Comment (AC2) · 3 Feb 2021

**Response to Anonymous Referee 2**
**This manuscript presents a well thought out study to quantify methane emissions by termites in the Amazon rain forest. The authors reviewed the literature extensively and compared/discussed with their findings. I have some comments**

**that I think will make the study more valuable.**

—————————————————————————————————————————————————

Thank you for your kind words and the time you spent on reviewing our manuscript. We are grateful for your suggestions, which we have used to improve the manuscript. Below you will find a point to point response to each of your raised concerns and, if applicable, the corrected and improved manuscript text.

In addition, we would like to point out that the given termite emission estimates have changed due to an improved termite weight determination.

—————————————————————————————————————————————————

**1. Please provide the estimate of $CH_4$ emissions by termite and put in context with the overall $CH_4$ budget globally or in the Amazon. This manuscript presents $CH_4$ emission factors only. Without knowing how many termite mounds in Amazon, it's difficult to imagine the scale of the global $CH_4$ budget. I think this is one of a key messages for readers.**

Thank you for this interesting point. Below we will:

- elaborate on our considerations regarding the termite mound-upscaling;

- provide a 'back of the envelope' estimate on the role of termite emissions in the Amazon $CH_4$ budget, and in the global $CH_4$ budget;

- show the revised manuscript text.

**Termite mound upscaling:** Based on mound density numbers, it is difficult to state
a termite $CH_4$ emission estimate for the whole Amazon. As stated in the discussion (§4.3), mound density numbers vary largely between ecosystems. There is only little data available on mound density numbers, and most Amazon studies were performed in close proximity to our fieldsite (due to the research activities of local institute INPA). While this relatively large amount of local studies is unique and useful for the upscaling for our *local* ecosystem, it is unwise to assume that these mound density numbers apply to the whole Amazon. For this reason, we choose to only state a mound $CH_4$ emission estimate for our specific ecosystem, and to inform the readers about the limitations of this estimate.

**Back-of-the-envelope estimate for the global $CH_4$ budget:** By use of the data presented in the comprehensive modeling study of Kirsche et al. (2013), the following back-of-the-envelope estimate can be made:

Kirsche et al (2013) (Table 1) stated an annual global termite emission of 11 Tg $CH_4$ year$^{-1}$. They state that 36% of termite emissions originate from the region 'tropical South America' (p 818, first sentence), which calculates to 3.96 Tg $CH_4$ year. Substituting the used termite emission factor of 2.8 $\mu$g $CH_4$ $g_{termite}^{-1}$ h$^{-1}$ by the value found in our study of 5.6 $\mu$g $CH_4$ $g_{termite}^{-1}$ h$^{-1}$, would lead to a doubling of the regions estimated termite emission, namely 7.92 Tg instead of 3.96. The global estimate would increase from 11 Tg to 14.96 Tg.

The termite emission factor is a practical estimate of the average termite emission, which can be used for $CH_4$ budget studies. Since our study only measured one termite species, and there is likely a variation between species and ecosystems, we do not suggest that the currently used termite emission factor of 2.8 $\mu$g $CH_4$ $g_{termite}^{-1}$ h$^{-1}$ should be replaced by our value. We do however want to show and point out that the termite emission factor is still an uncertain part in the tropical $CH_4$ budget.

To include the reader in this train-of-thought, we have revised this part of the manuscript:

**Revised text in §4.3:** As a 'back-of-the-envelope' calculation, based on Kirsche et al. (2013): 36% of global termite emission (11 Tg) is expected to come from the region of 'tropical South America' (0.36*11=3.96 Tg). Substituting the emission factor of 2.8 with the newly found 5.6 $\mu$g CH$_4$ g$_{termite}^{-1}$ h$^{-1}$ would increase this regions estimate to 7.92 Tg, and the global estimate to 14.96 Tg.

Our study points out that termite emissions are still an uncertain source in the CH$_4$ budget, and are especially poorly quantified for the Amazon rain forest. Measurement of CH$_4$ emissions from different termite species, preferably covering species of different feeding or nesting habits, in combination with more precise termite distribution and abundance data, would allow more precise estimates and a better understanding of the role of termites in the CH$_4$ budget.

—————————————————————————————————————————————

**2. The first sentence in the Introduction section, it says "Methane (CH$_4$) is the second most important long-lived anthropogenic greenhouse gas." I think CH$_4$ has been recognized to be "short-lived" climate pollutant.**

Thank you for pointing this out. We have changed the first sentence to:

**Revised text:** Methane (CH$_4$) is the second most important anthropogenic greenhouse gas, but its natural sources are still not well understood.

—————————————————————————————————————————————

**3. In the Introduction section, Line 35, it says "Recently, it was shown that termites have a mitigating effect during droughts in tropical rain forests". Please elaborate what mitigating effect.**
Ashton et al. (2019) performed a termite suppression experiment and found that termite activity increased during drought, resulting in accelerated litter decomposition, elevated soil moisture, greater soil nutrient heterogeneity, and higher seedling survival rates. The authors suggested different underlying mechanisms for this response such as more favorable conditions for tunneling (e.g., drier, less-waterlogged ground), increased foraging ability above ground in the absence of heavy rain, and/or reduced predation pressure from ants.

We have changed the text in the Introduction to:

**Previous text:** Recently, it was shown that termites have a mitigating effect during droughts in tropical rain forests.

**Revised text:** Recently, it was shown that termites increase their activity during droughts, resulting, among others, in enhanced litter decomposition, elevated soil moisture and higher seedling survival rates, thereby demonstrating a mitigating effect during droughts in tropical rain forests.
––––––––––––––––––––––––––––––––––––––––––––––––––––––––––––––––––––––––––––––

**4. In the Introduction and in Appendix, the authors touched on N$_2$O emissions from termite but didn't give conclusive results.**

We agree that this point is not sufficiently discussed. An elaboration on this subject can be found at point 7.

––––––––––––––––––––––––––––––––––––––––––––––––––––––––––––––––––––––––––––––

[Figure]

**5. Section 2.3, Line 129, LGR GHG analyzer was mentioned to be the instrument deployed to quantify CH$_4$ emissions in flux chambers. I think authors should add brief instrument performance specifications and details of what calibration and drift evaluation have been done in Amazon. While the absolute CH$_4$ concentrations in flux chamber measurements are not very critical, since it's to measure the CH$_4$ concentration increase, but the manuscript does not provide the measured concentrations and jumped directly to the emission factor estimates. For example, LGR UGGA precision is about 2 ppb. Does it perform the same in Amazon? Also, what CH$_4$ concentration increments measured in the flux chambers? If it was only 2 ppb, then that data would not be useful. I think it should be many times more than the instrument precision and drift.**

Thank you for raising this point. During the campaigns, we have set the Los Gatos instrument to the 10-second averaging modus. Calibration gases were measured every second day for 5 minutes, resulting in a precision ($1\sigma$) of $\sim$0.7 ppm and $\sim$3.0 ppb for respectively CO$_2$ and CH$_4$.

The concentration increases during the 20 min chamber closure were large. Concentrations were climbing from forest concentrations to concentrations of up to 5750 ppb CH$_4$, and up to 1950 ppm CO$_2$, thereby far exceeding the measurement precision of the Los Gatos instrument.

We have added the following lines to the revised manuscript (beginning of Results):

**Revised text in §3.1:** Headspace concentrations increased strongly during chamber closure, and chamber concentrations reached up to 5750 nmol CH$_4$ mol and 1950 $\mu$mol CO$_2$ mol$^{-1}$.
––––––––––––––––––––––––––––––––––––––––––––––––––––––––––––––––––––––––––––––

**6. Well-designed flux chambers should have a small mixing fan or internal distribution tubing to quantify fluxes. §2.5 describes how LGR sampling tubes were connected on top of a 220 L chamber, if the air inside is not well mixed, the two fittings on top of the chamber may not detect CH$_4$ at the bottom of the chamber.**

Thank you for raising this point. Below we will:

- clarify the locations of the inlet fittings;

- elaborate on why we did not install a fan, and how we ensured mixed chamber air;

- give the revised manuscript text.

The 220 L chamber had two fittings on each side of the bucket while the smaller soil chamber had the two fittings on top of the chamber. Re-reading §2.5, we agree with the reviewer that the text is confusing, and we have revised this part.

As a small side note, termite mounds emit CH$_4$ from its entire surface, thereby presenting a sphere-shaped source of 45-65 cm height *inside* the chamber head space. Therefore, we do not expect a large difference between CH$_4$ concentrations at the top and the bottom of the chamber headspace.

We were hesitant about installing a small mixing fan. On the one hand, the absence of a mixing fan might lead to an underestimation of the flux (Christiansen et al. 2011). On the other hand, a mixing fan might lead to turbulence in the head space (Janssens

et al. (2000), Pumpanen (2004)), which possibly induces unrepresentatively high $CH_4$ emissions from the mound.

Since we wanted to avoid overestimation of termite mound $CH_4$ fluxes, we decided to not install a mixing fan. Instead we installed a 4 inlet vertical sampling tube inside the chamber head space, a technique to minimize the effects of gas concentration gradients in the head space (Clough et al, 2020). Inside the chamber at fitting height ($\sim$30 cm), a T-piece with two 20 cm-long Teflon tubing was positioned vertically, and two small incisions were made, so that head space air was sampled from 4 different heights (approx. at 10, 25, 35 and 50 cm height from the soil). The sampling tube was tested in the lab to verify whether air was sampled from all 4 inlets.

We have added the following lines and references to the revised manuscript:

**Revised text in §2.3:** Two one-touch fittings (1/4 inch, SMC Pneumatics) were installed on each side of the bucket. To minimize the possible effects of gas concentration gradients in the headspace, we installed a 4 inlet vertical sampling tube inside the chamber, so that air was sampled from different heights ($\sim$10, $\sim$25, $\sim$35 and $\sim$50 cm) in the headspace (Clough et al, 2020).

**Revised text in §2.5:** To be able to connect the Los Gatos instrument, the soil chamber had two one-touch fittings on top.

**References**
- Christiansen, Jesper Riis, et al. "Assessing the effects of chamber placement, manual sampling and headspace mixing on $CH_4$ fluxes in a laboratory experiment." Plant and soil 343.1-2 (2011): 171-185.
- Clough, Timothy J., et al. "Global Research Alliance $N_2O$ chamber methodology guidelines: Design considerations." Journal of Environmental Quality 49.5 (2020): 1081-1091.
- Janssens, Ivan A., et al. "Assessing forest soil $CO_2$ efflux: an in situ comparison of four techniques." Tree physiology 20.1 (2000): 23-32.
- Pumpanen, Jukka, et al. "Comparison of different chamber techniques for measuring soil $CO_2$ efflux." Agricultural and Forest Meteorology 123.3-4 (2004): 159-176
* * *
**7. Appendix A1 and A2 talk about N$_2$O calibrations and measured concentrations. The measured N$_2$O concentrations are outside of the calibration range. While the lower range (333.7 ppb) is similar to NOAA's measurements in Brazil, the manuscript does not provide the FTIR instrument precision and therefore, it's difficult to determine whether the detected range (333.7-342.4 ppb) is within instrument drift or it's actually an increment of N$_2$O. I don't think the authors can conclude there isn't N$_2$O emissions.**

Thank you for pointing this out. Below we will:

- explain why FTIR N$_2$O concentration measurements outside the calibration range can be used, by stating the precision and linearity of this instrument;

- explain why we can conclude that there are low N$_2$O emissions, by calculating the methods detection limit;

- support our statement (very low N$_2$O emissions) with additional data.

First of all, to clarify, the mentioned range of 333.7-342.4 ppb was measured over *all* chambers during the whole week. Actual increments during individual chamber closures were a lot smaller, as discussed here below. We have clarified this in the revised manuscript text.

The FTIR-instrument has the following reported precision (1$\sigma$) for 10 minute-averaged spectral analyses: 0.02 $\mu$mol mol$^{-1}$, 0.2 nmol mol$^{-1}$, 0.2 nmol mol$^{-1}$, 0.06 nmol mol$^{-1}$, and 0.04 permil, for respectively CO$_2$, CH$_4$, N$_2$O, CO, and $\sigma^{13}$C of CO$_2$.

Measurements performed during the campaign week were set to 5 minutes, so that a precision of 1/sqrt(N) is achieved, which is 0.09 nmol mol$^{-1}$ for $N_2O$.

The FTIR instrument has been shown to be linear for all gases in the ambient concentration range, and linearity was tested for $N_2O$ in the range 300-350 ppb. So while the choice of calibration gases was not optimal, we are confident that the FTIR-instrument still performs well in this concentration range.

**Detection limit of measurements:**
As also requested by reviewer 1, we calculated the minimum $N_2O$ flux detectable by this instrument and method:

- Assuming bag samples taken at 2, 5 and 8 minutes during chamber closure.

- Given: collar area 0.25 m$^2$, chamber volume 220 L, mound volume 50 L, headspace volume 220-50 = 170 L.

- Assuming: molar volume of 24.5 L mol$^{-1}$ (1 atm, 25 $^\circ$C).

- Minimum detectable concentration difference is ($2\sigma$) 0.18 nmol mol$^{-1}$.

- A concentration difference between t=2 min and t=5 min of 0.18 nmol mol$^{-1}$ is caused by a flux of 0.027 nmol collar/mound$^{-1}$ s$^{-1}$.

So, given the parameters above, the chamber set up has a detection limit of 0.027 nmol mound$^{-1}$ s$^{-1}$.

The FTIR-instrument has a cross sensitivity with $CO_2$, which is well determined for $CO_2$ <800 $\mu$mol mol$^{-1}$, but is less certain for unnaturally high $CO_2$ concentrations.

[Figure]

For this reason, we preferred to only use the $N_2O$ headspace concentration measurements with $CO_2$ <800 $\mu$mol mol$^{-1}$. Only 5 mound chamber closures had two consecutive $N_2O$ concentration points (t=2min and t=5min) with $CO_2$ <800 $\mu$mol mol$^{-1}$, and only 3 sets of two-consecutive concentration points passed the minimum concentration difference of 0.18 nmol mol$^{-1}$. These differences were $\sim$0.2, $\sim$0.3 and $\sim$0.7 nmol mol$^{-1}$, leading to a calculated $N_2O$ flux of $\sim$0.03 - $\sim$0.11 nmol mound$^{-1}$ s$^{-1}$.

**Additional measurements to support statement 'very low $N_2O$ emissions':** In October 2020, additional valley soil $N_2O$ flux measurement were performed with the same chamber system and collars (5 collars, 3 repetitions), but with a longer closing time (35 min), without termite mounds (so lower $CO_2$), and with 4 measurements per chamber closure. Also during these measurements, concentration increases were very low. Out of 15 measurements, 8 measurements had an $R^2$>0.90, and calculated fluxes ranged between 0.008-0.106 nmol m$^{-2}$ s$^{-1}$ (average=0.032 nmol m$^{-2}$ s$^{-1}$, sd=0.33). Since the valleys are known to be low on nitrogen (Quesada et al., 2010), such low fluxes are expected, and similar $N_2O$ valley soil fluxes were found by Matson et al (1987) in a fieldsite closeby.

Since the 3 calculated mound $N_2O$ flux measurements are based on only 2 consecutive headspace concentration points, no uncertainty can be given, wherefore we preferred not to state the fluxes in the previous manuscript. For the revised manuscript, we have stated the detection limit, explain why not all mound fluxes could be calculated, and support our observation of low $N_2O$ mound fluxes by the additional soil $N_2O$ flux measurements:

**Appendix A2:** Gas samples (3 samples per chamber closure) revealed stable $N_2O$ concentrations, and headspace concentrations ranged between 333.7 and 342.4 nmol mol$^{-1}$ over the different chamber closures. Since headspace $CO_2$ concentrations sometimes exceeded 800 $\mu$mol mol$^{-1}$, and $N_2O$-$CO_2$

cross-sensitivity becomes uncertain at higher $CO_2$ concentrations, not all 3 headspace samples per chamber closure could be used, wherefore qualitative $N_2O$ flux estimates cannot be reported. As a back-of-the-envelope calculation, $N_2O$ fluxes were calculated if 2 consecutive headspace samples were with $CO_2$ <800 $\mu$mol mol$^{-1}$, and if a minimum $N_2O$ concentration difference of 0.18 nmol mol$^{-1}$ was found (FTIR precision ($\sigma$) for 5 min spectra is 0.09 nmol mol$^{-1}$), which gave us 3 mound flux estimates ranging between ~0.03 and ~0.11 nmol $N_2O$ mound$^{-1}$ s$^{-1}$. Similarly low fluxes were found during additionally performed flux measurements, performed as part of a substudy, which showed valley soil fluxes ranging between 0.008-0.106 nmol $N_2O$ m$^{-2}$ s$^{-1}$. The low mound fluxes would be in agreement with a previous study which suggested that termite mound $N_2O$ emissions are dependent on the N-content of the termites diet (Brauman et al., 2015), which is expected to be low in the valleys of this ecosystem (Quesada et al., 2010).

**References:**

-Matson, Pamela A., and Peter M. Vitousek. "Cross‐system comparisons of soil nitrogen transformations and nitrous oxide flux in tropical forest ecosystems." Global Biogeochemical Cycles 1.2 (1987): 163-170.

---

## Author Comment (AC3) · 3 Feb 2021

**Response to Anonymous Referee 3**
This study presented a global interesting issue of termite CH$_4$/CO$_2$ emission in an Amazonian tropical rainforest. As a case study, this in-situ measurement of termite mound emissions provided information about termite CH$_4$/CO$_2$ production under natural conditions, it will contribution some knowledge to Biogeosciences. However,

the field experiment was not well designed, and the limited data was not well analyzed. I would like to encourage the authors to revise the manuscript following my comments.

––––––––––––––––––––––––––––––––––––––––––––––––––––––––––––––––––––––––––––

**Thank you for your time spent on reviewing our submission!** We are grateful for your suggestions, which we have used to improve the manuscript. Below you will find a point to point response to each of your raised concerns and, if applicable, the corrected and improved manuscript text.

In addition we would like to point out that:

- we have uploaded a revised text of §4.1 (First paragraph of discussion), which is shown at the end of this review;

- we have uploaded a revised Figure 2, which is shown at the end of this review (Previous Figure 4 is Figure 2 in revised manusript);

- we have uploaded a revised Figure 4, which is shown at the end of this review (Previous Figure 5 is Figure 4 in revised manusript);

- we have uploaded 4 additional figures, belonging to point 7 and 8 of this review, which are shown at the end of this review;

- the given values in the text might have changed due to an improved termite weight determination.

**General comments**

––––––––––––––––––––––––––––––––––––––––––––––––––––––––––––––––––––––––––––

**1. "The blank measurements (collar with only soil and litter) showed an average**

$CH_4$ emission of 1.15 nmol collar$^{-1}$ s$^{-1}$" (L175) means the forest soil was a VERY LARGE $CH_4$ SOURCE (4.6 nmol m$^{-2}$ s$^{-1}$ or 23.2 $CH_4$ ha$^{-1}$ y$^{-1}$). It was a FUNDAMENTAL PROBLEM! Actually, the "blank" soil should be $CH_4$ sink. Even "1.15 nmol collar$^{-1}$ s$^{-1}$" was "-1.15 nmol collar$^{-1}$ s$^{-1}$", the soil $CH_4$ sink of "-23.2 kg $CH_4$ ha$^{-1}$ y$^{-1}$" was an unbelievable large value.

Though the reviewer correctly points out that most tropical forest soils are methane sinks, soil methane emissions in tropical ecosystems are common, especially when anaerobic conditions occur. Therefore, we disagree that this points to a fundamental problem.

In the revised manuscript we will substitute our blank collar measurement by a set of additional measurements from the surrounding area. These measurements show that the methane fluxes from the valley soil are spatially heterogeneous, but in general low. It is important to note that this heterogeneity has no impact on the given $CH_4$ emission estimates from the termite mounds, since the emissions measured from the mounds are on average a factor 627 higher than the average background soil $CH_4$ emission.

Below we will:

- provide additional information (measurements and literature) which show that soil valley $CH_4$ fluxes are heterogeneous but of low magnitude in comparison to the measured mound fluxes;

- compare the soil and mound fluxes by providing an improved Figure 4;

- provide text for the revised manuscript.

**Additional soil valley flux measurements**

Most tropical forest soils are methane sinks (Dutaur and Verchot, 2007, Kiese et al. 2003). Nevertheless, soil methane emission in tropical ecosystems can still be observed (Carmo et al. 2006), especially when anaerobic conditions occur, such as which can be found in the valley (Sihi et al. 2020, Moura et al. 2012). This is also observed by our set of additional measurements:

*Additional measurements:* *valley soil chamber flux measurements (small chamber set up as described in §2.5), 10 soil collars, 3 repetitions, $\sim$500 m from manuscript termite mounds, 5-50 m from igarapé (stream), measured in same week as termite mounds (March 2020), soil $CH_4$ fluxes ranged between -0.12 to 2.89 nmol m$^{-2}$ s$^{-1}$, (median=-0.02, average=0.15, sd=0.55).*

Our additional measurements show that valley soil fluxes are heterogeneous, and in general negative (median=-0.02), but that locations with relative high emissions (hotspots) can be found. Our mound adjacent soil fluxes were in general higher (0.3-8.9 nmol $CH_4$ m$^{-2}$ s$^{-1}$, 16 soil collars), showing that mound adjacent soils are deviating from the average valley soil, likely due to the nearby presence of an active termite mound.

The magnitude of the *original* blank collar fluxes (3.9-5.4 nmol $CH_4$ m$^{-2}$ s$^{-1}$) is quite similar to the magnitude of mound adjacent fluxes (0.3-8.9 nmol $CH_4$ m$^{-2}$ s$^{-1}$). While the blank collar was not directly located next to a mound ($\sim$5 m of mound nr. 15), the comparison with the different datasets points at the presence of a local $CH_4$ hotspot (Subke et al. 2018), thereby not being representative as a control collar. For the revised manuscript we will use the 10 additional soil collar measurements as our 'blank collar' reference point.
The aim of the blank 'control' measurement was to show the large difference between a 'normal' valley $CH_4$ emission (per area), and an emission (per area) when a termite mound is present. Considering the average mound emission of 25.2 nmol mound$^{-1}$ s$^{-1}$, and the average valley soil emission of 0.03 nmol collar$^{-1}$ s$^{-1}$ (0.15 nmol m$^{-2}$ s$^{-1}$), an average collar area emits a factor 630 more $CH_4$ when a termite mound is present. Including these complementary measurements will strengthen our message that termite mounds are hotspots in comparison to their surroundings.

We have included these additional measurements for comparison, by adapting Figure 4 (now renumbered as Figure 2, see end of this review), and by including these measurements at the following places in the manuscript:

––––––––––––––––––––––––––––––––––––––––––––––––––––––––––––––––––––––––––––––––

**In Methods, §2.5: Valley and mound adjacent soil fluxes**
Every mound adjacent soil flux measurement was 4 minutes, and the set of 4 collar measurements was performed once per mound, with exception of mound nr. 19. For mound nr. 13 and nr. 14, the measurements were performed on the $2^{nd}$ measurement day, for mound nr. 15 and nr. 16, the measurements were done on the $3^{rd}$ measurement day. Mound adjacent soil fluxes will be expressed per collar area (0.25 m$^2$), to be better comparable to mound emissions. The same chamber set up was used in a sub study at a close by transect ($\sim$ 500 m from termite mounds) where, among others, valley soil fluxes were measured (10 collars, 3 repetitions). Measured soil fluxes from the valley will be shown for comparison.

**In Results, §3.1: Mound $CH_4$ and $CO_2$ emissions**
Headspace concentrations increased strongly during chamber closure, and chamber concentrations reached up to 5750 nmol $CH_4$ mol$^{-1}$ and 1950 $\mu$mol $CO_2$ mol$^{-1}$. Mound $CH_4$ emissions ranged between 17.0 and 34.8 nmol mound$^{-1}$ s$^{-1}$ (Fig. 1), with an average emission of 25.2 nmol mound$^{-1}$ s$^{-1}$. Additional valley measurements showed heterogeneous soil $CH_4$ fluxes with small uptake and emission taking place alongside, ranging between -0.1 and 2.9 nmol m$^{-2}$ s$^{-1}$ (med=-0.02, avg=0.15, sd=0.54). Mound adjacent $CH_4$ soil fluxes, measured at 20 and 45 cm from the mound, ranged between 0.4 and

8.9 nmol $CH_4$ m$^{-2}$ s$^{-1}$ (avg=2.14, sd=2.00), and were on average enhanced in comparison to valley soils (Fig. 2). Soil valley $CO_2$ fluxes were found to range between 0.9 and 3.7 $\mu$mol m$^{-2}$ s$^{-1}$ (avg=2.14, sd=0.74) (Fig. 2). Mound adjacent soil $CO_2$ fluxes showed an average emission of 4.84 $\mu$mol $CO_2$ m$^{-2}$ s$^{-1}$ (range=2.0-10.1, sd=2.01), thereby being enhanced with respect to the surrounding soils (Fig. 2). Mound $CO_2$ emissions, corrected for the average valley soil respiration, were ranging between 1.1 and 13.0 $\mu$mol mound$^{-1}$ s$^{-1}$, with an average emission of 8.14 $\mu$mol mound$^{-1}$ s$^{-1}$ (Fig 1).

**In Discussion, §4.3**:
Valley soil $CH_4$ and $CO_2$ fluxes were similar to what was found by earlier studies (Souza (2005), Moura (2012), Chambers et al. (2004), Zanchi et al. (2012). On average, mound adjacent soil $CH_4$ and $CO_2$ fluxes were enhanced with respect to valley soils, although differences were small, and no clear emission pattern with 'distance to mound' was observed. While mound adjacent soil fluxes are possibly enhanced, we preferred to avoid overestimation, and decided to treat termite mounds as very local hot spots, with measured fluxes only representative for the collar area of 0.25 m$^2$. On average, $CH_4$ and $CO_2$ fluxes per collar area were found to be a factor ~630 and ~16 higher when an active termite mound was present.

From the reviewers comment, we realize that confusion might arise about the amount of mounds sampled. Below we will:

- clarify that we measured fluxes of 5, and not 19, termite mounds;

- clarify how many mound *sub*samples have been measured;

- report additional subsample measurements which confirm the termite emission factor, and present a new Figure 5, which will show these additional measurements;

- provide the improved manuscript text for §2.6, and for other parts of the manuscript.

Our mound selection procedure for the 5 mounds was as follows:

- Firstly, we searched for mounds, which were suitable for flux chamber measurements (sufficient space for collar installation, not attached to tree). We found 20 suitable and active mounds, and we sampled each mound and determined the species at the *Laboratory of Systematics and Ecology of Soil Invertebrates* at INPA. Table 1 in the manuscript gives an overview of the found species per mound.

- When further selecting individual mounds of these 20 mounds, we only choose mounds of the same species, so that effects of interspecies variation could be excluded.

- For practical reasons, we choose a set of mounds which were closely located to each other.

- With these criteria in mind, we selected the mounds from which fluxes would be measured, which were mounds nr. 13, nr. 14, nr. 15, nr. 16 and nr. 19.

The choice of limiting our flux measurements to 5 mounds was based on practical considerations (hours of daylight, days in the field, distance to cover), which were especially time constrained due to our additional bag sampling measurements (Appendix A). For a possible follow up study, we would leave this element out.

**Moreover, the authors should explain why the sub sample experiment was only conducted for one mound (L161: "only one sub sample was found suitable from the all 19 mounds").**

The sentence copied by the reviewer is different than the sentence stated at line 161, which was:

*'From the sample from mound 19, only one suitable sub sample was found'*

To clarify: for each of the 5 selected mounds, we sampled one solid (not crumbling) piece, of which we took 3 subsamples, of which we measured emissions and counted termites. In principle, this would lead to 15 subsamples. Nevertheless, due to practical problems at mound 19, we only managed to separate 1 suitable subsample, wherefore the total amount of subsamples was 13, as shown in the original Figure 5 of the manuscript.

In the last few months, we have performed additional measurements:

- Additional measurement 1 (AM1): performed in October 2020 (dry season), with 15 subsamples of the same mounds (mounds nr. 13, nr. 14, nr. 15, nr. 16 and nr. 19). A termite emission factor of 0.0002976 (se=$1.32*10^{-5}$) $CH_4$ per termite per second was found.

- Additional measurements 2 (AM2): performed in December 2020 (transition dry/wet season), with 5 subsamples, taken from a new mound of the same species. A termite emission factor of 0.0003043 (se=1.41*$10^{-5}$) $CH_4$ per termite per second was found.

For the revised manuscript, we have added these $CH_4$ termite emission measurements to the text and to Figure 5, to show the reader the consistency of the termite emission factor between mounds and seasons. Nevertheless, since we prefer to combine only measurements obtained during the same field campaign week, the manuscript estimates and derivations are based on the original determined termite emission factor of 0.0002985 nmol termite$^{-1}$ s$^{-1}$.

We have improved Figure 5 (in revised manuscript, renumbered as Fig. 4), which we uploaded, and which can be found at the end of this review. In the text, we have made the following changes:

**Revised text caption Table 1:** Termite mounds: location, dimensions, and observed species. Termite mound volumes were estimated by Eq. (1), and mound surfaces were estimated by mathematically considering the lower part of the mound as a column, and the upper part as half a sphere. In mound nr. 1, two different termite species were found. The five mounds indicated in bold (mound nr. 13, nr. 14, nr. 15, nr. 16 and nr. 19) were the mounds selected for flux measurements.

**Revised text in §2.6:** At mound nr. 13, nr. 14, nr. 15, nr. 16 and nr. 19, after the last mound flux measurement, a mound sample was taken of approximately 1 L volume. From this, three small sub samples were taken (volume not determined).

**Revised text in §2.6:** To verify whether the termite emission factor was stable between seasons and mounds, additional measurements were performed. In October 2020 (dry season), the same type of measurements were performed on 15 subsamples of the same termite mounds, and in December 2020 (transition dry-wet season), 5 subsamples of a different mound of the same species were analysed.

**Revised text in §3.2:** $CH_4$ and $CO_2$ emissions of 13 mound sub samples were measured. For each sub sample, the measured gas production was plotted over the counted termites (Fig. 4). The fitted line has a forced intercept at y=0. For $CH_4$, an emission of 0.0002985 nmol termite$^{-1}$ s$^{-1}$ was found (se=1.77*10$^{-5}$), fitted with an $R^2$ of 0.95 (n=13). The set of additional measurements resulted in similar termite emission factors namely 0.0002976 nmol termite$^{-1}$ s$^{-1}$ (se=1.32*10$^{-5}$) and 0.0003043 nmol termite$^{-1}$ s$^{-1}$ (se=1.41*10$^{-5}$), for respectively the measurements of October and December 2020. Given estimates in this paper are based on the originally determined termite emission factor of 0.0002985 nmol termite$^{-1}$ s$^{-1}$. For $CO_2$, an emission of 0.1316 nmol termite$^{-1}$ s$^{-1}$ was found (se=2.59*10$^{-2}$), with an $R^2$ of 0.68 (n=13). Excluding the out liers (32, 14.9 nmol s$^{-1}$ and 313, 80.9 nmol s$^{-1}$) gives an $R^2$ of 0.88 (n=11), with a $CO_2$ emission of 0.074 nmol termite$^{-1}$ s$^{-1}$ (se=8.5*10$^{-3}$).

**Revised text in §4.3:** Furthermore, exploratory dry season measurements of the same mounds showed emissions of the same magnitude (not shown), and additional dry season mound subsample measurements revealed very consistent termite $CH_4$ emission factors (Fig. 4). We therefore do not expect that mound $CH_4$ emissions are only of importance in the valleys, or only present in the wet season.
* * *
**4. In tropical forest, the termite mounds have different size and different shapes, and many are already not active mounds. This study only selected the relatively small size of termite mound (Table 1), thus it is not surprised that the authors gave the conclusion of weak correlation between $CH_4$ emission and mound size (3.1; Fig. 3).**

For this study, we only measured active termite mounds; but during our search in the first phase of the research, no abandoned epigeal mounds were found, and only 1 abandoned tree nest was found.

Furthermore, we also point out to the readers that termite mounds appear in many

different sizes and shapes (§4.1). Because we are aware that different species build different type of nests, we only searched for a *species-specific* correlation between mound size and mound emission.

It is common that a certain species-specific correlation is found between mound size and mound population (Lepage and Darlington, 2000, Pequeno et al. 2013), wherefore it is also reasonable to expect a relationship between mound size and mound emission. Nevertheless, as Pequeno et al. (2013) pointed out, mounds from the species *N. Brasiliensis* have been shown to **not** present a strong correlation between mound size and mound population. Therefore, it is not surprising that we also did not find a strong relationship between mound size and mound emission.

To shorten the manuscript, we have decided to remove the original Figure 3, and only report our findings in the text. The discussion on variation in termite mounds and shapes, and on correlation between emisssion and mound size, can be found in the Discussion in §4.1 and §4.2:

**Revised text in §4.1**: There is a large variety in type of termite mounds (shape and size are dependent on species, ecosystem, climate (Noirot and Darlington, 2000)), explaining the wide range of reported termite mound $CH_4$ emissions (Table 2, middle and lower part).

**Revised text in §4.2**: Interestingly, Pequeno et al. (2013) concluded that mound volume is a weak indicator for population size for nests of the species *N. brasiliensis*, as also indicated by the weak correlation we found between mound volume and mound $CH_4$ emissions .

––––––––––––––––––––––––––––––––––––––––––––––––––––––––––––––––––––––––––––––––

**5. This in-situ measurement could not be able to partition the contribution of**

**mound soil ($CO_2$ source but $CH_4$ sink) from termite, thus the termite $CH_4$ emission could be underestimated but termite $CO_2$ emission could be overestimated. The results should be calibrated, because the structure and nutrients of the mound-soil are different from the normal soil (blank soil in this study).**

Based on in-situ mound measurements as conducted here, it is impossible to partition the contribution of mound material vs termites. This was not done in any comparable studies. As the reviewer correctly points out, for $CH_4$ this will lead to an underestimation, and for $CO_2$ to an overestimation of estimated termite emissions. However, studies like ours determine the overall termite-induced emissions and this is the aim of our study.

Below we will:

- evaluate the impact of soil and mound emissions/uptake on our $CH_4$ and $CO_2$ termite estimates;

- elaborate on direct and indirect termite $CO_2$ emissions (termite-induced $CO_2$ emissions);

- show how we improved this part in the manuscript.

**The impact of mound emissions/uptake on our $CH_4$ and $CO_2$ termite estimates**

**For mound $CH_4$ emission**: overestimation is not expected: surrounding valley soils show heterogeneous but in general low magnitude (negative) fluxes, ranging between -0.03 to 0.72 nmol collar$^{-1}$ s$^{-1}$ (median=-0.01, average=0.03, sd=0.55, collar= 0.25 m$^{-2}$). Considering the average mound emission (25.2 nmol collar$^{-1}$ s$^{-1}$),

the contribution of an average soil $CH_4$ flux to the mound emission would lead to an overestimation of $< 1\%$.

As the reviewer correctly points out, underestimation is more likely, due to the uptake of $CH_4$ by mound material, as also discussed in the manuscript. To give a lower bound assessment, we have used the *net* mound $CH_4$ emissions for our ecosystem estimates.

**For mound $CO_2$ emission**: we cannot be sure which part of the mound emitted $CO_2$ derives directly from termites and which part derives from soil and mound respiration. To account for soil respiration, the most attainable approach is to determine the average $CO_2$ emission of the surrounding soils and subtract this value from the measured mound $CO_2$ emissions. Values shown in the manuscript are the corrected values.

Mound respiration however is an indirect effect of termite activity, and thereby a termite-*induced* emission. Partitioning direct and indirect termite $CO_2$ emissions is difficult, and impossible to determine without disturbing the mound. We will therefore clearly state this in the manuscript, and discuss that *direct* termite-emitted $CO_2$ emissions are presumably lower.

The topic of soil and mound respiration is discussed in the following places of the revised manuscript:

**Revised text in §2.3, last sentence**: Unless mentioned otherwise, given mound $CO_2$ emissions are corrected for the estimated contribution of soil respiration, by subtracting the average valley soil $CO_2$ emission (see §2.5).

**Revised text in §3.1:** Soil valley $CO_2$ fluxes were found to range between 0.9 and 3.7 $\mu$mol m$^{-2}$ s$^{-1}$ (avg=2.14, sd=0.74) (Fig. 2). Mound adjacent soil $CO_2$ fluxes showed an average emission of 4.84 $\mu$mol $CO_2$ m$^{-2}$ s$^{-1}$ (range= 2.0 - 10.1, sd=2.01), thereby being enhanced with respect to the surrounding soils (Fig. 2). Mound $CO_2$ emissions, corrected for the average valley soil respiration, were ranging between 1.1 and 13.0 $\mu$mol mound$^{-1}$ s$^{-1}$, with an average emission of 8.14 $\mu$mol mound$^{-1}$ s$^{-1}$.

**Revised text in §4.1:** Mound $CO_2$ emissions and the termite $CO_2$ emission factor were similar, or a little higher, in comparison to the few values found in literature. Nevertheless, since mound material and termites were measured together, the contribution of *indirect* termite emissions, i.e. mound respiration, cannot be quantified, so that the *direct* termite-produced $CO_2$ emission is presumably lower.

**Revised text in §4.3:** Nevertheless, since the 'emission per mound' as well as the 'termite emission factor' are both affected by indirect effects of termite activity (mound respiration), the contribution of *direct* termite-emitted $CO_2$ into the ecosystem is presumably smaller.

––––––––––––––––––––––––––––––––––––––––––––––––––––––––––––––––––––––––––––

**6. Chamber volume (CV in L145; L159-163, L258-262) is a major parameter for calculation of flux rate (Equation 2). If the exact volume of the sample mound was not known, means CV was not known, based on the calculation using equation 2, the estimated both CH$_4$ and CO$_2$ fluxes (Table 2, 3; L218-222, L241-243) would be absolutely under- or over-estimated.**

In all our assumptions, we have followed literature (Clough et al. (2019), Kirschke et al. (2013), Krishna and Araujo (1968), Pequeno et al. (2013), Ribeiro (1997), Sanderson (1996)), and have tried to aim for a lower bound appraisal. For example, for mound volume estimation, we have chosen to use the equation given by Pequeno (2013). Furthermore, we considered the mound as a solid body, even if a previous comparable study did not (Martius et al. 1993), thereby possible underestimating our mound emissions by $\sim 30\%$ (see text in §4.1, copied below).

So even if CV is an uncertain parameter, by communicating this clearly to the reader, and by demonstrating that our estimate is lower bound, our message, that termite mounds and termites are important in this ecosystem, remains strong.

**Revised text in §4.1:** An additional possible underestimation is caused by the estimated corrected chamber volume, as used in Eq. (2). In this study, we considered the mound volume as a solid body. A previous study considered the solid nest volume as 10% of the actual mound volume (Martius et al., 1993), leading to a larger corrected chamber volume, and therefore to larger calculated mound emissions. By use of this approach, average measured emissions would increase by almost 30% to be 32.7 nmol $CH_4$ mound$^{-1}$ s$^{-1}$ instead of 25.2 nmol $CH_4$ mound$^{-1}$ s$^{-1}$.
* * *
**7. In my experience, this R$^2$ > 0.95 (L178 and other places) was non-believable. The chamber was relatively (or very) large (220 L), UGGA internal (pump) flow was only about 350 mL min$^{-1}$, the chamber air could not be mixed without installing one or two micro fans inside the chamber, because it takes about 630 min to replace the chamber air if only depending on the UGGA internal pump. Particularly, the chamber was about 1 m high, the emitted $CH_4$ and $CO_2$ was not be able to be mixed inside the chamber if only depending on both diffusion and UGGA internal pump. Moreover, based on the bag sampling (A1), $CH_4$ flux could be estimated. The authors are suggested to compare the result with that of mound chamber and sub sample.**

Thank you for raising this topic, which we will answer point by point (7.1, 7.2, 7.3):

7.1: Mixing in the chamber, where we explain why we did not install a fan, and how we

ensured mixed chamber air;

7.2: Linearity of headspace concentration increase, where we show that, despite small fluctuations, linear regression ($dCO_2/dt$, $dCH_4/dt$ and $dCH_4/dCO_2$) was performed with an $R^2 > 0.95$;

7.3: FTIR bag measurements, where we elaborate on estimation of mound $CH_4$ fluxes based on bag measurements.

**7.1) Concerning the mixing of the chamber,** first a small side note: termite mounds emit $CH_4$ from its entire surface, thereby presenting a sphere-shaped source of 45-65 cm height *inside* the chamber head space. Therefore, we do not expect a large difference between $CH_4$ concentrations at the top and the bottom of the chamber head space.

We were hesitant about installing a small mixing fan. On the one hand, the absence of a mixing fan might lead to an underestimation of the flux (Christiansen et al. 2011). On the other hand, a mixing fan might lead to turbulence in the head space (Janssens et al. (2000), Pumpanen (2004)), which might induce unrepresentatively high $CH_4$ emissions from the mound.

Since we wanted to avoid overestimation of termite mound $CH_4$ fluxes, we decided to not install a mixing fan. Instead we installed a 4 inlet vertical sampling tube inside the chamber, a technique to minimize the effects of gas concentration gradients in the head space (Clough et al, 2020). Inside the chamber at fitting height ($\sim$30 cm), a T-piece with two 20 cm-long Teflon tubing was positioned vertically, and two small incisions were made, so that head space air was sampled from 4 different heights (approx. at 10, 25, 35 and 50 cm height from the soil). The sampling tube was tested in the lab to verify whether air was sampled from all 4 inlets.

We have added the following lines to the manuscript, and have added the following references to the manuscript.

**Revised text in §2.3:** Two one-touch fittings (1/4 inch, SMC Pneumatics) were installed on each side of the bucket. On the inside of the bucket, a 4 inlet vertical sampling tube was placed, so that air was sampled from different heights ( 10, 25, 35 and 50 cm) in the headspace (Clough et al, 2020).

* * *
**7.2) Concerning the linear regressions of $dCH_4/dt$ with $R^2 > 0.95$,** first of all, we would like to rectify two details from the manuscript. At line 132, we state that chambers were closed for 25 minutes, but this should have been 20 minutes. In addition, we state that we are correcting for sampling bag dilution (line 147), a decision we later reversed because gradients were only calculated over headspace concentrations after bag filling: this sentence should have been deleted.

In the figure (Review-Figure 7.2a, end of this review) we show the *last* 10 minutes (of total chamber closure) of five headspace chamber increases, measured on one day. As can be seen, even while fluctuations occur, the linear regression line still captures the shape of the line well, and still an $R^2 > 0.95$ can be found.

To further clarify the Review-Figure 7.2a: chamber closures were for 20 minutes, and sample bag filling (Appendix A of manuscript) was done at minute 3, 5 and 8. To determine the actual headspace concentration increase, we used the increment *after* the first 10 minutes, when the chamber was less disturbed by the bag sampling. The fluctuations, clearly visible for mound nr. 14 and nr. 15, take place at the 'beginning' of this second time window. Part of this might be explained by the remaining effect of the bag sampling, but we also expect that our presence close to the flux chamber (when closing and labeling the sampling bags) might have had an effect: in a different experiment, we saw headspace fluctuations, which disappeared when we distanced ourselves from the chamber. This is something we should keep in mind for a possible next experiment.

We realize that this part of the *Material and Methods* needs to be improved, and we have revised the text in §2.4 to:

**Revised text in §2.4:** Linear regression was used to derive the concentration increase, and given error bars are the propagated standard error of the linear regression slope. Concentration increases were calculated over the last 10 minutes of the chamber closure, to avoid possible effects of the bag filling. Nevertheless, if clear headspace concentration fluctuations were observed in the beginning of this time window, possibly by a remaining effect of the bag filling, the window was shortened by a maximum of 2 minutes (leaving a time window of 8 minutes). All calculated dC/dt increases showed a $R^2 > 0.95$.

—————————————————————————————————————————————————————

**Concerning the linear regressions of $dCH_4/dCO_2$ with $R^2 > 0.95$,** at line 178 we stated:

*The $CH_4$ and $CO_2$ concentration increases inside the closed flux chamber were*

*strongly correlated ($R^2 > 0.95$ for each chamber closure).*

This statement is true: during all chamber closures, fluctuations in $CH_4$ and $CO_2$ concentrations were strongly correlated, with $R^2 > 0.95$.

As also discussed in §4.1, both gases are showing a strong correlation, AND showing fluctuations of the same magnitude and at the same moment. We therefore assume that these fluctuations are caused by an external factor, like wind or human distur-bance, sucking/pushing out high-concentration air from the chamber. This can also be seen in the figure below (Review-Figure 7.2b), where some fluctuations seem to happen when bag filling is performed. Nevertheless, it can also be seen that the gradient recovers after each fluctuation. In addition, if chamber air is diluted, the gradient will be underestimated, thereby not weakening the message of our paper.
––––––––––––––––––––––––––––––––––––––––––––––––––––––––––––––––––––––––––

**7.3) Concerning the FTIR bag measurements,** bag samples were aimed to be sampled at 2, 5 and 8 minutes after chamber closure ($\Delta t=3$ min). Nevertheless, during the field campaign, variation in $\Delta t$ occurred, such as due to changing pump performance (due to varying battery voltage), or due to timing inconsistencies. Since $\Delta t$ between bag samples is not known with certainty, a flux based on the bag samples *alone* cannot be given. As described in the manuscript, we have used the Los Gatos fluxes to deduct the FTIR fluxes.

**Revised text in A2:** To calculate the fluxes of $N_2O$ and CO, FTIR-measured bag concentrations of $N_2O$, CO and $CO_2$ were used. For each chamber closure, the $dN_2O/dt$, $dCO/dt$ and $dCO_2/dt$ were calculated so that the ratios $dN_2O/dCO_2$ and $dCO/dCO_2$ could be derived. To calculate the fluxes of $N_2O$ and CO, the ratios were combined with the in-situ measured mound $CO_2$ flux, as measured by the Los Gatos instrument. This approach was chosen because the intended 3 min bag sampling interval was not always

accomplished, so that a fixed Δt could not be assumed with certainty.

For the reviewers interest, here below (Review-Figure 7.3) we show one example of bag concentrations, measured by the FTIR, in comparison to Los Gatos concentrations. During this measurement, sampling with 3 minutes interval was close to accomplished, so that Δt approximated 3 min.
* * *
**8.  Data was too limited; I strongly encourage the authors to show the data measured in the dry season (L348-350) and compare it with that of wet season showed in this manuscript.**

The measurements in the dry season were performed as an *exploratory* measurement, to see whether the mounds were still active, and fluxes were similar as in the wet season.  Nevertheless, due to time limitations, measurements were only performed once. For this reason, we do not show them in the manuscript.

For the reviewers interest, we can show the additional measurements from October 2020 here in the review (Review-Figure 8, dark red bars). Measurements from mound nr. 13, nr. 15 and nr. 16 were in the same range as measured in March 2020, while fluxes from mound nr.  14 and nr.  19 were deviating.  Considering the long time period which passed (∼6 months), the change could be due to increased/decreased population size and/or activity, or (in case of mound nr.  14) a collar which was not well installed. Since it was outside the scope of the presented research, we have not structurally looked into the reasons for the difference, and prefer not to speculate too much. Nevertheless, these measurements confirm that the mounds are also active in the dry season, and remain hotspots in the ecosystem.

Additional 'dry season' measurements of mound *sub* samples, used to determine the termite emission factor, were performed in two sets. For the revised manuscript, the new figure and revised text can be found at point 3 of this review.
* * *
**9. Overall, using the limited data to scale up it to ecosystem (4.3) and global (4.4) levels would no doubt create large uncertainty. The authors are suggested to cancel or at least shorten these two issues.**

**For our upscaling to ecosystem level (§4.3):** while this estimate is based on limited data, it is important to note that up scaling was only done for our *local* ecosystems $CH_4$ budget.

In addition, our fieldsite is situated in a geographical unique region: due to the nearby-presence of the institute INPA (which has been doing Amazon research since the 50's), many termite and ecosystem studies have been performed closeby (see bulletpoints below). Therefore, assumptions (mound density numbers, termite abundance) and comparisons (available ecosystem $CO_2$ and $CH_4$ fluxes) can be stated with more certainty than anywhere else in the Amazon. So, because of this strong complementary local dataset, we can estimate and evaluate the role of termites for our *local* $CH_4$ budget

**Local studies:**
-5 local studies ($<$ 50 km) reported mound density numbers (Queiroz, (2004), Oliveira et al., (2016), Dambros et al., (2016), (de Souza and Brown, (1994), Ackerman et al., (2007);
-1 local study ($<$ 50 km) studied the weight and mound-population dynamics of the same termite species (Pequeno 2013);
- Several studies focussing on ecosystem $CO_2$ and $CH_4$ were performed at the exact same fieldsite (Chambers et al.,

(2004), Moura. (2012), de Souza (2005), Zanchi et al. (2014), Querino et al. (2011).

**For our upscaling to global levels,** we have followed the method and assumptions as described by Kirschke et al. (2013). To clarify, we only have substituted the 'termite emission factor' value, all the other upscaling has been adapted from Kirscke et al. (2013). In addition, it is important to make the link to the global levels, which informs the reader about the important role of model parameters (termite density and termite emission factors), thereby clearly showing that this is an uncertain part of the $CH_4$ budget.

As suggested by Reviewer 2, the text on the global estimate has been extended and improved:

*Termites contribution to tropical South America CH$_4$ budget (in §4.3)*
In current $CH_4$ budget studies, a termite emission factor of 2.8 $\mu$g $CH_4$ $g_{termite}^{-1}$ $h^{-1}$ is used for '*Tropical ecosystems and Mediterranean shrub lands* (Kirschke et al., 2013; Saunois et al., 2020), which is mainly based on field studies in Africa and Australia (Brümmer et al., 2009a; Jamali et al., 2011a, b; Macdonald et al., 1998; MacDonald et al., 1999). The only termite emission factor measured for the Amazon rain forest is by Martius et al. (1993) (3.0 $\mu$g $g_{termite}^{-1}$ $h^{-1}$) for a wood-feeding termite species, which are expected to emit less $CH_4$ than soil-feeding termites (Bignell and Eggleton, 2000; Brauman et al., 1992). As a 'back-of-the-envelope' calculation, based on (Kirschke et al., 2013): 36% of global termite emission (11 Tg) is expected to come from the region of 'tropical South America' (0.36*11=3.96 Tg). Substituting the emission factor of 2.8 with the newly found 5.6 $\mu$g $CH_4$ $g_{termite}^{-1}$ $h^{-1}$ would increase this regions estimate to 7.92 Tg, and the global estimate to 14.96 Tg.

Our study points out that termite emissions are still an uncertain source in the $CH_4$ budget, and are especially poorly quantified for the Amazon rain forest. Measurement of $CH_4$ emissions from different termite species, preferably covering species of different feeding or nesting habits, in combination with more precise termite distribution and abundance data, would allow more precise estimates and a better understanding of the role of termites in the $CH_4$ budget.
* * *
**Minor Comments:**

**L10 (L211, L284): Reads are easily be confused by the colony size and population, also the colony size of 50-120 thousands individuals and 54.6-116.6*10$^3$ termites per mound should be unified.**
Thank you for this point. We have improved this, and have now tried to use words for large numbers (as advised by the guidelines of Biogeosciences). We have unified this in the revised manuscript.

**L120: Change "mound 15" to "mound #15".**
We have made all mound numbering consistent by adding 'nr' every time a specific mound is mentioned, and we have used '#' when discussing a measurement repetition (For example, measurement #1, #2, and #3 of mound nr. 13.)

**L120: Only one control (blank) made this result (also see above) weaker.**
We have revised this part of the manuscript, as demonstrated at point 1 in this review.

**L130: The distance between the UGGA and chamber was 2 m.**
This tubing was of 2 meter length, but the distance was usually a little less. Two meter length was chosen to have some flexibility about where to place the UGGA.

**L131: It is about 350 mL/min (from LGR).**
We have corrected this.

**L150-157 (§2.5): Soil flux chamber had no mixing fan would have the same problem with the mound chamber (see above)**

For the small flux chamber, the volume is only 4.7 L, wherefore the circular LGR flow of 0.35 L/min induces basic chamber mixing. In addition, as found by different studies, a fan might induce unnatural turbulence, leading to an overestimation of the flux (Janssens et al. 2000, Pumpanen et al. 2004). Since we wanted to avoid overestimation of our fluxes, and since our $CO_2$ fluxes (without a fan), measured in different places in the ecosystem, are quite close to earlier studies (Chamber et al. 2004, Souza 2005, Zanchi et al. 2014), we decided to not install a small fan inside this chamber.

The revised manuscript text concerning these additional measurements can be found at point 5 of this review.

**L187-189: Move to the Method, and L189-192 move to the caption of Figure 4.**
Thank you for the suggestion, we have corrected this.

**L252-257: The statement of "air flow below the soil collar" does not make sense.**
We have rephrased the sentence.

**Equation 2: not completed; missed chamber pressure and chamber temperature.**
Since we are stating dC/dt in '$\mu$mol m$^{-3}$ s$^{-1}$, and not in '$\mu$mol mol$^{-1}$ s$^{-1}$, the pressure and chamber temperature term in this equation become redundant. We have chosen for this equation form, since we assume a stable temperature, as stated §2.4.

**L311: The statement of "Mound adjacent soil flux measurements showed no enhanced CH$_4$ and CO$_2$ fluxes in comparison to soils in the blank collar" does not consist with the results. For example, adjacent CO$_2$ flux (1.3) was almost three times of blank soil (0.47).**
Thank you for pointing this out. The revised manuscript text for this part is given in the beginning of this review (review point 5).

**L337: 11 g is the maximum value; the variation range should be listed. Consequentially, the following value of 0.5-1.0 nmol m$^{-2}$ s$^{-1}$ was overestimated.**
The biomass value of 11 g m$^{-2}$ has been stated and used as a standard for tropical rainforests in different previous studies (Bignell and Eggleton 2000, Sanderson, 1996, Sugimoto et al. 1998).

In addition, for our *local* ecosystem, the termite biomass estimate of 11 g termite m$^{-2}$ is **not** considered a maximum value, and possibly even an underestimation:

A recent paper links the termite biomass to GPP, thereby correcting the termite biomass estimate for less active tropical ecosystems (see figure S6 in Kirsche et al. 2013). Since we are only using the termite biomass estimate for our *local* ecosystem, for which the GPP has been estimated to be 3000 g C m$^{-2}$ year$^{-1}$ (Chambers et al. 2004), based on Figure S6 we deducted that the termite biomass is even higher than 11 g m$^{-2}$. This is also confirmed by a *local* study, performed in a fieldsite close by, where a termite biomass of 14-17 g m$^{-2}$ was found (Martius, 1998).

While the termite biomass is likely higher than 11 g m$^{-2}$ in our ecosystem, we prefer to stay in sync with previous studies on tropical ecosystems, and will continue with this lower bound appraisal for termite biomass.

**L415: Check the grammar.**

Thank you for pointing this out. We have revised this part to:

**Revised text in A1::** For calibration of the instrument, 2 calibration gases were used: Gas 1 with values 381.8 $\mu$mol $CO_2$ $mol^{-1}$, 2494.9 nmol $CH_4$ $mol^{-1}$, 336.6 nmol $N_2O$ $mol^{-1}$, 431.0 nmol CO $mol^{-1}$, and -7.95 permil $\sigma^{13}C$ of $CO_2$, and gas 2 with 501.6 $\mu$mol $CO_2$ $mol^{-1}$, 2127.0 nmol $CH_4$ $mol^{-1}$, 327.8 nmol $N_2O$ $mol^{-1}$, 256.7 nmol CO $mol^{-1}$, and -14.41 permil for $\sigma^{13}C$ of $CO_2$.

**A3: Shorten or discuss the scientific meaning of 13$CO_2$ in this study.**

We have shortened this part, and moved a part of the information to the figures caption. The new text is as follows:

[revised manuscript text omitted]

---

## Author Response (AR2)

**Hella van Asperen et al.**

v\_asperen@iup.physik.uni-bremen.de

Received and published: 3 February 2021

**Response to Referee 1**
Van Asperen and co-authors studied methane and  $CO_2$  emissions by a termite species at an upland site in the amazon basin. They report individual and mound-based emission factors comparable to previous studies, and suggest that methane emissions can

be employed as as rapid and non-invasive method to estimate mount populations.

**Strength:** The manuscript addresses a timely and important research question (methane emissions by termites) and provides a much needed data point in a previously understudied areas (termites in the neotropics). The authors followed state of the art measurements at a surely logistically challenging field location. As a bonus, the authors present both a comprehensive literature review and some very rare data on emissions of other trace gases (N2O, CO) in the appendix. The manuscript is generally well written and surely of great interest for the Biogeosciences readership.

**Limitations:** Some of the measurements were poorly replicated: Only one control collar was placed at distance from termite mounts, and for termite weight estimates, only one measurement is presented.

**Dear Lukas Kohl,**

Thank you for your kind words and the time you spent on reviewing our manuscript. We are grateful for your suggestions, which we have used to improve the manuscript. Below you will find a point to point response to each of your raised concerns and, if applicable, the corrected and improved manuscript text.

In addition, we would like to point out that:

 we have uploaded a revised text of §4.1 (First paragraph of discussion), which is shown at the end of this review;

- we have uploaded a revised Figure 2, which is shown at the end of this review (Previous Figure 4 is Figure 2 in revised manusript);
- we have uploaded the revised Table 2, which is shown at the end of this review (Previous Table 2 and 3 are now merged into Table 2);
- we have uploaded one additional figure, belonging to a discussion point in this review ('Review-Figure 1'), which is shown at the end of this review;
- the given values in the text might have changed due to an improved termite weight determination.

The choice for only one blank measurement was due to practical limitations, often a leading factor in these logistically challenging field conditions. To improve this part of the manuscript, we will substitute our blank measurement by additionally measured valley soil fluxes, performed as part of sub study.

Below we will:

- · report the values of the additionally measured soil valley fluxes;
- argue why these values are more suitable then the original blank control value;
- provide the revised manuscript text and the new Figure 4 wherein the soil valley measurements are shown.

**Additional measurements:** Additional flux measurements were done in the same week (March 2020) and performed in the same valley at approx. 500 m distance from the termite mounds. The chamber set up was as described in §2.5, with 10 soil collars

СЗ

and 3 repetitions. We observed soil CH4 fluxes ranging between -0.12 to 2.89 nmol  $m^{-2} s^{-1}$ , (median= -0.02, average=0.15, sd=0.55).

Fluxes from the *original* blank collar ranged between 3.9-5.4 nmol m-2 s-1 and were thereby higher than the additionally measured soil valley fluxes (-0.12 to 2.89 nmol m-2 s-1). The original blank collar fluxes were however quite similar to the mound adjacent fluxes (0.3-8.9 nmol m-2 s-1, 16 locations). While the blank collar was not closely located to a mound (~5 m of mound nr. 15), comparison to these 2 sets of measurements points at the presence of a local CH4 hotspot (Subke et al. 2018), thereby not being representative as a control collar. For the revised manuscript we will use the additional soil valley flux measurements as our 'blank collar'.

The aim of the blank 'control' measurement was to show the large difference between a 'normal' valley  $CH_4$  emission (per area), and an emission (per area) when a termite mound is present. Considering the average mound emission of 25.2 nmol mound-1 s-1, and the average valley soil emission of 0.03 nmol collar-1 s-1 (0.15 nmol m-2 s-1), an average collar area emits a factor 630 more  $CH_4$  when a termite mound is present. Including these complementary measurements will strengthen our message that termite mounds are hotspots in comparison to their surroundings.

We have included these additional measurements for comparison, by adapting Figure 4, and by adapting the manuscript text. Manuscript parts with major changes have been copied here below. In the revised manuscript, the original Figure 4 is now Figure 2.

In Methods, §2.5: Valley and mound adjacent soil fluxes

Every mound adjacent soil flux measurement was 4 minutes, and the set of 4 collar measurements

was performed once per mound, with exception of mound nr. 19. For mound nr. 13 and nr. 14, the measurements were performed on the  $2^{nd}$  measurement day, for mound nr. 15 and nr. 16, the measurements were done on the  $3^{rd}$  measurement day. Mound adjacent soil fluxes will be expressed per collar area (0.25 m2), to be better comparable to mound emissions. The same chamber set up was used in a sub study at a close by transect (~ 500 m from termite mounds) where, among others, valley soil fluxes were measured (10 collars, 3 repetitions). Measured soil fluxes from the valley will be shown for comparison.

**In Results, §3.1: Mound CH4 and CO2 emissions**

Headspace concentrations increased strongly during chamber closure, and chamber concentrations reached up to 5750 nmol CH4 mol-1 and 1950  $\mu$ mol CO2 mol-1. Mound CH4 emissions ranged between 17.0 and 34.8 nmol mound-1 s-1 (Fig. 1), with an average emission of 25.2 nmol mound-1 s-1. Additional valley measurements showed heterogeneous soil CH4 fluxes with small uptake and emission taking place alongside, ranging between -0.1 and 2.9 nmol m-2 s-1 (med=-0.02, avg=0.15, sd=0.54). Mound adjacent CH4 soil fluxes, measured at 20 and 45 cm from the mound, ranged between 0.4 and 8.9 nmol CH4 m-2 s-1 (avg=2.14, sd=2.00), and were on average enhanced in comparison to valley soils (Fig. 2). Soil valley CO2 fluxes were found to range between 0.9 and 3.7  $\mu$ mol m-2 s-1 (avg=2.14, sd=2.01), thereby being enhanced with respect to the surrounding soils (Fig. 2). Mound CO2 emissions, corrected for the average valley soil respiration, were ranging between 1.1 and 13.0  $\mu$ mol mound-1 s-1, with an average emission of 8.14  $\mu$ mol mound-1 s-1 (Fig 1).

**In Discussion, §4.3:**

Valley soil CH4 and CO2 fluxes were similar to what was found by earlier studies (Souza (2005), Moura (2012), Chambers et al. (2004), Zanchi et al. (2012). On average, mound adjacent soil CH4 and CO2 fluxes were enhanced with respect to valley soils, although differences were small, and no clear emission pattern with 'distance to mound' was observed. While mound adjacent soil fluxes are possibly enhanced, we preferred to avoid overestimation, and decided to treat termite mounds as very local hot spots, with measured fluxes only representative for the collar area of 0.25 m2. On average, CH4 and CO2 fluxes per collar area were found to be a factor  $\sim$ 630 and  $\sim$ 16 higher when an active termite mound was present.

**References:**

- Chambers, Jeffrey Q., et al. "Respiration from a tropical forest ecosystem: partitioning of sources and low carbon use efficiency." Ecological Applications 14.sp4 (2004): 72-88.

- Moura, V. S. d.: Investigação da variação espacial dos fluxos de metano no solo em floresta de terra firme na

Amazônia Central, MSc thesis INPA/UEA, 2012.

- Souza, Juliana Silva de. "Dinâmica espacial e temporal do fluxo de CO2 do solo em floresta de terra firme na Amazônia Central." (2005).

- Subke, Jens-Arne, et al. "Rhizosphere activity and atmospheric methane concentrations drive variations of methane fluxes in a temperate forest soil." Soil Biology and Biochemistry 116 (2018): 323-332.

**Termite weight estimates**

Following the suggestion of the reviewer we repeated the measurement and use an improved weight estimate in the revised manuscript.

We repeated the measurement as described in §2.7, with a larger sample size: we measured the weight of 4 samples of each 100 termites, which resulted in an average calculated weight of 2.832 mg, 2.986 mg, 3.085 mg and 3.141 mg. The former measurement (as given in previous manuscript) with 80 termites, gave an average weight of 3.330 mg. Averaging these 5 values results in a termite weight of 3.0748 mg (sd=0.1847).

Such variation in average termite weight can be expected, due to genetics and environmental differences during development. In addition, our values are close to the values as measured by Pequeno et al (2013), who reported a termite weight of 3.0 mg (sd=0.4) for the species *N. Brasiliensis*.

In the manuscript, we will use a termite weight of 3.07 mg (sd=0.18) for the species *N. Brasiliensis*, and we will indicate the propagated uncertainty range in the relevant calculations. The new termite weights lead to the following revised manuscript text:

**Previous text §2.7:** Termite mass was measured in the *Laboratory of Systematics and Ecology of Soil Invertebrates* at INPA. 80 living workers of the species *N. Brasiliensis* were weighted by use of a precision scale (FA2104N). Reported individual termite mass is fresh weight per termite (mg termite-1).

**Revised text §2.7:** Termite mass was measured in the *Laboratory of Systematics and Ecology of Soil Invertebrates* at INPA. 480 living workers of the species *N. Brasiliensis* were weighted in 5 subgroups (4x n=100, 1x n=80) by use of a precision scale (FA2104N). Reported individual termite mass is fresh weight per termite (mg termite-1).

**Previous text §3.2:** The living weight of 80 workers was measured to be 0.264 g, which is 3.3 mg per worker. This value is similar to what was found by Pequeno et al. (2017), who measured 3.0 ( $\pm$  0.4) mg for workers and 6.6 ( $\pm$  0.3) mg for soldiers. The species *N. Brasiliensis* has a relatively low soldiers:workers ratio of 1:100 (Krishna and Araujo, 1968). For our calculations we will use an average fresh weight of 3.33 mg termite-1 for the species *N. Brasiliensis*.

**Revised text §3.2:** The average weight of 5 subsets of living workers of the species *N. Brasiliensis* was determined, and found to range between 2.83 and 3.33 mg, with an average weight of 3.07 mg (sd=0.18), which is similar as what was found by Pequeno et al. (2013), who reported 3.0 mg (sd=0.4). Since the species *N. Brasiliensis* has a relatively low soldiers:workers ratio of 1:100 (Krishna and Araujo, 1968), we will use the worker weight 3.07 (sd= 0.18) mg termite-1 as an average termite weight for the species *N. Brasiliensis*.

**Possible improvement:**

• While the manuscript is generally very well written, I would encourage the authors to focus on editing the discussion section, which reads less easily than the rest of the manuscript. Some of this could be done by shortening and streamlining this section, which is rather long and at times meandering.

Thank you for the suggestion. We have shortened the discussion by following the different suggestions (see revised §4.1, as shown at the end of this review). In addition, we have taken out one figure (original Figure 3: mound volume and height vs emissions), since the content of the figure did not add much to the text.

**• The authors could also improve the quality figures and tables (see below), most importantly remove the grid lines from the figures for easier readability.**

We have re-plotted all figures following your suggestions. Figure 2 (original Figure 4) is shown at the end of this review.

**• Overall, this is a very nice contribution and it was a pleasure to review it!**

Thank you once more for your review and your comments!

**L48: 'which is around' - approximately instead of around, also better state the range in % as well given that 2-15 Tg is quite a wide range.**

We have replaced 'which is around' for 'approximately', and will express the range in %:

**Revised text:** More recent literature uses estimates in the range of 2-15 Tg  $CH_4$  per year (Ciais et al., 2014; Kirschke et al., 2013; Sanderson, 1996; Saunois et al., 2020), which is approximately 0.5-4% of the total estimated natural source  $CH_4$  emission (Saunois et al., 2020).

L64: 'termite  $CO_2$  measurements' - measurements of termite  $CO_2$  emissions We have corrected this.

L64: to avoid mixing weight units (gramm and tons), Pg instead of Gt We will use Pg instead of Gt.

**L119: what material was your chamber built out of?**

The large flux chamber (220L) was created from a bucket from polythene, and purchased at a common household store. The collars were made from stainless steel. The small flux chamber (4.7L) and the collars were created from a common PVC sewage pipe, purchased at a construction store.

The following text has been added to §2.3 and §2.5:

Revised text in §2.3: A flux chamber was created by use of a 220 L slightly cone-shaped polythene bucket.

Revised text in §2.5: The chamber and collars were created from a common PVC sewage pipe.

L130: when were your measurements conducted (date, in what season?) Measurements were performed in March 2020, in the wet season (stated in §2.1).

L134: 'molar density': concentration We have corrected this.

L142: 'increase': concentration change, as you could see uptake We have corrected this.

L176-181: The section could be improved.

We have corrected this. See below for the improved text.

**Revised text §3.1:** During chamber closure, the concentration changes in CH4 and CO2 were strongly correlated ( $R^2 > 0.95$  for each chamber closure). The ratio between the mound CH4 and CO2 emission (CH4/CO2) ranged between 2.1 and 17.1 \*10-3, and showed a constant ratio when data from mound 19 (furthest away from other mounds), and mound 6 (different species) were excluded (average ratio: 2.8\*10-3). The smallest mound (nr. 19) clearly showed smaller-than-average emissions, but in general no strong correlation was found between mound CH4 emissions and mound height ( $R^2$ =0.07) or volume ( $R^2$ =0.08), and a small correlation was found between mound CO2 emissions and mound height ( $R^2$ =0.44).

**L195: can you state an uncertainty of this weight per individual?**

Please see the beginning of this review for an elaborate answer.

L217: no need to state the original unit here, just state the values converted to the unit used in your study.

We have corrected this.

L223-235: I recommend streamlining/shortening this segment. Acknowledging mount uptake is important, but it's not the focus of your study and comes out of left field here. Focus on why this is important to understanding your results. We have shortened this part. The improved section §4.1 can be found at the end of this review.

**L237-241: I would move this comparison with literature data up to L215-219.** We have moved this part.

**L243: To be honest, these variations among individual measurements look pretty trivial to me and may not need such extensive discussion (which ends up questioning your measurements).**

We have reduced this part, but have kept one sentence (see revised §4.1). In case someone else would like to do similar measurements, it is good to be aware of this possible minimal transport below the collar.

L249-254: This can be tested by looking at the concentration curves within individual closures. If a relevant air exchange between chamber and ambient air occurred, concentrations should be non-linear (dCH4/dt decreasing over time, following a y=a + b\*e-c\*t function). If this is the case, fluxes should be calculated by fitting such an exponential function and calculating the slope as d[CH4/dt] at t=0.

We observed little variations in the linear increase, and variations were at the same moment and with the same magnitude for  $CO_2$  and  $CH_4$ . We expect that this is a result from minor air transport below the collar, possibly as a result of little disturbances (bag filling, a forest breeze, our presence close to the flux chamber). These fluctuations are not continuous, and the gradient recovers itself after a fluctuation. As can be seen in added figure (Review-Figure 1), the concentration increase still can be represented well by a linear increase with a strong  $R^2$  ( $R^2 > 0.95$ ).

**L280: it would be good to add an uncertainty range to your population estimates** Thank you for this suggestion.**

We have propagated the uncertainty of our emission factor 0.0002985 (se= $1.77*10^{-5}$ ), to define an uncertainty range in our population estimate. For example, for mound nr.

13, a range of 89.5-100.9 thousand termites will be given.

**L288: 'contemplated': considered?** We have corrected this.**

L291-292: 'hypothesize': don't use hypothesize that for claims you do not test. 'It is therefore likely that.' We have corrected this.

**L303: 'drawback': disadvantage** We have corrected this.

L304-305: 'is proposed': by whom? The authors? If that's that's the case, say so (ok, sorry for the snarky tone. Use active voice here - 'We propose a follow-up study to directly compare')

Thank you for the suggestion, this is indeed unclear. We have corrected this.

**L311: 'it was decided': same here, use active voice: 'we decided .. to avoid overestimating ..'**

We have corrected this.

L418-419: 'indicating no or very low  $N_2O$  emissions': Can you provide an uncertaintyrange for that estimate (e.g., limit of detection for fluxes?)

We have calculated a detection limit of 0.027 nmol N2O m-2 s-1. Here below we will:

 elaborate on how this detection limit is determined (specifications FTIRinstrument and assumptions);

- support our statement (very low N2O emissions) with additional data;
- give the revised manuscript text.

**Detection limit of measurements:** Reviewer 2 posed a similar question, and also asked about the precision and calibration of the FTIR-instrument. For completeness, we give the information here as well.

The FTIR-instrument has the following precision ( $\sigma$ ) for 10 minute-averaged spectral analyses: 0.02  $\mu$ mol mol-1, 0.2 nmol mol-1, 0.2 nmol mol-1, 0.06 nmol mol-1, and 0.04 permil, for respectively CO2, CH4, N2O, CO, and  $\delta^{13}$ C of CO2. Measurements performed during the campaign week were set to 5 minutes, so that a precision of 1/sqrt(N) is achieved, which is 0.09 nmol mol-1 for N2O. The FTIR instrument has been shown to be linear for all gases in the ambient concentration range, and linearity was tested for N2O in the range 300-350 ppb. For the detection limit, we state the following:

- Assuming bag samples taken at 2, 5 and 8 minutes during chamber closure.
- Given: collar area 0.25  $m^2,$  chamber volume 220 L, mound volume 50 L, headspace volume 220-50 = 170 L.
- Assuming: Molar volume of 24.5 L mol-1 (1 atm, 25 °C).
- Minimum detectable concentration difference is  $(2\sigma)$  0.18 nmol mol-1.
- A concentration difference between t=2 min and t=5 min of 0.18 nmol mol-1, is caused by a flux of 0.027 nmol collar/mound-1 s-1.

The FTIR-instrument has a cross sensitivity with CO2, which is well determined for CO2 <800  $\mu$ mol mol-1, but is less certain for high CO2 concentrations. For this C13

reason, we prefer to only use the N2O headspace concentration measurements with  $CO_2 < 800 \ \mu mol \ mol^{-1}$ . Only 5 mound chamber closures had two consecutive N2O concentration points (t=2 min and t=5 min) with  $CO_2 < 800 \ \mu mol \ mol^{-1}$ , and only 3 sets of two-consecutive concentration points passed the minimum concentration difference of 0.18 nmol mol-1. These differences were  $\sim 0.2, \ \sim 0.3$  and  $\sim 0.7 \ nmol \ mol^{-1}$ , leading to calculated N2O fluxes ranging between  $\sim 0.03 \ - \ \sim 0.11 \ nmol \ mound^{-1} \ s^{-1}$ .

Additional measurements to support statement 'very low N2O emissions': In October 2020, additional valley soil N2O flux measurement were performed with the same chamber system and collars (5 collars, 3 repetitions), but with a longer closing time (35 min), without termite mounds (so lower CO2), and with 4 measurements per chamber closure. Also during these measurements, concentration increases were very low. Out of 15 measurements, 8 measurements had an R2 > 0.90, and calculated fluxes ranged between 0.008-0.106 nmol m-2 s-1 (average=0.032 nmol m-2 s-1, sd=0.33). Since the valleys are known to be low on nitrogen (Quesada et al., 2010), such low fluxes are expected, and similar N2O valley soil fluxes were found by Matson et al., (1987) in a fieldsite closeby.

For the revised manuscript: since the 3 calculated mound N2O flux measurements are based on only 2 consecutive headspace concentration points, no uncertainty can be given, wherefore we preferred not to state the fluxes in the previous manuscript. For the revised manuscript, we state the detection limit, explain why not all mound fluxes could be calculated, and support our observation of low N2O mound fluxes by the additional soil N2O flux measurements:

**Appendix A2:** Gas samples (3 samples per chamber closure) revealed stable N2O concentrations, and headspace concentrations ranged between 333.7 and 342.4 nmol mol-1 over the different chamber closures. Since headspace CO2 concentrations sometimes exceeded 800  $\mu$ mol mol-1, and N2O-CO2

cross-sensitivity becomes uncertain at higher CO2 concentrations, not all 3 headspace samples per chamber closure could be used, wherefore qualitative N2O flux estimates cannot be reported. As a back-of-the-envelope calculation, N2O fluxes were calculated if 2 consecutive headspace samples were with CO2 <800  $\mu$ mol mol-1, and if a minimum N2O concentration difference of 0.18 nmol mol-1 was found (FTIR precision ( $\sigma$ ) for 5 min spectra is 0.09 nmol mol-1), which gave us 3 mound flux estimates ranging between ~0.03 and ~0.11 nmol N2O mound-1 s-1. Similarly low fluxes were found during additionally performed flux measurements, performed as part of a substudy, which showed valley soil fluxes ranging between 0.008-0.106 nmol N2O m-2 s-1. The low mound fluxes would be in agreement with a previous study which suggested that termite mound N2O emissions are dependent on the N-content of the termites diet (Brauman et al., 2015), which is expected to be low in the valleys of this ecosystem (Quesada et al., 2010).

**References:**

-Matson, Pamela A., and Peter M. Vitousek. "Cross-system comparisons of soil nitrogen transformations and nitrous oxide flux in tropical forest ecosystems." Global Biogeochemical Cycles 1.2 (1987): 163-170.

**Figures**

- Remove grid lines (counter-intuitively, this makes figures easier to read), place ticks inwards.
- Fig 1: remove 'per mount' on the y axis, it's redundant with the unit on that axis.
- Fig 4: A broken axis might work better than the inserts here (if you keep the inserts, state the y axis scale). The figure could also be simplified by showing the means + SD of the four mounts instead of values for individual mounts. Also, I think the direction in which you placed the soil collars from the mount wasn't chosen deliberately, so your x axis could be just 'distance from the center of the mount', combining your flux measurements at the same distance at either side of the mount.
- Fig 5: number instead of amount

Thank you for these suggestions.

- Figure 1: we have removed 'per mound', and have removed the gridlines.
- Figure 4 (now Fig 2): we have implement a 'broken y-axes', and have added additional measurements. We have chosen to keep the mound in the middle, to better visual the actual mound, and to visually separate the emissions measured on each side of the mound.
- Figure 5 (now Fig 4): we have corrected this.

**Tables**

I recommend combining Table 2 and 3 after removing reported value and reported unit (these can be placed in a supplement) to keep the table easier to read. State the unit of the converted values in the table header. This leaves the following columns: [Study] [Study area] [CH4 emission (state units)] [CO2 emission (state units)] [CH4:CO2 ratio (state units)] [Species]. Such a table would give a much better overview.

Thank you for this suggestion. We have merged the two tables, and have taken some columns out. Since it might not always be clear to which value we are referring, especially when data is taken from a graph, we prefer to also state the original value and unit. Nevertheless, we have tried to improve the readability by giving this part a smaller fontsize. The new table can be found at the end of this review.

**Revised Discussion part §4.1**

**$CH_4$ and $CO_2$ emissions**

Measured mound CH4 emissions were of similar magnitude to emissions found by previous studies (Table 2). The termite emission factor, determined for the soil-feeding species *N. brasiliensis*, was found to be 0.35 (sd= 0.02)  $\mu$ mol g-1termite h-1, which is similar to values found for other species in literature (Table 2, upper part), but almost two times higher than the average value reported by Martius et al (1993) for a wood-feeding species in the Amazon (0.19  $\mu$ mol CH4 g-1termite h-1). Our emission rate is within the reported range of 0.1-0.4  $\mu$ mol g-1termite h-1 for soil feeders (Sugimoto et al. 2000). Mound CO2 emissions and the termite CO2 emission factor were similar to a little higher in comparison to the few values found in literature. Nevertheless, since mound material and termites were measured together, the contribution of *indirect* termite termistes never.

There is a large variety in type of termite mounds (shape and size are dependent on, among others, species, ecosystem, climate (Noirot and Darlington, 2000)), explaining the wide range of reported termite mound  $CH_4$  emissions (Table 2, middle and lower part). In-situ measurement of termite mounds gives information about the *net*  $CH_4$  emission under natural conditions, but is unable to distinguish sources and sinks inside the mound. One known  $CH_4$  sink in termite mounds is the uptake by methanotrophic bacteria, which are also responsible for the  $CH_4$  uptake in aerobic soils. The presence and magnitude of this process have been discussed and reviewed by different studies (Khalil et al., 1990; Macdonald et al., 1998; Nauer et al., 2018; Seiler et al., 1984; Sugimoto et al., 1998a; Ho et al., 2013; Pester et al., 2007; Reuß et al., 2015). The role of possible mound  $CH_4$  uptake should also be acknowledged for the measurement of individual termite emissions (Table 2, upper part): most literature values, including values from this study, are based on termite incubation in presence of mound material, with ongoing  $CH_4$  uptake, wherefore actual termite  $CH_4$  emission values might be higher.

Small variation in emission magnitudes was observed between measurement days. This can be caused by a variation in colony size (due to foraging activities) or termite activity, driven by fluctuations in temperature or radiation (Jamali et al., 2011a; Ohiagu and Wood, 1976; Sands, 1965; Seiler et al., 1984).. However, as our termite mounds are in a tropical forest with relatively constant temperatures and only indirect daylight, strong diurnal temperature and radiation patterns are not expected. Small variation can also be caused by minimal air transport below the soil collar, through the porous upper soil layer; during preliminary tests *without* a collar, we observed that even a light forest breeze can cause chamber

headspace variations. In case our set up was subject to minor air transport below the collar, the given mound estimates will be slightly underestimated with respect to the actual mound fluxes. Another possible underestimation is caused by the estimated corrected chamber volume, as used in Eq. (2). In this study, we considered the mound volume as a solid body. A previous study considered the solid nest volume as 10% of the actual mound volume (Martius et al. 1993), leading to a larger corrected chamber volume, and therefore to larger calculated mound emissions. By use of this approach, average calculated emissions would increase by almost 30% to be  $32.7 \text{ nmol CH}_4 \text{ mound}^{-1} \text{ s}^{-1}$  instead of  $25.2 \text{ nmol CH}_4 \text{ mound}^{-1} \text{ s}^{-1}$ .

The mound emission  $CH_4/CO_2$  ratio was found to be relatively constant over 4 of the 5 mounds, with an average ratio of  $2.8 \times 10^{-3}$ . While values in literature indicate a wide range of reported  $CH_4/CO_2$  ratios (Table 2), both Seiler et al. (1984) as Jamali et al. (2013) found little variation between mounds of the same species, and concluded that the  $CH_4/CO_2$  emission ratio is species-specific. Our overall variation of a factor of ~4 for the  $CH_4/CO_2$  ratio of mound emissions of the same species is of the same magnitude as what was observed in earlier studies (Seiler et al., 1984; Jamali et al., 2013).

Figure 2. Measured mound emissions and mound-adjacent soil fluxes for CH4 (left) and CO2 (right) for mound nr. 13, nr. 14, nr. 15 and nr.16 expressed in much C25  $m^{-2}$  sr-1 for CH4 and  $\mu$ mol O25  $m^{-2}$  s  $r^{-1}$  for CO2 (collar area is 0.25  $m^{-2}$ ), Note that for CO2 here the net mound emissions per collar area, not corrected for soil respiration, are shown and stated. The centrally-placed markers are the measured mound emission dates for mound nr. 19; the larger marker indicates the day-specific mound emission when mound adjacent soil fluxess were measured. The grey bar indicates the range of additionally measured soil valley fluxes. The range and average flux for each group of measurements are given in the table. On average measured mound CH4 and CO2 fluxes were a factor 630 and 16 higher in comparison to the surrounfing soil valley fluxes.

Fig. 1. Revised Figure 2 (previously Figure 4)

22

Table 2. Overview of literature values for  $CH_4$  and  $CO_2$  emission of termites per weight (upper part), emission per termite mound (middle part), and emission per area (lower part). Values from this study are indicated in bold. If reported, the average and sd are given, otherwise a range is indicated. If multiple values were reported, measurements from higher soil-feeding termite species were selected. For each study, the graph or table where the data was found, is indicated. The  $CH_4/CO_2$  is given in molar ratio ( $10^{-3}$ ), a) Sawadogo et al. (2011) reported emissions per dry weight mass. To convert to fresh weight, a formula as reported DP equeno et al. (2017) was used. With an assumed dry weight of 0.5 mg, a conversion factor of 3.14 was deducted. b) Mound emissions are divided by collar area of 0.25 m2; c) Calculated based on average values in this table; d) *Necoapritemes brainliensis*; e) *Crenetimes* shouralis, *Cubitermes* spingliaber, *Cubitermes speciensus*, *Noditermes* spin, *Thoracotermes ancorhorax*; f) *Dicupiditermes* sunitohii, *Dicupiditermes* nenorous, *Periopritemes* senariber, *Procepritermes* nanoeras, *Banderas*; Homallotermes cleanorae, *Proaciculitermes* spin, *Procephatus*, *Nasutitermes* area univoj; j) *Noditermes* spin, *Chotemeres allobartanis*, *Cubitermes* spin, *Chotemeres allobartanis*, *Cubitermes* spin, *Chotemeres* allobartanis, *Cubitermes* spin, *Chotemeres* diluternatis, *Cubitermes* spin, *Chotemeres* allobartanis, *Cubitermes* spin, *Chotemeres* allobartanis, *Cubitermes* spin, *Chotemeres* diluternatis, *Cubitermes* spin, *Chotemeres* allobartanis, *Cubitermes* spin, *Chotemeres* allobartanis, *Cubitermes* spin, *Chotemeres* spin, *Chotemeres* allobartanis, *Cubitermes* spin, *Discoprittermes* spin,

|                                       |              |                                                                        | Studies reporting emission                                            | per gram termit                                             | ie                                                                           |                     |                             |
|---------------------------------------|--------------|------------------------------------------------------------------------|-----------------------------------------------------------------------|-------------------------------------------------------------|------------------------------------------------------------------------------|---------------------|-----------------------------|
| Study                                 | Study area   | $CH_4$ emission (µmol g tm -1 h -1 )  |                                                                       | $CO_2$ emission ( $\mu$ mol $g_{im}^{-1}$ h -1 ) |                                                                              | $CH_4/CO_2$         | Species                     |
| This study, Fig. 4                    | Amazon       | 0.35 (0.2)                                                             | (0.0002985 nmol im -1 s -1 )                    | 86.8 (10.0)                                                 | $(0.074 \text{ mmol tm}^{-1} \text{ s}^{-1})$                                | ~4°                 | Soil feeders (d)            |
| Brauman et al. (1992), Tab. 1         | Congo        | 0.39-1.09                                                              | $(0.39-1.09 \ \mu \text{ mol } g_{\text{fm}}^{-1} \text{ h}^{-1})$    |                                                             |                                                                              |                     | Soil feeders (e)            |
| Eggleton et al. (1999), Tab. 4        | Australia    | 0.17-0.27                                                              | $(0.17 \cdot 0.27 \ \mu \text{ mol g}_{1m}^{-1} \text{ h}^{-1})$      | 1.4-9.0                                                     | $(1.4-36.4 \mu mol g_{tm}^{-1} h^{-1})$                                      | 10-154              | Soil feeders (f)            |
| Fraser et al. (1986), Fig. 2          | Australia    | 0.04 (0.01)                                                            | $(0.67 (0.2) \text{ mg kg}_{tm}^{-1} h^{-1})$                         | 107 (4.5)                                                   | $(4.7 (0.2) g kg_{tm}^{-1} h^{-1})$                                          | $\sim 0.38^{\circ}$ | Wood feeders (g)            |
| Konaté et al. (2003), Tab. 1          | Ivory Coast  |                                                                        |                                                                       | 31.4-133.5                                                  | (31.4-133.5 amol mgm -1 h -1 )                         |                     | Fungi feeders (h)           |
| Martius et al. (1993), Tab. 1         | Amazon       | 0.19 (0.08)                                                            | $(3.0 (1.3) \mu g g_{tm}^{-1} h^{-1})$                                |                                                             |                                                                              |                     | Wood feeders (i)            |
| Rouland et al. (1993), Tab. 1         | Congo        | 0.53-1.09                                                              | $(0.53-1.09 \ \mu \text{ mol } g_{\text{tm}}^{-1} \text{ h}^{-1})$    |                                                             |                                                                              |                     | Wood feeders (j)            |
| Sawadogo et al. (2011), Tab. 1        | Burkino Faso | 0.10-0.12                                                              | $(0.30-0.39 \ \mu \text{ mol } g_{\text{fm}}^{-1} \ h^{-1})^{\alpha}$ | 19-25                                                       | (59.4-78.4 µmol g 0m -1 h -1 ) a | $\sim 5^{c}$        | Wood feeders (k) |
| Sugimoto et al. (1998a), Tab. 3       | Thailand     | 0.03-0.20                                                              | $(3.4-20.3*10^{-8} \text{ mol } g_{1m}^{-1} \text{ h}^{-1})$          |                                                             |                                                                              |                     | Soil feeders (1)            |
|                                       |              |                                                                        | Studies reporting emission                                            | er nest or mour                                             | nd                                                                           |                     |                             |
| Study                                 | Study area   | $CH_4$ emission (µmol mound -1 h -1 )            |                                                                       | $CO_2$ emission (mmol mound -1 h -1 ) |                                                                              | $CH_4/CO_2$         | Species                     |
| This study, Fig. 1                    | Amazon       | 61-125                                                                 | (17.0-34.8 nmol mound -1 x -1 )                 | 4-47                                                        | $(1.1-13.0 \mu mol mound^{-1} s^{-1})$                                       | 2.8 (0.4)           | Soil feeders (d)            |
| Khalil et al. (1990), Fig. 4 & Tab. 3 | Australia    | 9-135                                                                  | (0.04-0.6 µg mound=1 s=1)                                             | 4-92                                                        | (0.05-1 µg mound = 1 s = 1)                                                  | 0.12-11             | Wood feeders $^{(m)}$       |
| MacDonald et al. (1999), Tab. 4       | Cameroon     | 1-11                                                                   | (4.5-49 ng mound -1 x -1 )                      |                                                             |                                                                              |                     | Soil & wood feeders         |
| Martius et al. (1993), Tab. 1         | Amazon       | 125 (150)                                                              | (2.0 (2.4) mg nest -1 h -1 )                    |                                                             |                                                                              |                     | Wood feeders (i)            |
| Seiler et al. (1984), Tab. 1          | South Africa | 1-644                                                                  | (0.02-10.3 mg nest - 1 h - 1)                                         | 0.7-241                                                     | (0.03-10.6 g not -1 h - 1)                                                   | 0.07-8.7            | Soil & wood feeders         |
| Sugimoto et al. (1998a), Tab. 3       | Thailand     | 0.4-1.9                                                                | (4.2-18.7*10 -7 mol nest -1 h -1 )   |                                                             |                                                                              |                     | Soil feeders (1)            |
|                                       |              |                                                                        | Studies reporting emis                                                | sion per area                                               |                                                                              |                     |                             |
| Study                                 | Study area   | CH 4 emission ( $\mu$ mol m -2 h -1 ) |                                                                       | $CO_2$ emission (mmol m -2 h -1 )     |                                                                              | $CH_4/CO_2$         | Species                     |
| This study, Fig. 1                    | Amazon       | 245-501 b                                                   | (17.0-34.8 nmol mound -1 x -1 )                 | 16-187 b                                         | $(1.1-13.0 \mu \text{mol mound}^{-1} \text{s}^{-1})$                         | 2.8 (0.4)           | Soil feeders (e) |
| Brümmer et al. (2009a), Fig. 5        | Burkino Faso | 315.7                                                                  | $(3788.9 \ \mu g \ CH_4 \cdot C \ m^{-2} \ h^{-1})$                   | 37.3                                                        | $(447.0 \text{ mg CO}_2 \cdot C \text{ m}^{-2} \text{ h}^{-1})$              | $\sim 8.5^{\circ}$  | Soil feeders (p)            |
| Jamali et al. (2013), Fig. 1          | Australia    | 32-500                                                                 | $(379-6000 \mu g CH_4 \cdot C m^{-2} h^{-1})$                         | 0-129                                                       | $(0.1550 \text{ mg CO}_2 \cdot C \text{ m}^{-2} \text{ h}^{-1})$             | 2.7-11.0            | Wood feeders (q)            |
| Ourieur (2004) Tels 4                 | Amazon       | 10.24                                                                  | $(0.16, 0.32, \dots, -2, k-1)$                                        |                                                             |                                                                              |                     | unknown                     |

Fig. 2. Revised Table 2 (merge of Table 2 and 3)

28

Fig. 3. Review Figure 'Review-Figure 1'

Interactive comment on "The role of termite  $CH_4$  emissions on ecosystem scale: a case study in the Amazon rain forest" by Hella van Asperen et al, Received and published: 22 December 2020

This manuscript presents a well thought out study to quantify methane emissions by termites in the Amazon rain forest. The authors reviewed the literature extensively and compared/discussed with their findings. I have some comments

that I think will make the study more valuable.

Thank you for your kind words and the time you spent on reviewing our manuscript. We are grateful for your suggestions, which we have used to improve the manuscript. Below you will find a point to point response to each of your raised concerns and, if applicable, the corrected and improved manuscript text.

In addition, we would like to point out that the given termite emission estimates have changed due to an improved termite weight determination.

1. Please provide the estimate of  $CH_4$  emissions by termite and put in context with the overall  $CH_4$  budget globally or in the Amazon. This manuscript presents  $CH_4$  emission factors only. Without knowing how many termite mounds in Amazon, it's difficult to imagine the scale of the global  $CH_4$  budget. I think this is one of a key messages for readers.

Thank you for this interesting point. Below we will:

- elaborate on our considerations regarding the termite mound-upscaling;
- provide a 'back of the envelope' estimate on the role of termite emissions in the Amazon CH4 budget, and in the global CH4 budget;
- show the revised manuscript text.

Termite mound upscaling: Based on mound density numbers, it is difficult to state

a termite CH4 emission estimate for the whole Amazon. As stated in the discussion (§4.3), mound density numbers vary largely between ecosystems. There is only little data available on mound density numbers, and most Amazon studies were performed in close proximity to our fieldsite (due to the research activities of local institute INPA). While this relatively large amount of local studies is unique and useful for the upscaling for our *local* ecosystem, it is unwise to assume that these mound density numbers apply to the whole Amazon. For this reason, we choose to only state a mound CH4 emission estimate for our specific ecosystem, and to inform the readers about the limitations of this estimate.

**Back-of-the-envelope estimate for the global CH**4 **budget:** By use of the data presented in the comprehensive modeling study of Kirsche et al. (2013), the following back-of-the-envelope estimate can be made:

Kirsche et al (2013) (Table 1) stated an annual global termite emission of 11 Tg CH4 year-1. They state that 36% of termite emissions originate from the region 'tropical South America' (p 818, first sentence), which calculates to 3.96 Tg CH4 year. Substituting the used termite emission factor of 2.8  $\mu$ g CH4 g-1termite h-1 by the value found in our study of 5.6  $\mu$ g CH4 g-1termite h-1, would lead to a doubling of the regions estimated termite emission, namely 7.92 Tg instead of 3.96. The global estimate would increase from 11 Tg to 14.96 Tg.

The termite emission factor is a practical estimate of the average termite emission, which can be used for CH4 budget studies. Since our study only measured one termite species, and there is likely a variation between species and ecosystems, we do not suggest that the currently used termite emission factor of 2.8  $\mu$ g CH4 g-1termite h-1 should be replaced by our value. We do however want to show and point out that the termite emission factor is still an uncertain part in the tropical CH4 budget.

СЗ

To include the reader in this train-of-thought, we have revised this part of the manuscript:

**Revised text in §4.3:** As a 'back-of-the-envelope' calculation, based on Kirsche et al. (2013): 36% of global termite emission (11 Tg) is expected to come from the region of 'tropical South America' (0.36\*11=3.96 Tg). Substituting the emission factor of 2.8 with the newly found 5.6  $\mu$ g CH4 g-1termite h-1 would increase this regions estimate to 7.92 Tg, and the global estimate to 14.96 Tg.

Our study points out that termite emissions are still an uncertain source in the  $CH_4$  budget, and are especially poorly quantified for the Amazon rain forest. Measurement of  $CH_4$  emissions from different termite species, preferably covering species of different feeding or nesting habits, in combination with more precise termite distribution and abundance data, would allow more precise estimates and a better understanding of the role of termites in the  $CH_4$  budget.

2. The first sentence in the Introduction section, it says "Methane (CH4) is the second most important long-lived anthropogenic greenhouse gas." I think CH4 has been recognized to be "short-lived" climate pollutant.

Thank you for pointing this out. We have changed the first sentence to:

**Revised text:** Methane  $(CH_4)$  is the second most important anthropogenic greenhouse gas, but its natural sources are still not well understood.

3. In the Introduction section, Line 35, it says "Recently, it was shown that termites have a mitigating effect during droughts in tropical rain forests". Please elaborate what mitigating effect.

Ashton et al. (2019) performed a termite suppression experiment and found that termite activity increased during drought, resulting in accelerated litter decomposition, elevated soil moisture, greater soil nutrient heterogeneity, and higher seedling survival rates. The authors suggested different underlying mechanisms for this response such as more favorable conditions for tunneling (e.g., drier, less-waterlogged ground), increased foraging ability above ground in the absence of heavy rain, and/or reduced predation pressure from ants.

We have changed the text in the Introduction to:

**Previous text:** Recently, it was shown that termites have a mitigating effect during droughts in tropical rain forests.

**Revised text:** Recently, it was shown that termites increase their activity during droughts, resulting, among others, in enhanced litter decomposition, elevated soil moisture and higher seedling survival rates, thereby demonstrating a mitigating effect during droughts in tropical rain forests.

**4. In the Introduction and in Appendix, the authors touched on $N_2O$ emissions from termite but didn't give conclusive results.**

We agree that this point is not sufficiently discussed. An elaboration on this subject can be found at point 7.

5. Section 2.3, Line 129, LGR GHG analyzer was mentioned to be the instrument deployed to quantify  $CH_4$  emissions in flux chambers. I think authors should add brief instrument performance specifications and details of what calibration and drift evaluation have been done in Amazon. While the absolute  $CH_4$  concentrations in flux chamber measurements are not very critical, since it's to measure the  $CH_4$  concentration increase, but the manuscript does not provide the measured concentrations and jumped directly to the emission factor estimates. For example, LGR UGGA precision is about 2 ppb. Does it perform the same in Amazon? Also, what  $CH_4$  concentration increments measured in the flux chambers? If it was only 2 ppb, then that data would not be useful. I think it should be many times more than the instrument precision and drift.

Thank you for raising this point. During the campaigns, we have set the Los Gatos instrument to the 10-second averaging modus. Calibration gases were measured every second day for 5 minutes, resulting in a precision (1 $\sigma$ ) of ~0.7 ppm and ~3.0 ppb for respectively CO2 and CH4.

The concentration increases during the 20 min chamber closure were large. Concentrations were climbing from forest concentrations to concentrations of up to 5750 ppb  $CH_4$ , and up to 1950 ppm  $CO_2$ , thereby far exceeding the measurement precision of the Los Gatos instrument.

We have added the following lines to the revised manuscript (beginning of Results):

**Revised text in §3.1:** Headspace concentrations increased strongly during chamber closure, and chamber concentrations reached up to 5750 nmol CH4 mol and 1950  $\mu$ mol CO2 mol-1.

6. Well-designed flux chambers should have a small mixing fan or internal distribution tubing to quantify fluxes. §2.5 describes how LGR sampling tubes were connected on top of a 220 L chamber, if the air inside is not well mixed, the two fittings on top of the chamber may not detect  $CH_4$  at the bottom of the chamber.

Thank you for raising this point. Below we will:

- · clarify the locations of the inlet fittings;
- elaborate on why we did not install a fan, and how we ensured mixed chamber air;
- give the revised manuscript text.

The 220 L chamber had two fittings on each side of the bucket while the smaller soil chamber had the two fittings on top of the chamber. Re-reading §2.5, we agree with the reviewer that the text is confusing, and we have revised this part.

As a small side note, termite mounds emit  $CH_4$  from its entire surface, thereby presenting a sphere-shaped source of 45-65 cm height *inside* the chamber head space. Therefore, we do not expect a large difference between  $CH_4$  concentrations at the top and the bottom of the chamber headspace.

We were hesitant about installing a small mixing fan. On the one hand, the absence of a mixing fan might lead to an underestimation of the flux (Christiansen et al. 2011). On the other hand, a mixing fan might lead to turbulence in the head space (Janssens

et al. (2000), Pumpanen (2004)), which possibly induces unrepresentatively high  $CH_4$  emissions from the mound.

Since we wanted to avoid overestimation of termite mound  $CH_4$  fluxes, we decided to not install a mixing fan. Instead we installed a 4 inlet vertical sampling tube inside the chamber head space, a technique to minimize the effects of gas concentration gradients in the head space (Clough et al, 2020). Inside the chamber at fitting height (~30 cm), a T-piece with two 20 cm-long Teflon tubing was positioned vertically, and two small incisions were made, so that head space air was sampled from 4 different heights (approx. at 10, 25, 35 and 50 cm height from the soil). The sampling tube was tested in the lab to verify whether air was sampled from all 4 inlets.

We have added the following lines and references to the revised manuscript:

**Revised text in §2.3:** Two one-touch fittings (1/4 inch, SMC Pneumatics) were installed on each side of the bucket. To minimize the possible effects of gas concentration gradients in the headspace, we installed a 4 inlet vertical sampling tube inside the chamber, so that air was sampled from different heights ( $\sim$ 10,  $\sim$ 25,  $\sim$ 35 and  $\sim$ 50 cm) in the headspace (Clough et al, 2020).

Revised text in §2.5: To be able to connect the Los Gatos instrument, the soil chamber had two one-touch fittings on top.

**References**

- Christiansen, Jesper Riis, et al. "Assessing the effects of chamber placement, manual sampling and headspace mixing on CH4 fluxes in a laboratory experiment." Plant and soil 343.1-2 (2011): 171-185.

- Clough, Timothy J., et al. "Global Research Alliance N2O chamber methodology guidelines: Design considerations." Journal of Environmental Quality 49,5 (2020): 1081-1091.

- Janssens, Ivan A., et al. "Assessing forest soil CO2 efflux: an in situ comparison of four techniques." Tree physiology 20.1 (2000): 23-32.

- Pumpanen, Jukka, et al. "Comparison of different chamber techniques for measuring soil CO2 efflux." Agricultural and Forest Meteorology 123.3-4 (2004): 159-176

7. Appendix A1 and A2 talk about N2O calibrations and measured concentrations. The measured N2O concentrations are outside of the calibration range. While the lower range (333.7 ppb) is similar to NOAA's measurements in Brazil, the manuscript does not provide the FTIR instrument precision and therefore, it's difficult to determine whether the detected range (333.7-342.4 ppb) is within instrument drift or it's actually an increment of N2O. I don't think the authors can conclude there isn't N2O emissions.

Thank you for pointing this out. Below we will:

- explain why FTIR N2O concentration measurements outside the calibration range can be used, by stating the precision and linearity of this instrument;
- explain why we can conclude that there are low  $N_2O$  emissions, by calculating the methods detection limit;
- support our statement (very low N2O emissions) with additional data.

First of all, to clarify, the mentioned range of 333.7-342.4 ppb was measured over *all* chambers during the whole week. Actual increments during individual chamber closures were a lot smaller, as discussed here below. We have clarified this in the revised manuscript text.

The FTIR-instrument has the following reported precision (1 $\sigma$ ) for 10 minute-averaged spectral analyses: 0.02  $\mu$ mol mol-1, 0.2 nmol mol-1, 0.2 nmol mol-1, 0.06 nmol mol-1, and 0.04 permil, for respectively CO2, CH4, N2O, CO, and  $\sigma^{13}$ C of CO2.

Measurements performed during the campaign week were set to 5 minutes, so that a precision of 1/sqrt(N) is achieved, which is 0.09 nmol mol-1 for N2O.

The FTIR instrument has been shown to be linear for all gases in the ambient concentration range, and linearity was tested for  $N_2O$  in the range 300-350 ppb. So while the choice of calibration gases was not optimal, we are confident that the FTIR-instrument still performs well in this concentration range.

**Detection limit of measurements:**

As also requested by reviewer 1, we calculated the minimum  $N_2O$  flux detectable by this instrument and method:

- Assuming bag samples taken at 2, 5 and 8 minutes during chamber closure.
- Given: collar area 0.25  $m^2,$  chamber volume 220 L, mound volume 50 L, headspace volume 220-50 = 170 L.
- Assuming: molar volume of 24.5 L mol-1 (1 atm, 25 °C).
- Minimum detectable concentration difference is  $(2\sigma)$  0.18 nmol mol-1.
- A concentration difference between t=2 min and t=5 min of 0.18 nmol mol-1 is caused by a flux of 0.027 nmol collar/mound-1 s-1.

So, given the parameters above, the chamber set up has a detection limit of  $\,$  0.027 nmol mound^{-1} s^{-1}.

The FTIR-instrument has a cross sensitivity with CO2, which is well determined for CO2 <800  $\mu$ mol mol-1, but is less certain for unnaturally high CO2 concentrations.

For this reason, we preferred to only use the N2O headspace concentration measurements with CO2 <800  $\mu$ mol mol $^{-1}$ . Only 5 mound chamber closures had two consecutive N2O concentration points (t=2min and t=5min) with CO2 <800  $\mu$ mol mol $^{-1}$ , and only 3 sets of two-consecutive concentration points passed the minimum concentration difference of 0.18 nmol mol $^{-1}$ . These differences were  $\sim$ 0.2,  $\sim$ 0.3 and  $\sim$ 0.7 nmol mol $^{-1}$ , leading to a calculated N2O flux of  $\sim$ 0.03 -  $\sim$ 0.11 nmol mound $^{-1}$  s^{-1}.

Additional measurements to support statement 'very low N2O emissions': In October 2020, additional valley soil N2O flux measurement were performed with the same chamber system and collars (5 collars, 3 repetitions), but with a longer closing time (35 min), without termite mounds (so lower CO2), and with 4 measurements per chamber closure. Also during these measurements, concentration increases were very low. Out of 15 measurements, 8 measurements had an R2>0.90, and calculated fluxes ranged between 0.008-0.106 nmol m-2 s-1 (average=0.032 nmol m-2 s-1, sd=0.33). Since the valleys are known to be low on nitrogen (Quesada et al., 2010), such low fluxes are expected, and similar N2O valley soil fluxes were found by Matson et al (1987) in a fieldsite closeby.

Since the 3 calculated mound N2O flux measurements are based on only 2 consecutive headspace concentration points, no uncertainty can be given, wherefore we preferred not to state the fluxes in the previous manuscript. For the revised manuscript, we have stated the detection limit, explain why not all mound fluxes could be calculated, and support our observation of low N2O mound fluxes by the additional soil N2O flux measurements:

**Appendix A2:** Gas samples (3 samples per chamber closure) revealed stable N2O concentrations, and headspace concentrations ranged between 333.7 and 342.4 nmol mol-1 over the different chamber closures. Since headspace CO2 concentrations sometimes exceeded 800  $\mu$ mol mol-1, and N2O-CO2

cross-sensitivity becomes uncertain at higher CO2 concentrations, not all 3 headspace samples per chamber closure could be used, wherefore qualitative N2O flux estimates cannot be reported. As a back-of-the-envelope calculation, N2O fluxes were calculated if 2 consecutive headspace samples were with CO2 <800  $\mu$ mol mol-1, and if a minimum N2O concentration difference of 0.18 nmol mol-1 was found (FTIR precision ( $\sigma$ ) for 5 min spectra is 0.09 nmol mol-1), which gave us 3 mound flux estimates ranging between ~0.03 and ~0.11 nmol N2O mound-1 s-1. Similarly low fluxes were found during additionally performed flux measurements, performed as part of a substudy, which showed valley soil fluxes ranging between 0.008-0.106 nmol N2O m-2 s-1. The low mound fluxes would be in agreement with a previous study which suggested that termite mound N2O emissions are dependent on the N-content of the termites diet (Brauman et al., 2015), which is expected to be low in the valleys of this ecosystem (Quesada et al., 2010).

**References:**

-Matson, Pamela A., and Peter M. Vitousek. "Cross-system comparisons of soil nitrogen transformations and nitrous oxide flux in tropical forest ecosystems." Global Biogeochemical Cycles 1.2 (1987): 163-170.

Interactive comment on "The role of termite  $CH_4$  emissions on ecosystem scale: a case study in the Amazon rain forest" by Hella van Asperen et al.

This study presented a global interesting issue of termite  $CH_4/CO_2$  emission in an Amazonian tropical rainforest. As a case study, this in-situ measurement of termite mound emissions provided information about termite  $CH_4/CO_2$  production under natural conditions, it will contribution some knowledge to Biogeosciences. However,

the field experiment was not well designed, and the limited data was not well analyzed. I would like to encourage the authors to revise the manuscript following my comments.

**Thank you for your time spent on reviewing our submission!** We are grateful for your suggestions, which we have used to improve the manuscript. Below you will find a point to point response to each of your raised concerns and, if applicable, the corrected and improved manuscript text.

In addition we would like to point out that:

- we have uploaded a revised text of §4.1 (First paragraph of discussion), which is shown at the end of this review;
- we have uploaded a revised Figure 2, which is shown at the end of this review (Previous Figure 4 is Figure 2 in revised manusript);
- we have uploaded a revised Figure 4, which is shown at the end of this review (Previous Figure 5 is Figure 4 in revised manusript);
- we have uploaded 4 additional figures, belonging to point 7 and 8 of this review, which are shown at the end of this review;
- the given values in the text might have changed due to an improved termite weight determination.

**General comments**

1. "The blank measurements (collar with only soil and litter) showed an average

CH4 emission of 1.15 nmol collar-1 s-1" (L175) means the forest soil was a VERY LARGE CH4 SOURCE (4.6 nmol m-2 s-1 or 23.2 CH4 ha-1 y-1). It was a FUNDAMENTAL PROBLEM! Actually, the "blank" soil should be CH4 sink. Even "1.15 nmol collar-1 s-1" was "-1.15 nmol collar-1 s-1", the soil CH4 sink of "-23.2 kg CH4 ha-1 y-1" was an unbelievable large value.

Though the reviewer correctly points out that most tropical forest soils are methane sinks, soil methane emissions in tropical ecosystems are common, especially when anaerobic conditions occur. Therefore, we disagree that this points to a fundamental problem.

In the revised manuscript we will substitute our blank collar measurement by a set of additional measurements from the surrounding area. These measurements show that the methane fluxes from the valley soil are spatially heterogeneous, but in general low. It is important to note that this heterogeneity has no impact on the given  $CH_4$  emission estimates from the termite mounds, since the emissions measured from the mounds are on average a factor 627 higher than the average background soil  $CH_4$  emission.

Below we will:

- provide additional information (measurements and literature) which show that soil valley CH4 fluxes are heterogeneous but of low magnitude in comparison to the measured mound fluxes;
- compare the soil and mound fluxes by providing an improved Figure 4;
- provide text for the revised manuscript.

**C3**

**Additional soil valley flux measurements**

Most tropical forest soils are methane sinks (Dutaur and Verchot, 2007, Kiese et al. 2003). Nevertheless, soil methane emission in tropical ecosystems can still be observed (Carmo et al. 2006), especially when anaerobic conditions occur, such as which can be found in the valley (Sihi et al. 2020, Moura et al. 2012). This is also observed by our set of additional measurements:

**Additional measurements:** valley soil chamber flux measurements (small chamber set up as described in §2.5), 10 soil collars, 3 repetitions, ~500 m from manuscript termite mounds, 5-50 m from igarapé (stream), measured in same week as termite mounds (March 2020), soil CH4 fluxes ranged between -0.12 to 2.89 nmol m-2 s-1, (median=-0.02, average=0.15, sd=0.55).

Our additional measurements show that valley soil fluxes are heterogeneous, and in general negative (median=-0.02), but that locations with relative high emissions (hotspots) can be found. Our mound adjacent soil fluxes were in general higher (0.3-8.9 nmol CH4 m-2 s-1, 16 soil collars), showing that mound adjacent soils are deviating from the average valley soil, likely due to the nearby presence of an active termite mound.

The magnitude of the *original* blank collar fluxes (3.9-5.4 nmol  $CH_4 m^{-2} s^{-1}$ ) is quite similar to the magnitude of mound adjacent fluxes (0.3-8.9 nmol  $CH_4 m^{-2} s^{-1}$ ). While the blank collar was not directly located next to a mound ( $\sim$ 5 m of mound nr. 15), the comparison with the different datasets points at the presence of a local  $CH_4$  hotspot (Subke et al. 2018), thereby not being representative as a control collar. For the revised manuscript we will use the 10 additional soil collar measurements as our 'blank collar' reference point.

The aim of the blank 'control' measurement was to show the large difference between a 'normal' valley  $CH_4$  emission (per area), and an emission (per area) when a termite mound is present. Considering the average mound emission of 25.2 nmol mound-1 s-1, and the average valley soil emission of 0.03 nmol collar-1 s-1 (0.15 nmol m-2 s-1), an average collar area emits a factor 630 more  $CH_4$  when a termite mound is present. Including these complementary measurements will strengthen our message that termite mounds are hotspots in comparison to their surroundings.

We have included these additional measurements for comparison, by adapting Figure 4 (now renumbered as Figure 2, see end of this review), and by including these measurements at the following places in the manuscript:

**In Methods, §2.5: Valley and mound adjacent soil fluxes**

Every mound adjacent soil flux measurement was 4 minutes, and the set of 4 collar measurements was performed once per mound, with exception of mound nr. 19. For mound nr. 13 and nr. 14, the measurements were performed on the  $2^{nd}$  measurement day, for mound nr. 15 and nr. 16, the measurements were done on the  $3^{rd}$  measurement day. Mound adjacent soil fluxes will be expressed per collar area (0.25 m2), to be better comparable to mound emissions. The same chamber set up was used in a sub study at a close by transect (~ 500 m from termite mounds) where, among others, valley soil fluxes were measured (10 collars, 3 repetitions). Measured soil fluxes from the valley will be shown for comparison.

**In Results, §3.1: Mound CH4 and CO2 emissions**

Headspace concentrations increased strongly during chamber closure, and chamber concentrations reached up to 5750 nmol  $CH_4$  mol-1 and 1950  $\mu$ mol  $CO_2$  mol-1. Mound  $CH_4$  emissions ranged between 17.0 and 34.8 nmol mound-1 s-1 (Fig. 1), with an average emission of 25.2 nmol mound-1 s-1. Additional valley measurements showed heterogeneous soil  $CH_4$  fluxes with small uptake and emission taking place alongside, ranging between -0.1 and 2.9 nmol m-2 s-1 (med=-0.02, avg=0.15, sd=0.54). Mound adjacent  $CH_4$  soil fluxes, measured at 20 and 45 cm from the mound, ranged between 0.4 and

8.9 nmol CH4 m-2 s-1 (avg=2.14, sd=2.00), and were on average enhanced in comparison to valley soils (Fig. 2). Soil valley CO2 fluxes were found to range between 0.9 and 3.7  $\mu$ mol m-2 s-1 (avg=2.14, sd=0.74) (Fig. 2). Mound adjacent soil CO2 fluxes showed an average emission of 4.84  $\mu$ mol CO2 m-2 s-1 (range=2.0-10.1, sd=2.01), thereby being enhanced with respect to the surrounding soils (Fig. 2). Mound CO2 emissions, corrected for the average valley soil respiration, were ranging between 1.1 and 13.0  $\mu$ mol mound-1 s-1, with an average emission of 8.14  $\mu$ mol mound-1 s-1 (Fig 1).

**In Discussion, §4.3:**

Valley soil CH4 and CO2 fluxes were similar to what was found by earlier studies (Souza (2005), Moura (2012), Chambers et al. (2004), Zanchi et al. (2012). On average, mound adjacent soil CH4 and CO2 fluxes were enhanced with respect to valley soils, although differences were small, and no clear emission pattern with 'distance to mound' was observed. While mound adjacent soil fluxes are possibly enhanced, we preferred to avoid overestimation, and decided to treat termite mounds as very local hot spots, with measured fluxes only representative for the collar area of 0.25 m2. On average, CH4 and CO2 fluxes per collar area were found to be a factor ~630 and ~16 higher when an active termite mound was present.

**References:**

- Carmo, Janaina Braga do, et al. "A source of methane from upland forests in the Brazilian Amazon." Geophysical Research Letters 33.4 (2006).

- Chambers, Jeffrey Q., et al. "Respiration from a tropical forest ecosystem: partitioning of sources and low carbon use efficiency." Ecological Applications 14.sp4 (2004): 72-88.

- Dutaur, Laure, and Louis V. Verchot. "A global inventory of the soil CH4 sink." Global biogeochemical cycles 21.4 (2007).

- Kiese, Ralf, et al. "Seasonal variability of N2O emissions and CH4 uptake by tropical rainforest soils of Queensland, Australia." Global Biogeochemical Cycles 17.2 (2003).

-Moura, V. S. d.: Investigação da variação espacial dos fluxos de metano no solo em floresta de terra firme na Amazônia Central, MSc thesis INPA/UEA, 2012.

- Sihi, Debjani, et al. "Representing methane emissions from wet tropical forest soils using microbial functional groups constrained by soil diffusivity." Biogeosciences Discussions (2020): 1-28.
- Souza, Juliana Silva de. "Dinâmica espacial e temporal do fluxo de CO2 do solo em floresta de terra firme na

Souza, Juliana Silva de. "Dinâmica espacial e temporal do fluxo de CO2 do solo em floresta de terra firme na Amazônia Central." (2005).
 Subke, Jens-Arne, et al. "Rhizosphere activity and atmospheric methane concentrations drive variations of methane

Subke, Jens-Arne, et al. "Rhizosphere activity and atmospheric methane concentrations drive variations of methane fluxes in a temperate forest soil." Soil Biology and Biochemistry 116 (2018): 323-332.
Zanchi, Fabrício B., et al. "Soil CO2 exchange in seven pristine Amazonian rain forest sites in relation to soil

- Zanchi, Fabrício B., et al. "Soil CO2 exchange in seven pristine Amazonian rain forest sites in relation to soil temperature." Agricultural and Forest Meteorology 192 (2014): 96-107.

2. An early study in a Southeast Asian tropical forest showed that the populations of termites was  $3,000 - 4,000 \text{ m}^{-2}$ , 60% of which being wood-feeding termites and 30% being either litter-feeding or humus-feeding species (Chiba, 1978). This population density was supported by many recent studies showed in this manuscript (L356-358). Why this study did not include the major termite species (wood-feeding)?

When designing this field study, we decided to focus only on 1 species, so that effects of interspecies variability could be excluded. In addition, since mound emission was one of the focus points, our preference was to look for an epigeal nest (mound) building species.

Wood-feeding termite species are most likely **not** the major termite species in the Amazon rainforest. The distribution of feeding groups within an assemblage varies around the globe, so while wood-feeding termites might be the major termite species in a Southeast Asian tropical forest (Chiba, 1978), this can be different in other tropical forests.

Jones and Eggleton (2011), compiling data of global biogeography of termites, states that soil-wood interface feeders, such as *N. Brasiliensis*, composes the most diverse and dominant group in Neotropical rainforests (page 491). In addition, the species *N. Brasiliensis* is one of the most common species in our region, and one of the most abundant among mound-builder species (Dambros et al 2016, Pequeno et al. 2013).

In the revised manuscript, we have added the following lines to the Introduction:

**Revised text in Introduction:** In addition, for the Amazon, it is expected that most termites are soil-feeding (Jones and Eggleton, 2011), a group which are expected to be the strongest emitters of  $CH_4$  (Bignell and Eggleton, 2000; Brauman et al., 1992).

**Revised text in Introduction:** In this paper, we are presenting a case study performed in a tropical rain forest in the Amazon, where we measured the emission of  $CH_4$  and other gases of epigeal (above-ground) termite nests of the species *Neocapritermes Brasiliensis*, a soil-feeding species abundant in the Amazon (Constantino, 1992; Pequeno et al., 2013), and one of the most common species in the region (Dambros et al. 2016).

**References:**

- Dambros, Cristian S., et al. "Association of ant predators and edaphic conditions with termite diversity in an Amazonian rain forest." Biotropica 48.2 (2016): 237-245

- Jones, D. T., and P. Eggleton. "Global biogeography of termites: a compilation of sources. In 'Biology of Termites: A Modern Synthesis'.(Eds DE Bignell, Y. Roisin and N. Lo.) pp. 477–498." (2011).

3. Large variations in both CH4 and CO2 emissions (Figure 1; L221-222, L240) among the mounds suggest that the five applicates (mounds) was not enough to represent the ecosystem level CH4 and CO2 emissions. From your statement (2.6: sub sample), I would guess that your CH4/CO2 flux measurements were conducted for all the 19 mounds but not only 5 mounds (Figure 1). If my guess is correct, the authors should explain (in the Method) the reasons for not including the data from other mounds, for example, the other mounds were not active mounds.

From the reviewers comment, we realize that confusion might arise about the amount of mounds sampled. Below we will:

• clarify that we measured fluxes of 5, and not 19, termite mounds;

- clarify how many mound *sub*samples have been measured;
- report additional subsample measurements which confirm the termite emission factor, and present a new Figure 5, which will show these additional measurements;
- provide the improved manuscript text for §2.6, and for other parts of the manuscript.

Our mound selection procedure for the 5 mounds was as follows:

- Firstly, we searched for mounds, which were suitable for flux chamber measurements (sufficient space for collar installation, not attached to tree). We found 20 suitable and active mounds, and we sampled each mound and determined the species at the *Laboratory of Systematics and Ecology of Soil Invertebrates* at INPA. Table 1 in the manuscript gives an overview of the found species per mound.
- When further selecting individual mounds of these 20 mounds, we only choose mounds of the same species, so that effects of interspecies variation could be excluded.
- For practical reasons, we choose a set of mounds which were closely located to each other.
- With these criteria in mind, we selected the mounds from which fluxes would be measured, which were mounds nr. 13, nr. 14, nr. 15, nr. 16 and nr. 19.

**C9**

The choice of limiting our flux measurements to 5 mounds was based on practical considerations (hours of daylight, days in the field, distance to cover), which were especially time constrained due to our additional bag sampling measurements (Appendix A). For a possible follow up study, we would leave this element out.

**Moreover, the authors should explain why the sub sample experiment was only conducted for one mound (L161: "only one sub sample was found suitable from the all 19 mounds").**

The sentence copied by the reviewer is different than the sentence stated at line 161, which was:

**'From the sample from mound 19, only one suitable sub sample was found'**

To clarify: for each of the 5 selected mounds, we sampled one solid (not crumbling) piece, of which we took 3 subsamples, of which we measured emissions and counted termites. In principle, this would lead to 15 subsamples. Nevertheless, due to practical problems at mound 19, we only managed to separate 1 suitable subsample, wherefore the total amount of subsamples was 13, as shown in the original Figure 5 of the manuscript.

In the last few months, we have performed additional measurements:

• Additional measurement 1 (AM1): performed in October 2020 (dry season), with 15 subsamples of the same mounds (mounds nr. 13, nr. 14, nr. 15, nr. 16 and nr. 19). A termite emission factor of 0.0002976 (se= $1.32 \times 10^{-5}$ ) CH4 per termite per second was found.

Additional measurements 2 (AM2): performed in December 2020 (transition dry/wet season), with 5 subsamples, taken from a new mound of the same species. A termite emission factor of 0.0003043 (se=1.41\*10-5) CH4 per termite per second was found.

For the revised manuscript, we have added these CH4 termite emission measurements to the text and to Figure 5, to show the reader the consistency of the termite emission factor between mounds and seasons. Nevertheless, since we prefer to combine only measurements obtained during the same field campaign week, the manuscript estimates and derivations are based on the original determined termite emission factor of 0.0002985 nmol termite-1 s-1.

We have improved Figure 5 (in revised manuscript, renumbered as Fig. 4), which we uploaded, and which can be found at the end of this review. In the text, we have made the following changes:

**Revised text caption Table 1:** Termite mounds: location, dimensions, and observed sp